

# Candidate phases for $SU(2)$ adjoint QCD$_4$ with two flavors from $\mathcal{N} = 2$ supersymmetric Yang-Mills theory

Clay Córdova[1] and Thomas T. Dumitrescu[2]

**1** School of Natural Sciences, Institute for Advanced Study, Princeton, NJ 08540, USA
**2** Department of Physics, Harvard University, Cambridge, MA 02138, USA

## Abstract

We study four-dimensional adjoint QCD with gauge group $SU(2)$ and two Weyl fermion flavors, which has an $SU(2)_R$ chiral symmetry. The infrared behavior of this theory is not firmly established. We explore candidate infrared phases by embedding adjoint QCD into $\mathcal{N} = 2$ supersymmetric Yang-Mills theory deformed by a supersymmetry-breaking scalar mass $M$ that preserves all global symmetries and 't Hooft anomalies. This includes 't Hooft anomalies that are only visible when the theory is placed on manifolds that do not admit a spin structure. The consistency of this procedure is guaranteed by a nonabelian spin-charge relation involving the $SU(2)_R$ symmetry that is familiar from topologically twisted $\mathcal{N} = 2$ theories. Since every vacuum on the Coulomb branch of the $\mathcal{N} = 2$ theory necessarily matches all 't Hooft anomalies, we can generate candidate phases for adjoint QCD by deforming the theories in these vacua while preserving all symmetries and 't Hooft anomalies. One such deformation is the supersymmetry-breaking scalar mass $M$ itself, which can be reliably analyzed when $M$ is small. In this regime it gives rise to an exotic Coulomb phase without chiral symmetry breaking. By contrast, the theory near the monopole and dyon points can be deformed to realize a candidate phase with monopole-induced confinement and chiral symmetry breaking. The low-energy theory consists of two copies of a $\mathbb{CP}^1$ sigma model, which we analyze in detail. Certain topological couplings that are likely to be present in this $\mathbb{CP}^1$ model turn the confining solitonic string of the model into a topological insulator. We also examine the behavior of various candidate phases under fermion mass deformations. We speculate on the possible large-$M$ behavior of the deformed $\mathcal{N} = 2$ theory and conjecture that the $\mathbb{CP}^1$ phase eventually becomes dominant.

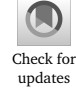

# 1 Introduction

In this paper we analyze four-dimensional adjoint QCD with gauge group $SU(2)$ and two Weyl fermion flavors. We describe several candidate IR phases for this theory, among them a scenario with confinement and chiral symmetry breaking, which results in two copies of a $\mathbb{CP}^1$ sigma model at low energies, as well as more exotic phases where chiral symmetry is unbroken and there are deconfined gauge fields. We motivate these phases by embedding adjoint QCD into a larger parent theory with adjoint scalars and $\mathcal{N} = 2$ supersymmetry. We explore the Coulomb branch of the $\mathcal{N} = 2$ theory in the presence of a certain supersymmetry-breaking scalar mass term, as well as other deformations or modifications that preserve all symmetries and 't Hooft anomalies, and show how each candidate phase emerges from the deformed low-energy theory.

## 1.1 Adjoint QCD in four dimensions

Adjoint QCD with gauge group $SU(2)$ and two fermions belongs to a larger class of theories with $SU(N_c)$ gauge group and $N_f$ massless flavors of two-component Weyl fermions $\lambda^i_\alpha$ in the adjoint representation of $SU(N_c)$.[1] Here $i = 1, \ldots, N_f$ is a flavor index (until further notice, we suppress gauge indices). These theories are asymptotically free for any $N_c$, as long as $N_f \leq 5$. Since the fermions transform in the adjoint representation, all such theories possess a $\mathbb{Z}^{(1)}_{N_c}$ generalized 1-form global symmetry associated with the center of the $SU(N_c)$ gauge group, whose realization in the IR is related to confinement [2,3].[2] Theories with a fixed value of $N_c$ but different values of $N_f$ are related, because the number of flavors can be reduced by adding mass terms for some of the fermions.

We can broadly distinguish adjoint QCD theories with different values of $N_f$ as follows:

- The case $N_f = 0$ corresponds to pure $SU(N_c)$ Yang-Mills (YM) theory. The only adjustable coupling is the $\theta$-angle. The theory is believed to confine, so that all Wilson loops obey an area law.[3] This amounts to the statement that the $\mathbb{Z}^{(1)}_{N_c}$ center symmetry is unbroken [3]. If $\theta = 0$ or $\theta = \pi$, the theory possesses a time-reversal symmetry $\mathsf{T}$. Pure $SU(N_c)$ YM theory with $\theta = 0$ is believed to flow to a trivial phase with a unique vacuum in the deep IR. However, the theory at $\theta = \pi$ has a mixed 't Hooft anomaly that involves $\mathsf{T}$ and the $\mathbb{Z}^{(1)}_{N_c}$ 1-form global symmetry [4], and therefore it cannot be completely trivial at low energies. The simplest scenario is that the theory has a unique vacuum for all $\theta \neq \pi$, but spontaneously breaks $\mathsf{T}$ when $\theta = \pi$, which leads to two degenerate vacua (see [4,5] and references therein). Other possibilities are discussed in [4].

- The case $N_f = 1$ corresponds to pure $\mathcal{N} = 1$ supersymmetric Yang-Mills (SYM) theory with gauge group $SU(N_c)$ (see e.g. [6] for a summary of its basic properties). The theory has a classical $\mathsf{U}(1)_r$ symmetry, under which the adjoint gauginos $\lambda_\alpha$ carry charge 1. It is explicitly broken to $\mathsf{Z}_{2N_c}$ by an Adler-Bell-Jackiw (ABJ) anomaly, so that the $\theta$-angle can be set to zero. The theory is therefore invariant under time reversal $\mathsf{T}$. There are $N_c$ supersymmetric (and hence degenerate) vacua [7], which are the result of spontaneously breaking $\mathsf{Z}_{2N_c} \to \mathsf{Z}_2$ via a gaugino condensate, $\langle \mathrm{tr}(\lambda^\alpha \lambda_\alpha) \rangle \neq 0$.[4] In [9] it was shown that $\mathcal{N} = 1$ SYM theory with $N_c = 2$ confines via monopole condensation (in particular,

---

[1]We use the conventions of [1] for two-component fermions (see also appendix A).

[2]We use a superscript $(p)$ to denote $p$-form symmetries, as well as differential $p$-forms.

[3]This has not been conclusively established for all values of $N_c$ and $\theta$ (see e.g. [4] and references therein).

[4]The fact that the gaugino condensate is necessarily non-vanishing in every supersymmetric vacuum of the $\mathcal{N} = 1$ SYM theory was established in [8].

the $\mathbb{Z}_{N_c}^{(1)}$ center symmetry is unbroken), by embedding it inside pure $\mathcal{N} = 2$ SYM theory with gauge group $SU(2)$. Crucially, the low-energy dynamics of the latter theory could be determined exactly [9,10]. The generalization to $N_c \geq 3$ was found in [11,12]. Perturbing $\mathcal{N} = 1$ SYM theory by a small supersymmetry-breaking gaugino mass $m \in \mathbb{C}$ (whose phase can break $\mathsf{T}$ and generate an effective $\theta$-angle in the resulting low-energy pure YM theory) is consistent with the expected behavior of the $N_f = 0$ theory summarized above (see e.g. [4] and references therein).

- Adjoint QCD with $2 \leq N_f \leq 5$ possesses a continuous $SU(N_f)$ flavor symmetry, under which the fermions $\lambda_\alpha^i$ transform in the fundamental representation. Note that this is a chiral symmetry; only a subgroup of $SU(N_f)$ acts in a vector-like fashion, i.e. on full Dirac fermions. The theory also has a discrete $\mathsf{Z}_{2N_f N_c}$ flavor symmetry, which (as in the $N_f = 1$ case) is the remnant of a classical $\mathsf{U}(1)_r$ symmetry under which the $\lambda_\alpha^i$ have charge 1 that is broken to $\mathsf{Z}_{2N_f N_c}$ by an ABJ anomaly. Therefore $\theta$ can again be set to zero, so that the theory is invariant under time reversal $\mathsf{T}$.

The standard lore (see for instance [13] and references therein) is that theories with larger values of $N_f$ flow to an interacting CFT (similar to the Banks-Zaks phase of conventional QCD with fundamental quarks [14]), while theories with smaller values of $N_f$ undergo confinement and break chiral symmetry via the condensation of a fermion bilinear,

$$\mathcal{O}^{ij} \sim \mathrm{tr}\left(\lambda^{\alpha i}\lambda_\alpha^j\right). \tag{1}$$

When $N_f = 2$, such a chiral condensate implies that the IR theory includes a $\mathbb{CP}^1$ sigma model describing the Nambu-Goldstone (NG) bosons that are the result of spontaneously breaking $SU(2) \to U(1)$. Since the phase of the chiral condensate can take $N_c$ distinct values, there are in fact $N_c$ degenerate copies of a $\mathbb{CP}^1$ model.

Although plausible, it is not known whether the phases described above are actually realized by the dynamics. One is therefore free to contemplate other candidate phases, such as the recent proposal [15], which we will comment on below, and the forthcoming work [16]. Since it is possible to flow from a given value of $N_f$ to lower values by adding large fermion masses, any candidate scenario for the theories with $N_f \geq 2$ should reproduce the expected behavior of the theories with $N_f = 0$ and $N_f = 1$ under such an RG flow.

In this paper we will systematically explore candidate IR phases for $SU(N_c)$ adjoint QCD with $N_f = 2$ flavors that are motivated by softly-broken $\mathcal{N} = 2$ SYM theory. For simplicity we focus on the case $N_c = 2$. For the remainder of this paper we will therefore use the term adjoint QCD to refer to the two-flavor theory with gauge group $SU(2)$ (i.e. we will take $N_f = N_c = 2$).

## 1.2 't Hooft anomalies and the nonabelian spin-charge relation

Before investigating candidate scenarios for adjoint QCD, it is essential to understand the a priori constraints implied by the global symmetries, as well as the associated 't Hooft anomalies, which must match along any RG flow [17] and constitute a powerful constraint on consistent candidate phases. The global symmetries and 't Hooft anomalies of adjoint QCD are described in section 2.

As we will explain in section (2.4), some 't Hooft anomalies of adjoint QCD only become visible when the theory is placed on euclidean 4-manifolds $\mathcal{M}_4$ that do not admit a spin structure. This means that the second Stiefel-Whitney class $w_2(\mathcal{M}_4)$ of $\mathcal{M}_4$ does not vanish. Since adjoint QCD contains fermions, which typically require a spin structure, it may seem surprising that it is possible to consistently place the theory on non-spin manifolds. The fact this can be

done is due to a special property of adjoint QCD, which is also shared by the $\mathcal{N} = 2$ parent theory (see below): fermion parity $(-1)^F$ is identified with the non-trivial central element $1$ of the $SU(2)$ flavor symmetry,

$$(-1)^F = -\mathbb{1} \in SU(2). \tag{2}$$

Concretely, this means that fermions have to transform in half-integer spin representations of the $SU(2)$ flavor symmetry, while bosons can only transform in integer spin representations. One can view (2) as a generalization of the conventional abelian spin-charge relation (see for instance [18]) to $SU(2)$ representations. For this reason we will refer to (2) as a nonabelian spin-charge relation.

As was emphasized in [19], identifications of global symmetries, such as (2), enable us to access a larger class of background field configurations than is naively possible. It follows from (2) that we can consider both non-spin four-manifolds, where $w_2(\mathcal{M}_4) \neq 0$, as well as bundles for the quotient symmetry $SU(2)/\mathbb{Z}_2 = SO(3)$. Such bundles $\mathcal{B}$ are characterized by a second Stiefel-Whitney class $w_2(\mathcal{B})$, which vanishes when $\mathcal{B}$ is an $SU(2)$ bundle. The identification (2) implies that turning on $\mathcal{M}_4$ and $\mathcal{B}$ is only consistent if

$$w_2(\mathcal{B}) = w_2(\mathcal{M}_4). \tag{3}$$

This can be viewed as a generalization of a spin$^c$ structure to $SU(2)$ background gauge fields. The backgrounds in (3) enable to exhibit more 't Hooft anomalies, and hence more constraints on the dynamics of adjoint QCD. For instance, we will see below that the recent proposal [15] must be modified in order to render it consistent with anomaly matching on non-spin manifolds.

## 1.3 Embedding of adjoint QCD into $\mathcal{N} = 2$ supersymmetric Yang-Mills theory

Our primary tool for systematically exploring candidate phases for adjoint QCD is to embed it inside pure $\mathcal{N} = 2$ SYM theory with gauge group $SU(2)$.[5] The details are described in section 2.3. Under this embedding, the adjoint QCD fermions are identified with the $\mathcal{N} = 2$ gauginos. In addition, the supersymmetric theory possesses a complex scalar $\phi$ in the adjoint representation of $SU(2)$.

In order to flow from the parent $\mathcal{N} = 2$ theory to adjoint QCD, we deform the former by a large, supersymmetry-breaking mass term for $\phi$,[6]

$$\Delta V \sim M^2 \, \mathrm{tr}\left(\overline{\phi}\phi\right). \tag{4}$$

Pure $\mathcal{N} = 2$ SYM theory is asymptotically free and therefore characterized by a strong coupling scale $\Lambda$. If we choose $M \gg \Lambda$, the scalar $\phi$ is very heavy and can be reliably integrated out in the UV. Consequently, the mass-deformed $\mathcal{N} = 2$ theory with $M \gg \Lambda$ flows to adjoint QCD in the IR.

It is crucial to our analysis below that the $\mathcal{N} = 2$ theory has the same global symmetries and 't Hooft anomalies as adjoint QCD. For instance, the $SU(2)_R$ symmetry of the $\mathcal{N} = 2$ theory, under which the supercharges $Q_\alpha^i$ and the gauginos $\lambda_\alpha^i$ transform as doublets, is identified with the $SU(2)$ flavor symmetry of adjoint QCD. (From now on we will therefore use the supersymmetric terminology and refer to this symmetry as $SU(2)_R$.) Moreover, unlike other possible mass terms (including those that involve the holomorphic operator $u = \mathrm{tr}\,\phi^2$ discussed below), the deformation in (4) preserves all global symmetries. Thus all symmetries and anomalies are visible for any value of the mass $M$.

---

[5]Basic aspects of $\mathcal{N} = 2$ supersymmetry, including the supersymmetric lagrangians and transformation rules needed throughout this paper, are summarized in appendix B.

[6]The scalar potential of the $\mathcal{N} = 2$ SYM theory has flat directions. Therefore inverting the sign of the mass term in (4) leads to runaway behavior.

The fact that the $\mathcal{N} = 2$ theory and adjoint QCD share the same symmetries and 't Hooft anomalies extends to the nonabelian spin charge relation (2), and the associated 't Hooft anomalies on non-spin manifolds. If we activate a very special configuration of background fields satisfying (3) in the $\mathcal{N} = 2$ SYM theory, we obtain the topologically twisted theory described in [20], whose supersymmetric observables capture the Donaldson invariants (see e.g. [21]) of the 4-manifold $\mathcal{M}_4$. For this reason the global properties of $\mathcal{N} = 2$ SYM on such manifolds have been thoroughly investigated. This includes certain subtle sign factors discussed in [21–25], which only arise on non-spin manifolds. In section 2.4.3, we will interpret some of these signs as arising from a mixed 't Hooft anomaly that involves the $\mathbb{Z}_2^{(1)}$ center symmetry and $w_2(\mathcal{M}_4)$.

Since the scalar mass term (4) completely breaks supersymmetry, it is challenging to quantitatively analyze the deformed $\mathcal{N} = 2$ theory when $M \gg \Lambda$. By contrast, when the scalar mass is small, $M \ll \Lambda$, the effects of the deformation (4) can be determined using the exact, weakly-coupled IR effective description of the $\mathcal{N} = 2$ theory established in [9].

Our philosophy below will be to see what phases for adjoint QCD are suggested by the $\mathcal{N} = 2$ theory if we allow ourselves to naively extrapolate to large $M$, or to continuously vary the IR couplings. Of course it is not possible to use this approach to determine which, if any, of these phases is actually realized by the dynamics. However this approach ensures that such candidate scenarios automatically match all 't Hooft anomalies. One such candidate phase will turn out to be the $\mathbb{CP}^1$ phase with confinement and chiral symmetry breaking discussed around (1). The various candidate phases we consider are summarized in section 1.6 below.

## 1.4 Review of Seiberg-Witten theory

We begin our investigation at $M = 0$ and gradually increase the mass. Our starting point is therefore the $\mathcal{N} = 2$ supersymmetric theory, whose IR behavior was determined exactly in [9]. Here we recall some basic features of this low-energy description (additional details can be found in section 4):

- There is a moduli space of supersymmetric (and therefore degenerate) vacua labeled by the vev $\langle u \rangle \in \mathbb{C}$ of the $\mathcal{N} = 2$ chiral operator $u = \mathrm{tr}(\phi^2)$.[7] This operator transforms as follows under the $\mathsf{Z}_8$ flavor symmetry and time reversal,

$$\mathsf{r}(u) = -u\,, \qquad \mathsf{T}(u) = u\,. \tag{5}$$

- For generic values of $u$, the low-energy theory consists of a single $U(1)$ vector multiplet of $\mathcal{N} = 2$ supersymmetry, which contains the $U(1)$ field strength $f^{(2)}$ and its superpartners, all of which are neutral under the $U(1)$ gauge symmetry,

$$\varphi\,, \qquad \rho_\alpha^i\,, \qquad f^{(2)}\,. \tag{6}$$

Here $\rho_\alpha^i$ is the $U(1)$ gaugino,[8] and $\varphi$ is a complex scalar. Since there is an unbroken $U(1)$ gauge symmetry, the theory is a Coulomb phase, and consequently the moduli space of vacua is known as the Coulomb branch. The couplings of the low-energy theory are functions of $u$. One such coupling is the complexified $U(1)$ gauge coupling

$$\tau = \frac{\theta}{2\pi} + \frac{2\pi i}{e^2}\,. \tag{7}$$

---

[7]We will frequently write $u$ for the vev $\langle u \rangle$ when there is no potential for confusion.

[8]We denote the $U(1)$ gaugino in the deep IR by $\rho_\alpha^i$ in order to distinguish it from the $SU(2)$ gauginos $\lambda_\alpha^i$ of the UV theory.

As we vary $u$, the low-energy physics changes in a smooth fashion, but the $U(1)$ gauge theory may undergo electric-magnetic duality transformations, which act on $\tau$ by $SL(2,\mathbb{Z})$ modular transformations. In the solution of [9], duality acts on a pair $\big(a_D(u), a(u)\big)$ of special coordinates on the Coulomb branch, which transforms as an $SL(2,\mathbb{Z})$ doublet.

Let us summarize the broken and unbroken symmetries on the Coulomb branch:

1.) The $SU(2)_R$ symmetry, under which the gaugino $\rho_\alpha^i$ transforms as a doublet, is unbroken for all $u$.

2.) Since $\mathsf{r}(u) = -u$ (see (5)), the $\mathsf{Z}_8$ symmetry is spontaneously broken to its $\mathsf{Z}_4$, except at the special point $u = 0$. As we will explain later, the $\mathsf{Z}_4$ symmetry acts on $f^{(2)}$ via charge conjugation $\mathsf{C}$, i.e. $\mathsf{r}^2(f^{(2)}) = -f^{(2)} = \mathsf{C}\big(f^{(2)}\big)$.

At the special point $u = 0$ the $\mathsf{Z}_8$ symmetry is restored and its generator acts as a square root of $\mathsf{C}$. In a suitable electric-magnetic duality frame this is the $S$ duality transformation. Consequently, the $U(1)$ gauge coupling at $u = 0$ necessarily takes the self-dual value $\tau = i$.

3.) The $\mathbb{Z}_2^{(1)}$ center symmetry is accidentally enhanced to the $U(1)_{\text{electric}}^{(1)} \times U(1)_{\text{magnetic}}^{(1)}$ 1-form symmetries [3] of the low-energy $U(1)$ gauge theory.[9] Since the theory is in a Coulomb phase, these symmetries (and hence the $\mathbb{Z}_2^{(1)}$ center symmetry of the UV theory) are spontaneously broken, and all Wilson-'t Hooft loops obey a perimeter law.

4.) Since $\mathsf{T}(u) = u$ (see (5)) and $\mathsf{T}$ is anti-unitary, it maps the vev $\langle u \rangle$ to its complex conjugate $(\langle u \rangle)^*$. Time reversal symmetry is therefore preserved on the real axis $\langle u \rangle \in \mathbb{R}$, but it is spontaneously broken when $\text{Im}\langle u \rangle \neq 0$.

- At two special special points $u = \pm\Lambda^2$ on the Coulomb branch,[10] there are additional massless particles. In the duality frame that is adapted to the weakly coupled region $|u| \gg \Lambda^2$, the light particles at $u = \Lambda^2$ are magnetic monopoles whose magnetic and electric charges are given by $(n_m = 1, n_e = 0)$. The light particles at $u = -\Lambda^2$ are dyons of charge $(n_m = 1, n_e = 2)$. Consequently these points in the $u$-plane are known as the monopole and dyon points. They are exchanged by the spontaneously broken $\mathsf{Z}_8$ symmetry.

The additional particles that become light at the monopole point are described by an $\mathcal{N} = 2$ hypermultiplet with field content,

$$h_i, \qquad \psi_{+\alpha}, \qquad \psi_{-\alpha}. \tag{8}$$

The scalar $h_i$ is an $SU(2)_R$ doublet, while the two fermions $\psi_{\pm\alpha}$ are $SU(2)_R$ singlets. In a suitable electric-magnetic duality frame, the hypermultiplet $h_i$ carries electric charge 1 under the low-energy $U(1)$ gauge symmetry. (Its complex conjugate $\bar{h}^i$ carries charge $-1$, while its superpartners $\psi_{+\alpha}$ and $\psi_{-\alpha}$ carry charges 1 and $-1$, respectively.) In the vicinity of $u = \Lambda^2$, the physics at very low energies is therefore described by the renormalizable $\mathcal{N} = 2$ SQED theory consisting of the abelian vector multiplet (6) and the charged hypermultiplet (8).

The scalar potential of this theory is given by

$$V = \frac{e^2}{2}\left(\bar{h}^i h_i\right)^2 + 2|\varphi|^2 \bar{h}^i h_i. \tag{9}$$

---

[9] The precise embedding depends on the duality frame.

[10] Since $\Lambda$ is the only mass scale of the $\mathcal{N} = 2$ theory, it is common to set $\Lambda = 1$. We will not do so here, since we would like to compare $\Lambda$ to the scalar mass $M$ in (4).

Here $\varphi$ has been defined to vanish at the monopole point, and $e$ is the low-energy $U(1)$ gauge coupling, which flows logarithmically to zero at $\varphi = 0$. The potential (9) sets $h_i = 0$, while $\varphi$ is a flat direction that parametrizes the Coulomb branch in the vicinity of the monopole point. Note that the mass of $h_i$ is proportional to $|\varphi|$.

## 1.5  Soft supersymmetry breaking

In order to study the effect of the mass deformation (4) when the mass $M$ is small, $M \ll \Lambda$, we must map the non-holomorphic operator $\mathrm{tr}(\overline{\phi}\phi)$ from the UV to the weakly-coupled Coulomb branch description in the deep IR. The problem of tracking soft supersymmetry-breaking masses along RG flows and across dualities has been thoroughly investigated, see for instance [26–40], and has recently been reinvigorated in the context of three-dimensional dualities [41–44]. The case of pure $\mathcal{N} = 2$ SYM with gauge group $SU(2)$ and a non-holomorphic soft scalar mass was analyzed in [38], and we will reproduce some of their results. We will follow the approach of [40], where non-holomorphic scalar masses were analyzed by tracking operators associated with the stress tensor multiplet of the supersymmetric theory.[11] In fact, the mass deformation that we are interested in is the bottom component $\mathcal{T}$ of the $\mathcal{N} = 2$ stress tensor multiplet [47],[12]

$$\mathcal{T} = \frac{2}{g^2}\,\mathrm{tr}\left(\overline{\phi}\phi\right). \tag{10}$$

Here $g$ is the $SU(2)$ gauge coupling in the UV.[13] Since $\mathcal{T}$ is the bottom component of the $\mathcal{N} = 2$ stress tensor supermultiplet, it can be reliably tracked to the deep IR. The details will be explained in section 5. Here we only summarize the results:

- Away from the monopole and dyon points, the operator $\mathcal{T}$ in (10) flows to the following duality-invariant combination of the special coordinates $a(u)$ and $a_D(u)$,

$$\mathcal{T} = \frac{i}{4\pi}\left(a\overline{a}_D - \overline{a}a_D\right). \tag{11}$$

If we use the exact formulas for $a(u)$ and $a_D(u)$ derived in [9], we find that $\mathcal{T}$ has a unique minimum at the origin $u = 0$ of the Coulomb branch. (This minimum was found in [38].) In particular, it does not display notable features near the monopole or dyon points. (See Figure 1.)

- Near the monopole point, where $\varphi = 0$, the operator $\mathcal{T}$ is given by

$$\mathcal{T} = \frac{1}{e^2}\,|\varphi|^2 - \frac{1}{2}\overline{h}^i h_i - \frac{i\Lambda}{2\pi^2}\left(\varphi - \overline{\varphi}\right) + \cdots. \tag{12}$$

As we will explain in section 5, the linear term $\sim i\left(\varphi - \overline{\varphi}\right)$ is an improvement term for the $\mathcal{N} = 2$ stress tensor multiplet that is needed to match the Coulomb branch expression (11). The ellipsis in (12) denotes higher-order terms that are not captured by the $\mathcal{N} = 2$ SQED description that applies at low energies and small $\varphi$.

Note that the deformation (12) includes a tachyonic mass for $h^i$. This will be important below when we speculate on the possible large-$M$ behavior of the theory.

---

[11]See [45,46] for other exact results that can be obtained by tracking the supersymmetric stress tensor multiplet along RG flows.

[12]The operator $\mathcal{T}$ coincides with the operator $U$ discussed in [45], which exists in certain $\mathcal{N} = 1$ theories. By contrast, the operator $\mathcal{T}$ is generically present in theories with $\mathcal{N} = 2$ supersymmetry.

[13]The factor $\frac{1}{g^2}$ in (10) arises because $\phi$ is conventionally normalized so that its kinetic term contains a non-canonical factor of $\frac{1}{g^2}$. This factor was omitted in (4) in order to simplify the discussion there.

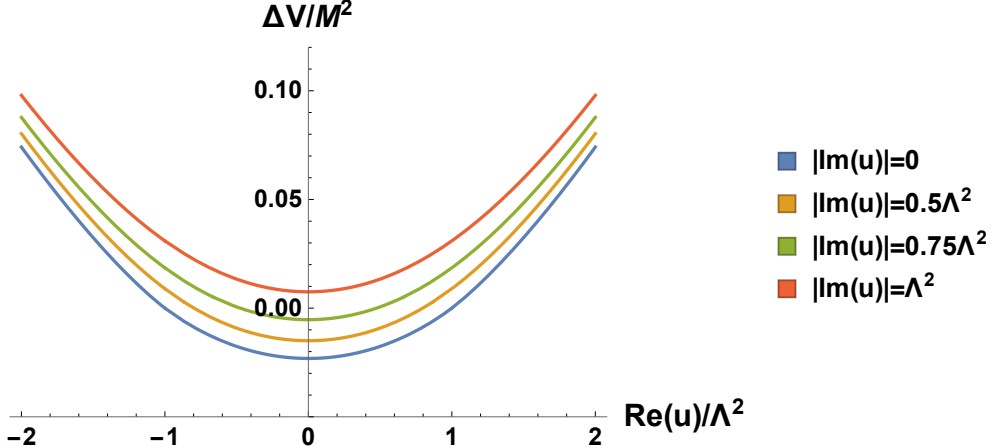

Figure 1: Real slices of the potential function (11) on the Coulomb branch. The function is symmetric about the real axis. The global minimum is at the origin $u = 0$.

It follows from the form of the function in (11) that a scalar mass deformation

$$\Delta V = M^2 \mathcal{T} = \frac{2M^2}{g^2} \operatorname{tr}\left(\overline{\phi}\phi\right), \qquad M \ll \Lambda, \tag{13}$$

lifts all points on the Coulomb branch, except for the origin $u = 0$, which is stable minimum of the function (11) (see Figure 1). The scalar $\varphi$ in the low-energy $U(1)$ vector multiplet (6) acquires a mass $\sim M$, while the gaugino $\rho_\alpha^i$ and the $U(1)$ gauge field $f^{(2)}$ remain massless.

It is important to understand the compatibility of this result with the expression (12) for $\mathcal{T}$ near the monopole point. Classically, the linear term in (12) leads to a local minimum of the potential at $\varphi \approx -ie^2\Lambda$, which would be reliable if $e$ were parametrically small. This, however, is incompatible with the fact that the function (11) shown in Figure 1 has no such local minimum. This superficial mismatch between the two descriptions does not occur, because the effective value of the $U(1)$ gauge coupling $e$ that is implied by the exact solution of [9] is not in fact small. More precisely, $e$ runs logarithmically to zero at the monopole point, but this is not sufficient to overcome its large threshold value at the strong-coupling scale $\Lambda$.

For these reasons, we will not be able to quantitatively analyze the behavior of the $\mathcal{N} = 2$ SQED theory at the monopole point under the deformation (12). Instead, we will use this theory as inspiration to propose a candidate phase for adjoint QCD.

## 1.6 Candidate phases for adjoint QCD

As we have seen, a small soft mass $M$ in (13) stabilizes the theory at the point $u = 0$ of the Coulomb branch. There is a massless gaugino, which is acted on by the unbroken $SU(2)_R$ chiral symmetry, and a massless photon signifying a Coulomb phase (in particular the theory is deconfined). Note that while this scenario may seem exotic, the fact that it emerges from the $\mathcal{N} = 2$ SYM theory in the UV and the weakly coupled Seiberg-Witten description in the deep IR implies that it automatically matches all 't Hooft anomalies.

In this exotic Coulomb phase the $\mathbb{Z}_8$ symmetry is unbroken. Moreover, its realization involves the action of electric-magnetic duality. As long as this symmetry remains unbroken, this means that the coupling is pinned to the self-dual value $\tau = i$. Moreover, r relates any dyon $(n_m, n_e)$ to an exactly degenerate dyon with charges $(n_e, -n_m)$. Some possible implications of these facts will be discussed below.

We would now like to understand whether this phase persists to larger values of $M$, for which we can no longer reliably analyze the soft mass deformation. Broadly speaking, there are three scenarios for what may happen as we increase $M$:

1.) The phase at $u = 0$ may persist for all values of $M$ (in particular for $M \gg \Lambda$). If this is the case, it describes the IR behavior of adjoint QCD. While consistent, this scenario (in particular the fact that $\tau = i$ and the realization of the unbroken $\mathsf{Z}_8$ symmetry via electric-magnetic duality) is quite exotic. Note however, that it is not possible to simply remove the $U(1)$ gauge field from the spectrum (e.g. by higgsing) since its presence is tied to the realization of the $\mathsf{Z}_8$ symmetry via $S$-duality.

   If this scenario is realized in adjoint QCD, we must invoke a transition to a confining phase once we add fermion masses, in order to match the known behavior of adjoint QCD with $N_f = 0, 1$ (see section 1.1).[14]

2.) The vacuum at $u = 0$ persists, but there is a second order phase transition at which additional massless degrees of freedom appear.

3.) There is a first-order transition to a different vacuum.

Without some additional guidance, it is difficult to determine which of the options listed above is realized. In particular, it is not clear what other states or vacua should be considered in options 2.) and 3.). It is therefore desirable to have a better sense of the consistent candidate phases that could conceivably be realized once we explore the deformed $\mathcal{N} = 2$ SYM theory for larger values of the soft scalar mass $M$.

Since every point on the Coulomb branch automatically satisfies 't Hooft anomaly matching, it represents a consistent candidate phase. Moreover, even if these points are not minima of the potential at parametrically small values of $M$, the potential at larger values of $M$ could take a different shape and favor points other that $u = 0$ on the Coulomb branch. Of course it is entirely possible that the physics at larger values of $M$ is completely divorced from any Coulomb-branch intuition. In this case we have very little to say.

With this in mind, we will explore different points on the $u$-plane. Moreover, we allow ourselves to contemplate modifications of the low-energy theory at these points, as long as these modifications preserve all symmetries and 't Hooft anomalies. For instance, at the monopole point we will permit ourselves to investigate the regime where the gauge coupling of the $\mathcal{N} = 2$ SQED theory is small, $e \ll 1$. We make two additional simplifying, but well-motivated, assumptions:

- We focus on the real axis $u \in \mathbb{R}$. Vacua with $\mathrm{Im}\, u \neq 0$ break time reversal.[15]

- We focus on the strong-coupling region $|u| \lesssim \Lambda^2$. At sufficiently large values of $u$, the scalar mass deformation reliably produces a large potential that pushes us into the interior of the $u$-plane. This is reflected in Figure 1.

### 1.6.1 A phase with confinement and unbroken chiral symmetry

Let us first discuss a generic point $u \neq 0, \pm\Lambda^2$ on the real axis. Even though all such points are lifted once we turn on the soft scalar mass $M$, they still furnish a candidate phase that matches all 't Hooft anomalies. Moreover, it is possible that the shape of the effective potential changes

---

[14]In the context of Seiberg-Witten theory, adding soft fermion masses typically drives the theory toward the monopole and dyon points. For instance, this is the case for the $\mathcal{N} = 1$ preserving deformation in [9]. More general soft fermion masses are analyzed in [38].

[15]It is sometimes possible to justify this assumption by adapting the arguments in [48], see e.g. section 3.4.

as we increase $M$, so that these points are eventually favored. In the $\mathcal{N} = 2$ theory, the low-energy field content is given by the abelian vector multiplet $\left(\varphi, \rho_\alpha^i, f^{(2)}\right)$. The $\mathsf{Z}_8$ symmetry is spontaneously broken to $\mathsf{Z}_4$ and the low-energy gauge coupling $\tau$ need not take a particular fine-tuned value.[16]

We can now attempt to deform the theory while preserving all symmetries and 't Hooft anomalies. To start, we can imagine that the scalars acquire a mass. A more subtle question is whether it is possible to deform the theory so as to remove the $U(1)$ gauge field. For instance, it was recently conjectured [15] that adjoint QCD (with $N_f = N_c = 2$) flows to a theory with two vacua that spontaneously break $\mathsf{Z}_8$ to $\mathsf{Z}_4$ and only the gaugino $\rho_\alpha^i$, but no gauge fields. This conjecture was motivated by the known behavior of adjoint QCD on a small $S^1$ (with periodic, i.e. non-thermal, boundary conditions for the fermions) established in [49, 50] (see for instance [51] for a recent review with references).

In order to remove the $U(1)$ gauge field in a way that can be interpreted as confinement from the point of the UV theory (in particular, the $\mathbb{Z}_2^{(1)}$ center symmetry should be unbroken), it is appropriate to use a duality frame that is adapted to the monopole point. As was explained in [23, 24] (and will be reviewed in section 4.3), this dual $U(1)$ gauge field $f_D^{(2)}$ is in fact a spin$^c$ gauge field. (This is ultimately a consequence of the mixed anomaly involving the $\mathbb{Z}_2^{(1)}$ center symmetry and $w_2(\mathcal{M}_4)$ mentioned in section 1.2.) It is therefore inconsistent to completely remove this $U(1)$ gauge field by higgsing.

To see this, imagine lifting the photon by adding heavy charged scalar fields. If such a field condenses, it can higgs the $U(1)$ gauge symmetry. However since $f_D^{(2)}$ is a spin$^c$ gauge field it can only couple to odd-charge Higgs fields in half-integer-spin representations of $SU(2)_R$ (e.g. doublets), or to even-charge Higgs fields in integer-spin representations of $SU(2)_R$. If one is seeking a scenario with unbroken $SU(2)_R$ symmetry, the only option is an $SU(2)_R$-neutral Higgs field of even $U(1)$ gauge charge, which can at most Higgs the $U(1)$ gauge field to a $\mathbb{Z}_2$ gauge field. Moreover, since the massive Higgs field does not contribute to any anomalies, this argument also shows that the resulting $\mathbb{Z}_2$ gauge theory (together with the massless gaugino $\rho_\alpha^i$) is sufficient to match all 't Hooft anomalies.

The resulting candidate phase nicely matches the results of [49, 50], because adjoint QCD on a small $S^1$ is known to posses two vacua. These could then be interpreted as the two possible values of the $\mathbb{Z}_2$ Wilson line on $S^1$. As was already mentioned above, the fact that we have higgsed the dual gauge field $f_D^{(2)}$ implies that the UV theory is confined. In particular, all $SU(2)$ Wilson loops obey an area law, and the $\mathbb{Z}_2^{(1)}$ center symmetry is unbroken. Nevertheless there is an emergent, deconfined $\mathbb{Z}_2$ gauge theory in the IR, and it gives rise to loop operators that do not obey an area law.

We would like to make two additional comments about this phase:

- As was the case for the exotic Coulomb phase at $u = 0$, we must invoke phase transitions under fermion mass deformations to make the candidate phase discussed here consistent with the known behavior of $N_f = 0, 1$ adjoint QCD.

- In the discussion above we postulated the existence of scalar Higgs field that couples to $f_D^{(2)}$ with charge 2. In terms of the UV variables, this amounts to a magnetic monopole of charge 2. This also fits nicely with the results of [49, 50], where such magnetic charge 2 excitations become available once the theory is compactified. However, there is no evidence for the existence of such an excitation in the spectrum of the four-dimensional $\mathcal{N} = 2$ SYM theory. For instance, the BPS particle spectrum was determined in [9, 52] and consists of particles of magnetic charge zero or one.

---

[16]As we will discuss in section 4.1, the $\theta$-angle in the region $|u| < 1$ is $\pi$, up to a duality transformation, while it vanishes when $|u| > 1$, up to a duality transformation.

### 1.6.2 The $\mathbb{CP}^1$ phase

Finally, we consider the monopole (and, using $\mathsf{Z}_8$, also the dyon) point. As was already explained in section 1.3, there is no reliable vacuum at these points when $M$ is small. Instead, we use the same logic as above and modify the theory without changing its symmetries or 't Hooft anomalies. In this case we make the gauge coupling $e$ parametrically small, so that the $\mathcal{N} = 2$ SQED theory at the monopole point becomes weakly coupled and can be analyzed. The outcome of this analysis is another candidate phase of adjoint QCD, which turns out to be the $\mathbb{CP}^1$ sigma model anticipated around (1).

If we deform this SQED theory (with small $e$ and scalar potential (9)) by the operator (12), we find that the tachyonic mass term for the hypermultiplet scalar in (12) eventually leads to a non-zero vev $\langle h_i \rangle \neq 0$. Since $h_i$ carries $U(1)$ gauge charge 1, it completely higgses the gauge symmetry. In terms of the UV variables, the theory confines, and the $\mathbb{Z}_2^{(1)}$ center symmetry is unbroken. However, because $h_i$ also transforms in an $SU(2)_R$ doublet, this global symmetry is spontaneously broken to its $U(1)_R$ subgroup. The model also preserves the time-reversal symmetry $\widetilde{\mathsf{T}} = \mathsf{Tr}^2$, even though $\mathsf{r}^2$ and $\mathsf{T}$ are separately broken. By tracking a supersymmetric descendant of the chiral operator $u$ from the UV to the IR, one can show that the vev of $h_i$ induces a UV gaugino condensate $\mathrm{tr}(\lambda^i \lambda^j) \sim \overline{h}^{(i} h^{j)} \neq 0$. It is remarkable that the IR theory on the Coulomb branch has the right degrees of freedom to realize this scenario.

The low-energy theory of the NG bosons for the spontaneous breaking $SU(2)_R \to U(1)_R$ is a $\mathbb{CP}^1$ sigma model. This theory has two interesting topological couplings that arise from $\mathcal{N} = 2$ SQED by integrating out the massive fermions. Both of them have important implications for the solitonic states of the model:

- There is a discrete $\theta$-angle associated with $\pi_4(\mathbb{CP}^1) = \mathbb{Z}_2$ [53, 54]. The model has skyrmion particles associated with the Hopf map $\pi_3(\mathbb{CP}^1) = \mathbb{Z}$. (See for instance [55] for a discussion in the context of adjoint QCD.) In fact, skyrmion number is only conserved modulo 2 (for reasons closely related to those discussed in [56]), and coincides with fermion parity $(-1)^F$, due to the presence of the discrete theta term. Moreover, both quantum numbers also coincide with the central element $-\mathbb{1} \in SU(2)_R$. Therefore the skyrmions have exactly the right quantum numbers to describe hadronic states created by local operators constructed out of gauginos $\lambda_\alpha^i$ and gauge fields.

- There is also an ordinary $\theta$-angle of the form $\Omega \wedge \Omega$, where $\Omega$ is the pullback to spacetime of the $\mathbb{CP}^1$ Kähler form.[17] It takes the value $\theta = \pi$, which is consistent with time reversal invariance.

  The model has solitonic strings associated with $\pi_2(\mathbb{CP}^1) = \mathbb{Z}$, a $\mathbb{Z}_2$ subgroup of which is identified with the $\mathbb{Z}_2^{(1)}$ center symmetry. Therefore the strings are stable modulo 2. These are the confining strings of the theory. (In the $\mathcal{N} = 2$ SQED description, they are ANO strings.) When we turn on background fields for the unbroken $U(1)_R \subset SU(2)_R$, we find that the $\theta$-term for the pions induces a $\theta$-term of the form $\frac{\theta_R}{2\pi} G_R^{(2)}$ on the string worldsheet, with $\theta_R = \pi$. Here $G_R^{(2)}$ is the background spin$^c$ field strength associated with the $U(1)_R$ symmetry. This means that, though gapped, the worldsheet theory on the confining string is in fact a topological insulator. It is protected by the unbroken $U(1)_R$ and time reversal $\widetilde{\mathsf{T}}$ symmetries.

So far we have only discussed the theory near the monopole point. The theory near the dyon point is identical. Therefore we get two copies of the $\mathbb{CP}^1$ model related by the sponta-

---

[17]Whether or not such a theta term is generated depends on the sign of the vev of $\varphi$. A non-trivial $\theta$-angle is generated if $\varphi$ lies inside the strong-coupling region. This is plausible, given the form of the scalar potential discussed above.

neously broken $Z_8$ symmetry. This is precisely what is expected from chiral symmetry breaking by a gaugino condensate. Unlike the phases without chiral symmetry breaking discussed above, the $\mathbb{CP}^1$ phase very economically captures the expected properties of the $N_f = 0, 1$ theories when we add small fermion masses, which can be analyzed in the $\mathbb{CP}^1$ model using spurions:

- If we add generic masses, we flow to pure YM theory with $\theta \neq \pi$. In this case we find a unique vacuum.

- If we add mass for only one gaugino, we find the two degenerate vacua expected in $\mathcal{N} = 1$ SYM theory.

- We can add the same complex mass $m$ for both gauginos, which preserves the $U(1)_R$ symmetry. The phase of $m$ determines the $\theta$-angle of the resulting YM theory at low energies. As long as $\theta \neq \pi$, we find a single vacuum, but when $\theta = \pi$, we find two exactly degenerate vacua (one on each $\mathbb{CP}^1$), as expected.

  When $m > 0$, we preserve the $U(1)_R$ and $\widetilde{\mathsf{T}}$ symmetries that protect the topological insulator on the worldsheet of the confining string. However, for sufficiently large $m$ we flow to pure YM at $\theta = 0$, whose strings are expected to have trivial ground states. This is only possible if the theory on the string undergoes a phase transition (while the bulk remains gapped). The simplest possibility is that a single massive Dirac fermion on the string worldsheet becomes massless. This is precisely the scenario that occurs when the mass-deformed $\mathbb{CP}^1$ phase is realized in the deformed $\mathcal{N} = 2$ SQED theory (see section (5)). There the massless Dirac fermion that is responsible for the transition on the string worldsheet arises from conventional fermion zero modes on Abrikosov-Nielsen-Olesen (ANO) strings.

### 1.6.3 Speculations on phase transitions

Having analyzed possible phases for adjoint QCD suggested by Seiberg-Witten theory, we are left to speculate which phase is ultimately realized. As was emphasized before, this is a question of energetics (i.e. the shape of the effective potential), and general constraints such as 't Hooft anomaly matching are not sufficient to distinguish between different candidate scenarios. Even though we cannot do this in a controlled fashion, we permit ourselves the following speculations:

- Perhaps the most striking feature of the mass deformation (12) is that it gives a tachyonic mass to the hypermultiplet $h_i$. If this behavior persists at larger values of $M$, the hypermultiplet will eventually become light and condense. If this crude, but suggestive picture can be taken seriously, it is natural to conjecture that the $\mathbb{CP}^1$ phase is eventually reached for large enough values of $M$.

- Even if the $\mathbb{CP}^1$ phase ultimately takes over, there is a question of how this happens. One possibility is that there is a first-order transition, i.e. the vacuum at $u = 0$ disappears and the two $\mathbb{CP}^1$ vacua appear somewhat closer to the monopole and dyon points. One could also imagine passing through other intermediate phases without chiral symmetry breaking, such as the one discussed in section 1.6.1.

- A more dramatic scenario is that the theory remains at $u = 0$ until the monopoles and dyons become massless. The unbroken $Z_8$ symmetry there relates monopoles and dyons with exactly the right quantum numbers to be identified as the particles that become light at the points $u = \pm \Lambda^2$ (see section (4.2.2)). If the picture suggested by the form of the mass deformation (12) is correct, increasing $M$ will make them lighter and lighter,

but they will stay exactly degenerate as long as the theory is at $u = 0$. If the monopole and the dyon (which are mutually non-local) become simultaneously massless, there is necessarily an interacting CFT at the transition point [57]. If the naive picture suggested by the form of (12) continues, the theory will then transition out of the CFT and into the $\mathbb{CP}^1$ phase.

Our work using soft supersymmetry breaking of supersymmetric field theories to explore candidate phases of non-supersymmetric gauge theories can be generalized to a variety of other models. See for instance [58–60] for recent work on increasing the number of colors or otherwise changing the gauge group, [61, 62] for recent examples attempting to reduce the initial amount of supersymmetry from $\mathcal{N} = 2$ to $\mathcal{N} = 1$, and [63, 64] for recent examples changing the dimension of the theory under investigation.

# 2 The UV theory: $SU(2)$ adjoint QCD with $N_f = 2$ flavors

In this section we review the lagrangian and global symmetries of adjoint QCD and its $\mathcal{N} = 2$ supersymmetric extension. We also describe in detail the various 't Hooft anomalies that constrain RG flows, including some that become visible only when the theory is placed on non-spin manifolds.

## 2.1 Lagrangian

Adjoint QCD with gauge group $SU(2)$ and $N_f = 2$ massless flavors is described by the following classical lagrangian,[18]

$$\mathscr{L} = -\frac{1}{4g^2} v^{A\mu\nu} v^A_{\mu\nu} - \frac{i}{g^2} \overline{\lambda}^A_i \overline{\sigma}^\mu D_\mu \lambda^{iA} + \frac{\theta}{64\pi^2} \varepsilon^{\mu\nu\rho\lambda} v^A_{\mu\nu} v^A_{\rho\lambda}. \tag{14}$$

Let us discuss the parameters and fields that appear in this lagrangian:

- $g$ is the dimensionless gauge coupling. In the quantum theory it runs and is traded for a physical strong coupling scale $\Lambda$. At one loop $\frac{1}{g^2} \to \frac{\beta}{8\pi^2} \log \frac{\Lambda_{\text{UV}}}{\Lambda}$, where $\beta = \frac{14}{3}$ and $\Lambda_{\text{UV}}$ is a UV cutoff. The theory is therefore asymptotically free.

- $v^A_{\mu\nu}$ is the $SU(2)$ field strength. Here $A = 1, 2, 3$ is an adjoint triplet index of $SU(2)$.[19] In terms of the $SU(2)$ gauge field $v^A_\mu$,

$$v^A_{\mu\nu} = \partial_\mu v^A_\nu - \partial_\nu v^A_\mu + \varepsilon_{ABC} v^B_\mu v^C_\nu. \tag{15}$$

Here $\varepsilon_{ABC}$ is the totally antisymmetric symbol, normalized as $\varepsilon_{123} = 1$. We will occasionally use the $SU(2)$ generators $t^A = \frac{1}{2}\sigma^A$ in the fundamental representation (here $\sigma^A$ are the Pauli matrices). They are hermitian and satisfy

$$\left[t^A, t^B\right] = i\varepsilon_{ABC} t^C, \qquad \text{tr}\left(t^A t^B\right) = \frac{1}{2}\delta^{AB}. \tag{16}$$

The connection 1-form $v^{(1)}$ and the field-strength 2-form $v^{(2)}$ are then defined as follows,

$$v^{(1)} = v^A_\mu t^A dx^\mu, \qquad v^{(2)} = \frac{1}{2} v^A_{\mu\nu} t^A dx^\mu \wedge dx^\nu = dv^{(1)} - iv^{(1)} \wedge v^{(1)}. \tag{17}$$

---

[18]We work in Minkowski signature, so that the path integral weight is $e^{iS}$ with $S = \int d^4x \, \mathscr{L}$. Throughout, we use the conventions of [1].

[19]We use uppercase latin letters $A, B, C, \ldots$ from the beginning of the alphabet to denote adjoint triplet indices of the $SU(2)$ gauge group. We will not distinguish between raised and lowered indices, since we can freely change the placement of any index using $\delta^{AB}, \delta_{AB}$.

- There are $N_f = 2$ left-handed Weyl fermions $\lambda_\alpha^{iA}$ in the adjoint representation of the $SU(2)$ gauge group. The index $i = 1, 2$ is a flavor doublet index.[20] The right-handed hermitian conjugate of $\lambda_\alpha^{iA}$ is

$$\overline{\lambda}_{\dot\alpha i}^A = \left(\lambda_\alpha^{iA}\right)^\dagger. \tag{18}$$

Note that hermitian conjugation exchanges raised and lowered flavor indices (see appendix A). For completeness we write out the covariant derivative in (14),

$$D_\mu \lambda_\alpha^{iA} = \partial_\mu \lambda_\alpha^{iA} + \varepsilon_{ABC}\, v_\mu^B \lambda_\alpha^{iC}. \tag{19}$$

In the $\mathcal{N} = 2$ SYM parent theory (see section 2.3), the $\lambda_\alpha^{iA}$ are identified as gauginos. It is therefore natural to normalize them as in (14), so that their kinetic term contains a factor of $\frac{1}{g^2}$.

- As we will review in section 2.2, the classical $\theta$-parameter in (14) is absent in the quantum theory. We nevertheless include it here, because it will be useful when we discuss 't Hooft anomalies in section 2.4, and mass deformations in section 2.5.

The $\theta$-parameter appears in the functional integral as $e^{i\theta n}$, where $n$ is the $SU(2)$ instanton number on the spacetime 4-manifold $\mathcal{M}_4$,

$$n = \frac{1}{8\pi^2}\int_{\mathcal{M}_4} \mathrm{tr}\left(v^{(2)} \wedge v^{(2)}\right) = \frac{1}{64\pi^2}\int_{\mathcal{M}_4} d^4x\, \varepsilon^{\mu\nu\rho\lambda} v_{\mu\nu}^A v_{\rho\lambda}^A. \tag{20}$$

The quantization of $n$ on orientable 4-manifolds[21] is as follows (see e.g. [65–67]):

- If the gauge group is $SU(2)$, the instanton number $n \in \mathbb{Z}$.
- If the gauge group is $SO(3)$ and the 4-manifold is spin, then $2n \in \mathbb{Z}$.
- If the gauge group is $SO(3)$ and the 4-manifold is not spin, then $4n \in \mathbb{Z}$.

In this paper we will focus on two-flavor adjoint QCD with gauge group $SU(2)$. Therefore $\theta$ is an angle with standard periodicity $\theta \sim \theta + 2\pi$. However, when we analyze the 't Hooft anomalies of this theory in section 2.4, it will occasionally be useful to refer to the theory with $SO(3)$ gauge group.

## 2.2 Symmetries

In addition to Poincaré symmetry, adjoint QCD with $SU(2)$ gauge group and $N_f = 2$ massless flavors has several global symmetries, which we now describe in turn.

### 2.2.1 $SU(2)_R$ flavor symmetry

There is an $SU(2)_R$ symmetry under which the fermions $\lambda_\alpha^{Ai}$ transform as doublets. The flavor index $i = 1, 2$ is the corresponding doublet index. We refer to the symmetry as an $R$-symmetry, because this terminology is standard in the $\mathcal{N} = 2$ SYM parent theory (see section 2.3), where $SU(2)_R$ acts on the supercharges.

---

[20]We use lowercase latin letters $i, j, k, \dots$ from the middle of the alphabet to denote flavor doublet indices. These indices are acted on by an $SU(2)_R$ flavor symmetry (see section 2.2). It is therefore natural to raise and lower them using the antisymmetric invariant symbols $\varepsilon^{ij}, \varepsilon_{ij}$ of $SU(2)_R$ (see appendix A).

[21]As we will mention in section 2.2.4, the fact that the theory preserves parity and time-reversal means that it is possible to study it on unorientable 4-manifolds.

The $\mathbb{Z}_2$ center of $SU(2)_R$ is generated by the element $-\mathbb{1} \in SU(2)_R$. This element multiplies all fermions by a sign and is therefore be identified with fermion parity $(-1)^F$

$$-\mathbb{1} = (-1)^F \,. \tag{21}$$

As we describe in section 2.4.2 this is an analog of the spin-charge relation discussed in [18] for $SU(2)_R$ representations.

### 2.2.2   $\mathsf{Z}_8$ flavor symmetry

The classical lagrangian (14) has a $\mathsf{U}(1)_r$ flavor symmetry under which the fermions $\lambda_\alpha^{Ai}$ have charge 1. More precisely, the flavor symmetry that acts faithfully on the fermions is

$$U(2) = \frac{SU(2)_R \times \mathsf{U}(1)_r}{\mathbb{Z}_2} \,. \tag{22}$$

The quotient identifies the element $-1 \in \mathsf{U}(1)_r$ with the central element $-\mathbb{1} \in SU(2)_R$. As in (21), both elements are further identified with fermion parity,

$$-1 = -\mathbb{1} = (-1)^F \,. \tag{23}$$

Quantum mechanically, the $\mathsf{U}(1)_r$ symmetry is explicitly broken to its $\mathsf{Z}_8$ subgroup by an Adler-Bell-Jackiw (ABJ) anomaly. This leads to the following non-conservation equation for the classically conserved $\mathsf{U}(1)_r$ current,

$$j_r^\mu = \frac{1}{g^2} \bar{\lambda}_i^A \bar{\sigma}^\mu \lambda^{iA} \,, \qquad \partial_\mu j_r^\mu = \frac{1}{8\pi^2} \varepsilon^{\mu\nu\rho\lambda} v_{\mu\nu}^A v_{\rho\lambda}^A \,. \tag{24}$$

Therefore a $\mathsf{U}(1)_r$ rotation by an angle $\chi_r \sim \chi_r + 2\pi$ shifts the $\theta$-angle in (14) as follows,

$$\theta \;\rightarrow\; \theta + 8\chi_r \,. \tag{25}$$

We can use such a rotation to set $\theta = 0$, which explains why it is not a parameter of the quantum theory. Since both $\theta$ and $\chi_r$ have periodicity $2\pi$, it follows from (25) that the ABJ anomaly explicitly breaks

$$\mathsf{U}(1)_r \quad\longrightarrow\quad \mathsf{Z}_8 \,. \tag{26}$$

The unbroken $\mathsf{Z}_8$ symmetry is generated by $\mathsf{U}(1)_r$ rotations with angle $\chi_r = \frac{\pi}{4}$. We will denote the $\mathsf{Z}_8$ generator by r and use multiplicative notation, so that

$$\mathsf{r}\big(\lambda_\alpha^{iA}\big) = e^{\frac{i\pi}{4}} \lambda_\alpha^{iA} \,. \tag{27}$$

Since $\mathsf{r}^4 = -1 \in \mathsf{U}(1)_r$, it follows from (23) that r satisfies the following relations,

$$\mathsf{r}^4 = -\mathbb{1} = (-1)^F \,. \tag{28}$$

The ABJ anomaly therefore explicitly breaks the classical $U(2)$ flavor symmetry in (22) to

$$\frac{SU(2)_R \times \mathsf{Z}_8}{\mathbb{Z}_2} \,, \tag{29}$$

where the $\mathbb{Z}_2$ quotient enforces the identification in (28).[22]

---

[22]A further quotient that also involves the Lorentz group is needed to enforce the identification with $(-1)^F$. This will be important in section 2.4.2.

### 2.2.3 $\mathbb{Z}_2^{(1)}$ 1-form center symmetry

Since we are studying $SU(2)$ gauge theory with matter in the adjoint representation (the fermions $\lambda_\alpha^{iA}$), but no fundamental matter, there is a generalized 1-form global symmetry $\mathbb{Z}_2^{(1)}$ associated with the center of the $SU(2)$ gauge group [2,3].[23] It is common to refer to $\mathbb{Z}_2^{(1)}$ as a center symmetry. The action of the $\mathbb{Z}_2^{(1)}$ symmetry on a Wilson line $W_j$ in the spin-$j$ representation of $SU(2)$ (with $j \in \frac{1}{2}\mathbb{Z}$) is $W_j \to (-1)^{2j} W_j$.

As is the case for an ordinary flavor symmetry, the vacuum of the theory may or may not preserve the $\mathbb{Z}_2^{(1)}$ center symmetry, i.e. it can be unbroken or spontaneously broken. As was explained in [3], this is closely related to whether or not the theory confines. If all Wilson loops have an area law, then $\mathbb{Z}_2^{(1)}$ is unbroken. By contrast, if $\mathbb{Z}_2^{(1)}$ is spontaneously broken, then some $W_j$ with half-integer $j$ must have a perimeter law.

### 2.2.4 Parity P and time-reversal T

As was explained around (25), the $\theta$-angle in (14) can be set to zero using the ABJ anomaly of the classical $U(1)_r$ symmetry. The theory is therefore invariant under parity P and time-reversal T. In lorentzian signature P is unitary and T is anti-unitary. Since we are discussing a theory with gauge group $SU(2)$ (which admits no outer automorphisms), there is no separate notion of charge conjugation C. In this situation the CPT-theorem implies that P and T can be used interchangeably.

Lorentz invariance requires the action of P and T on two-component spinors $\psi_\alpha, \overline{\psi}_{\dot{\alpha}}$ to take the following form,

$$\mathsf{P}(\psi_\alpha) \sim \overline{\psi}^{\dot{\alpha}}, \qquad \mathsf{T}(\psi_\alpha) \sim \psi^\alpha. \tag{30}$$

It is natural to choose P and T to commute with the unitary operators that implement $SU(2)_R$ flavor transformations. Since P is unitary, it preserves the placement of $SU(2)_R$ doublet indices. By contrast, T is anti-unitary and therefore exchanges raised and lowered $SU(2)_R$ indices. Schematically,

$$\mathsf{P}(\mathcal{O}^i) \sim \mathcal{O}^i, \qquad \mathsf{T}(\mathcal{O}^i) \sim \mathcal{O}_i. \tag{31}$$

With these comments in mind, it can be checked that the following parity and time-reversal transformations are symmetries of the lagrangian (14) (once we set $\theta = 0$),

$$\begin{aligned}
\mathsf{P}(v_\mu^A(x)) &= \mathscr{P}_\mu{}^\nu v_\nu^A(\mathscr{P}x), & \mathsf{T}(v_\mu^A(x)) &= \mathscr{T}_\mu{}^\nu v_\nu^A(\mathscr{T}x), \\
\mathsf{P}(\lambda_\alpha^{iA}(x)) &= \overline{\lambda}^{\dot{\alpha}iA}(\mathscr{P}x), & \mathsf{T}(\lambda_\alpha^{iA}(x)) &= i\lambda_i^{\alpha A}(\mathscr{T}x).
\end{aligned} \tag{32}$$

Here $\mathscr{P}_\mu{}^\nu = \mathrm{diag}(1,-1,-1,-1)$ and $\mathscr{T}_\mu{}^\nu = \mathrm{diag}(-1,1,1,1)$ are the parity and time-reversal Lorentz transformation matrices. It follows from (32) that

$$\mathsf{P}^2 = \mathsf{T}^2 = 1. \tag{33}$$

Note that $\mathsf{T}^2$ squares to 1, rather than to the more common $(-1)^F$, because the fermions $\lambda_\alpha^{iA}$ transform in the pseudo-real doublet representation of $SU(2)_R$. This is also responsible for the identification (21).

Although we will not utilize it below, the fact that adjoint QCD preserves parity and time-reversal means that it is possible to place the theory on non-orientable 4-manifolds. More specifically (33) means that (after analytic continuation to euclidean signature) the class of four-manifolds accessible are those with a Pin$^-$ structure (see e.g. [69].) In section 2.4.2 we will see that the class of 4-manifolds can be further enlarged if we include $SU(2)_R$ backgrounds.

---

[23]Here we use the notation of [68] for higher-form symmetries, whose form degree is indicated by a superscript in parentheses, but not for ordinary 0-form flavor symmetries.

## 2.3 Embedding into $\mathcal{N} = 2$ supersymmetric Yang-Mills theory

We now explain how adjoint QCD can be embedded into $\mathcal{N} = 2$ SYM theory. The basics of $\mathcal{N} = 2$ supersymmetry are reviewed in appendix B. Here we only present those aspects that will be needed below. The $\mathcal{N} = 2$ supersymmetry algebra takes the following form,

$$\left\{ Q_\alpha^i, \overline{Q}_{\dot\alpha j} \right\} = 2\delta^i{}_j \sigma^\mu_{\alpha\dot\alpha} P_\mu \,, \qquad \left\{ Q_\alpha^i, Q_\beta^j \right\} = 2\varepsilon_{\alpha\beta} \varepsilon^{ij} \overline{Z} \,. \tag{34}$$

Here $Q_\alpha^i$ and $\overline{Q}_{\dot\alpha i} = \left( Q_\alpha^i \right)^\dagger$ are the $\mathcal{N} = 2$ supercharges and $Z$ is a complex central charge, with hermitian conjugate $\overline{Z} = Z^\dagger$.

Pure $\mathcal{N} = 2$ SYM theory with gauge group $SU(2)$ is based on a single $\mathcal{N} = 2$ vector multiplet, whose component fields are given by

$$v_\mu^A \,, \qquad \phi^A \,, \qquad \lambda_\alpha^{iA} \,, \qquad D^{ijA} = D^{(ij)A} = \left( D_{ij}^A \right)^\dagger \,. \tag{35}$$

All fields are in the adjoint representation of $SU(2)$, and therefore carry a triplet index $A = 1, 2, 3$. The superpartners of the gauge field $v_\mu^A$ consist of complex scalars $\phi^A$ (whose hermitian conjugates we denote by $\overline{\phi}^A$), an $SU(2)_R$ doublet of gauginos $\lambda_\alpha^{iA}$ (whose hermitian conjugates we denote by $\overline{\lambda}_{\dot\alpha i}^A$), and a real $SU(2)_R$ triplet $D^{ijA}$ of auxiliary scalar fields. The supersymmetry transformations of the fields in (35) are summarized in appendix B. They satisfy the $\mathcal{N} = 2$ algebra (34) with central charge $Z = 0$ off shell, and modulo gauge transformations. The vector multiplet is characterized by the following supersymmetric constraints,

$$\overline{Q}_{\dot\alpha i} \phi^A = 0 \,, \qquad Q^{(i} Q^{j)} \phi^A = -\overline{Q}^{(i} \overline{Q}^{j)} \overline{\phi}^A \,. \tag{36}$$

The first constraint expresses the fact that the vector multiplet is chiral, while the second constraint expresses the reality of $D^{ijA}$, which is related to the Bianchi identity for the field strength $v_{\mu\nu}^A$ by supersymmetry.

In appendix B we also review the construction of the $\mathcal{N} = 2$ SYM lagrangian, which takes the following form when it is expressed in terms of the vector multiplet component fields (35),

$$\begin{aligned}
\mathcal{L} = \frac{1}{g^2} \bigg( & -\frac{1}{4} v_{\mu\nu}^A v^{A\mu\nu} + \frac{1}{4} D^{ijA} D_{ij}^A - D^\mu \overline{\phi}^A D_\mu \phi^A - i\overline{\lambda}_i^A \overline{\sigma}^\mu D_\mu \lambda^{iA} \\
& -\frac{1}{2} \left( i\varepsilon_{ABC} \overline{\phi}^B \phi^C \right)^2 + \frac{i}{\sqrt{2}} \varepsilon_{ABC} \overline{\phi}^A \lambda^{iB} \lambda_i^C + \frac{i}{\sqrt{2}} \varepsilon_{ABC} \phi^A \overline{\lambda}_i^B \overline{\lambda}^{iC} \bigg) \,.
\end{aligned} \tag{37}$$

Here the field strength $v_{\mu\nu}^A$ takes the same form as in (15), while the covariant derivatives are as in (19).

The auxiliary field $D^{ijA}$ in (37) can be integrated out by setting $D^{ijA} = 0$. The scalar potential is given by

$$V = \frac{1}{2} \left( i\varepsilon_{ABC} \overline{\phi}^B \phi^C \right)^2 \,. \tag{38}$$

This potential admits flat directions of the form $\phi^3 = 2a \in \mathbb{C}$. These are the classical precursors of the quantum vacua analyzed in [9] and reviewed in section 4.

For the purpose of this paper, the most important feature of the $\mathcal{N} = 2$ SYM lagrangian in (37) is that it reduces to the lagrangian (14) of $SU(2)$ adjoint QCD with $N_f = 2$ flavors (with $\theta = 0$),[24] once we delete all terms that involve the scalar fields $\phi^A$. Moreover, the two theories have the same global symmetries, once the symmetries of adjoint QCD discussed in section 2.2 are extended to the additional fields in the $\mathcal{N} = 2$ vector multiplet (35) in a suitable fashion:

---

[24]Since the $\theta$-term is topological, adding it to the lagrangian of $\mathcal{N} = 2$ SYM preserves supersymmetry. (Its supersymmetry variation is a total derivative.) Just as in adjoint QCD, we can set $\theta = 0$ using a $U(1)_r$ symmetry that suffers from an ABJ anomaly.

- As in section 2.2.1, there is an $SU(2)_R$ symmetry under which the gauginos $\lambda_\alpha^{iA}$ are doublets. The same is true of the $\mathcal{N}=2$ supercharge $Q_\alpha^i$. The auxiliary fields $D^{ijA}$ are triplets, while $v_\mu^A$ and $\phi^A$ are neutral.

- As in section 2.2.2, there is a classical $\mathsf{U}(1)_r$ symmetry under which the gauginos $\lambda_\alpha^{iA}$ have charge 1. The scalars $\phi^A$ have charge 2, $D^{ijA}$ and $v_\mu^A$ are neutral, and the supercharges $Q_\alpha^i$ have charge $-1$. The discussion around (24) applies without modification, so that $\mathsf{U}(1)_r$ is also broken to its $\mathsf{Z}_8$ subgroup by an ABJ anomaly. The $\mathsf{Z}_8$ generator r acts as follows,

$$\mathsf{r}\big(\phi^A\big) = i\phi^A, \qquad \mathsf{r}\big(\lambda_\alpha^{iA}\big) = e^{\frac{i\pi}{4}}\lambda_\alpha^{iA}, \qquad \mathsf{r}\big(Q_\alpha^i\big) = e^{-\frac{i\pi}{4}}Q_\alpha^i, \tag{39}$$

  while $v_\mu^A$ and $D^{ijA}$ are invariant. This action is consistent with the relation in (28).

- All fields transform in the adjoint representation of the $SU(2)$ gauge group. Consequently, the $\mathbb{Z}_2^{(1)}$ 1-form symmetry associated with the center of $SU(2)$ that was discussed in section 2.2.3 is also present in the $\mathcal{N}=2$ theory.

- The action of parity $\mathsf{P}$ and time-reversal $\mathsf{T}$ in (32) of section 2.2.4 can be extended to all fields in the $\mathcal{N}=2$ vector multiplet to give symmetries of (37),

$$\begin{aligned}
\mathsf{P}\big(v_\mu^A(x)\big) &= \mathscr{P}_\mu{}^\nu v_\nu^A(\mathscr{P}x), & \mathsf{T}\big(v_\mu^A(x)\big) &= \mathscr{T}_\mu{}^\nu v_\nu^A(\mathscr{T}x), \\
\mathsf{P}\big(\phi^A(x)\big) &= \overline{\phi}^A(\mathscr{P}x), & \mathsf{T}\big(\phi^A(x)\big) &= \phi^A(\mathscr{T}x), \\
\mathsf{P}\big(\lambda_\alpha^{iA}(x)\big) &= \overline{\lambda}^{\dot\alpha iA}(\mathscr{P}x), & \mathsf{T}\big(\lambda_\alpha^{iA}(x)\big) &= i\lambda_i^{\alpha A}(\mathscr{T}x), \\
\mathsf{P}\big(D^{ijA}(x)\big) &= D^{ijA}(\mathscr{P}x), & \mathsf{T}\big(D^{ijA}(x)\big) &= D_{ij}^A(\mathscr{T}x).
\end{aligned} \tag{40}$$

  By comparing with the supersymmetry transformations in appendix B, we can determine the action of $\mathsf{P}$ and $\mathsf{T}$ on the supercharges,

$$\mathsf{P}\big(Q_\alpha^i\big) = \overline{Q}^{\dot\alpha i}, \qquad \mathsf{T}\big(Q_\alpha^i\big) = -iQ_i^\alpha. \tag{41}$$

  Just as in (33), the transformations in (40) satisfy $\mathsf{P}^2 = \mathsf{T}^2 = 1$. Using (41), we can determine the transformation properties of the central charge $Z$ in the supersymmetry algebra (34),

$$\mathsf{P}(Z) = \overline{Z}, \qquad \mathsf{T}(Z) = -Z. \tag{42}$$

As was explained in the introduction, we will interpolate between $\mathcal{N}=2$ SYM and adjoint QCD by adding the following mass term for the scalars $\phi^A$,

$$\Delta V = \frac{M^2}{g^2}\phi^A\overline{\phi}^A. \tag{43}$$

We will study the deformed theory as a function of $M$ (which we take to be positive). The $\mathcal{N}=2$ theory corresponds to $M=0$, while two-flavor adjoint QCD is obtained in the limit $M \gg \Lambda$, where $\Lambda$ is the strong-coupling scale of the $\mathcal{N}=2$ theory.[25] Crucially, this mass deformation preserves all symmetries (and all 't Hooft anomalies, as we will see in section 2.4) as we interpolate between the two theories.

---

[25]The $\mathcal{N}=2$ theory is asymptotically free, just as adjoint QCD. Due to the extra complex scalars $\phi^A$ in the adjoint representation of $SU(2)$, the one-loop beta function is $\beta_{\mathcal{N}=2} = 4$. For our purposes, it will not be necessary to distinguish the strong coupling scales of the two theories, and we denote both of them by $\Lambda$.

## 2.4 Background fields and 't Hooft anomalies

In this section we describe the 't Hooft anomalies of adjoint QCD.[26] These anomalies must be reproduced in any phase of the theory [17] and hence constitute a powerful constraint on candidate IR scenarios. Since the $\mathcal{N} = 2$ theory enjoys the same symmetries as adjoint QCD and differs from it only by scalars, the two theories have the same anomalies. We will examine as many anomalies as possible, without claiming to be exhaustive. We will also discuss them via inflow (though in some cases, only schematically).

The allowed background fields and possible 't Hooft anomalies are dictated by the global symmetries of the theory. If we ignore the $Z_8$ flavor symmetry and the $\mathbb{Z}_2^{(1)}$ center symmetry, the symmetry class is CII (in condensed matter language), or $\text{Pin}^- \times_{\{\pm 1\}} SU_2$ in the notation of [70].

We will first summarize largely familiar anomalies that are visible on spin 4-manifolds. We then generalize to anomalies that are visible only on non-spin 4-manifolds. Here we benefit greatly from the fact that the global properties of the topologically twisted $\mathcal{N} = 2$ theory are very well understood.

### 2.4.1 Spin manifolds

We begin by assuming that spacetime $\mathcal{M}_4$ is an oriented manifold with a spin structure. When we discuss anomaly inflow we should therefore imagine that $\mathcal{M}_4$ is the boundary of a 5-manifold $\mathcal{M}_5$ with the same properties. We can then define conventional spinors and Dirac operators on both $\mathcal{M}_4$ and $\mathcal{M}_5$.

**The Witten anomaly for $SU(2)_R$**   The $SU(2)_R$ symmetry allows us to couple the fermions to background $SU(2)_R$ gauge fields $A_R^{(1)}$, with field strength $F_R^{(2)}$.[27] There are no 't Hooft anomalies under background gauge transformations that are continuously connected to the identity. However, there can be a $\mathbb{Z}_2$-valued 't Hooft anomaly (often referred to as the Witten anomaly) under large $SU(2)_R$ gauge transformations associated to $\pi_4(SU(2)) \cong \mathbb{Z}_2$ [71]. If the anomaly is present, a large gauge transformation $g$ modifies the partition function by a sign

$$Z\left[g\left(A_R^{(1)}\right)\right] = -Z\left[A_R^{(1)}\right]. \tag{44}$$

The partition function of the five-dimensional invertible spin-TQFT that captures the anomaly via inflow on a closed 5-manifold $\mathcal{M}_5$ is given by $(-1)^{\mathcal{I}[A_R^{(1)}]}$, where $\mathcal{I}[A_R^{(1)}]$ is the index of a certain real Dirac operator coupled to a fermion in the doublet representation of $SU(2)_R$ [71].

From [71] we recall that this anomaly is present if the theory under consideration contains an odd number of Weyl fermions in the doublet representation of $SU(2)_R$. This is indeed the case here: the gauginos $\lambda_\alpha^{iA}$ constitute three $SU(2)_R$ doublets. Note that even though this anomaly is discrete in nature (i.e. $\mathbb{Z}_2$-valued), it can only be matched by massless degrees of freedom [72]. We therefore conclude that the IR of adjoint QCD is necessarily gapless.

For future use, we also observe that the anomaly trivializes if we only consider the Cartan subgroup $U(1)_R \subset SU(2)_R$, because every $SU(2)_R$ doublet decomposes into two $U(1)_R$ representations of charge $\pm 1$. This is related to the fact that the $U(1)_R$ symmetry acts in a vector-like fashion, and can be preserved while giving mass to all fermions, while $SU(2)_R$ acts chirally. This will be important in section 2.5 below.

---

[26]See for instance [68] for a pedagogical introduction to 't Hooft anomalies.

[27]We use the same conventions for the $SU(2)_R$ background gauge field $A_R^{(1)}$ and its field strength $F_R^{(2)}$ as for the dynamical $SU(2)$ gauge field $v^{(1)}$ and its field strength $v^{(2)}$ (see (17)).

**Mixed anomalies involving $Z_8$ and $SU(2)_R$ or gravity**    Next we turn to 't Hooft anomalies associated with the $Z_8$ symmetry. To describe these it is convenient to embed $Z_8$ in the classically conserved $U(1)_r$ symmetry under which the gauginos $\lambda_\alpha^{iA}$ have charge 1. (See [73] for a related discussion.) We can therefore couple the classical theory to a $U(1)_r$ background gauge field $A_r^{(1)}$, with the understanding that $A_r^{(1)}$ will ultimately be restricted to a $Z_8$ gauge field.

The $U(1)_r$ symmetry has 't Hooft anomalies with itself, the $SU(2)_R$ flavor symmetry, and with background gravity. The corresponding anomaly coefficients are given by

$$\kappa_{r^3} = \operatorname{Tr} U(1)_r^{\ 3} = 6, \qquad \kappa_{rR^2} = \operatorname{Tr} U(1)_r SU(2)_R^2 = 3, \qquad \kappa_r = \operatorname{Tr} U(1)_r = 6. \tag{45}$$

The anomalies are summarized by the following anomaly inflow action for the background fields in five dimensions,

$$S_5[A_r^{(1)}, A_R^{(1)}] = i \int_{\mathcal{M}_5} \left( \frac{\kappa_{r^3}}{24\pi^2} A_r^{(1)} \wedge dA_r^{(1)} \wedge dA_r^{(1)} + \kappa_{rR^2} A_r^{(1)} \wedge c_2(R) - \frac{\kappa_r}{24} A_r^{(1)} \wedge p_1(\mathcal{M}_5) \right). \tag{46}$$

Here $c_2(R) = \frac{1}{8\pi^2} \operatorname{tr}\left(F_R^{(2)} \wedge F_R^{(2)}\right)$ is the second Chern class of the $SU(2)_R$ bundle and $p_1(\mathcal{M}_5)$ is the first Pontryagin class of the tangent bundle.

Let us first concentrate on the anomalies above that are linear in $A_r^{(1)}$ (i.e. the last two terms in (46)). If we perform a $U(1)_r$ background gauge transformation $A_r^{(1)} \to A_r^{(1)} + d\chi_r$, the action (46) captures the anomalous variation of the four-dimensional partition function. To see what remains when we restrict $A_r^{(1)}$ to a $Z_8$ gauge field we impose the constraint

$$\chi_r = \frac{2\pi}{8} k_r, \qquad k_r \in Z_8. \tag{47}$$

We then obtain the following anomalous variations of the four-dimensional partition function,

$$Z\left[A_r^{(1)} \to A_r^{(1)} + d\chi_r, A_R^{(1)}\right] = Z\left[A_r^{(1)}, A_R^{(1)}\right] \exp\left(\frac{i\pi k_r}{4}\left(\kappa_{rR^2} n_R - \frac{1}{8}\kappa_r \sigma\right)\right), \quad k_r \in Z_8. \tag{48}$$

Here $n_R = \int_{\mathcal{M}_4} c_2(R)$ is the $SU(2)_R$ instanton number and $\sigma = \frac{1}{3}\int p_1(\mathcal{M}_4)$ is the signature of $\mathcal{M}_4$. The instanton number is always an integer, $n_R \in \mathbb{Z}$. By contrast, $\sigma \in 16\mathbb{Z}$, since $\mathcal{M}_4$ is a spin manifold (see for instance [74]).

By choosing $k_r = 1$ (which corresponds to the $Z_8$ generator $r$) we deduce that:

- The mixed $Z_8$-$SU(2)_R$ anomaly is $\kappa_{rR^2} = 3 \pmod 8$.

- The mixed $Z_8$-gravity anomaly is $\kappa_r = 6 \pmod 4$.

Notice also that the $Z_4 \subset Z_8$ subgroup generated by $r^2$ (corresponding to $k_r = 2$ above) has no mixed 't Hooft anomaly with gravity when $\mathcal{M}_4$ is a spin manifold.

**The cubic $Z_8$ anomaly**    Although we will not use it in later sections, we briefly describe the discrete remnant of the anomaly in (46), which is cubic in the background field $A_r^{(1)}$. To characterize this term we note that abstractly a $Z_8$ background gauge field $z$ is a 1-cochain with values in $Z_8$. It is related to the $U(1)_r$ uplift via

$$A_r^{(1)} \quad \longrightarrow \quad \frac{2\pi}{8} z. \tag{49}$$

The quantity $k_r$ appearing above is then a gauge parameter for $z$.

To discuss the analog of the curvature $d\mathsf{A}_r^{(1)}$ we consider an integral uplift $\widetilde{\mathsf{z}}$ to a 1-chain with values in $\mathbb{Z}$. This cochain is not closed but, since $\mathsf{z}$ is closed, the coboundary $\delta\widetilde{\mathsf{z}}$ is a multiple of 8. We can use this to define an integral 2-cochain $\beta(\mathsf{z})$ as

$$\frac{\delta\widetilde{\mathsf{z}}}{8} = \beta(\mathsf{z}).\tag{50}$$

Note that although this definition appears to depend on the lift $\widetilde{\mathsf{z}}$ in fact the cohomology class of $\beta(\mathsf{z}) \in H^2(\mathcal{M}_4, \mathbb{Z})$ depends only on $\mathsf{z}$. More formally, the operation $\beta$ defined above is the Bockstein homomorphism associated to the exact sequence $\mathbb{Z} \to \mathbb{Z} \to \mathbb{Z}_8$.

When translating from the continuous $\mathsf{U}(1)_r$ to $\mathbb{Z}_8$, the Bockstein element $\beta(\mathsf{z})$ gives a discrete analog of the curvature

$$\frac{d\mathsf{A}_r^{(1)}}{2\pi} \quad \longrightarrow \quad \beta(\mathsf{z}).\tag{51}$$

In particular, the Bockstein element $\beta(\mathsf{z})$ is always a torsion element of the cohomology.

With the above ingredients, we can evaluate the cubic $\mathbb{Z}_8$ anomaly, which yields the following result for the five-dimensional partition function that captures the anomaly via inflow,

$$\exp\left(\frac{2\pi i}{8}\int_{\mathcal{M}_5} \mathsf{z} \cup \beta(\mathsf{z}) \cup \beta(\mathsf{z})\right).\tag{52}$$

Note that this is well defined under shifts of $\mathsf{z}$ by multiples of 8. Since this anomaly is only visible on manifolds with torsion in their cohomology we will not utilize it below.

**Mixed anomalies involving $\mathbb{Z}_8$ and the 1-form symmetry $\mathbb{Z}_2^{(1)}$**  Finally, we turn to anomalies that involve the 1-form symmetry $\mathbb{Z}_2^{(1)}$. This symmetry has no anomalies with itself: gauging it leads to the theory with $SO(3)$ gauge group (see below). However we will see that it has a mixed anomaly with the $\mathbb{Z}_8$ symmetry.

We first review how to couple the $\mathbb{Z}_2^{(1)}$ center symmetry to background fields. Additional details can be found in [2–5, 75]. The background field associated with $\mathbb{Z}_2^{(1)}$ is a 2-form $\mathbb{Z}_2$ gauge field $B_2$. It is a closed 2-cochain with coefficients in $\mathbb{Z}_2$,[28] which is subject to 1-form gauge transformations of the form $B_2 \to B_2 + \delta\Lambda_1$, where gauge parameter $\Lambda_1$ is a $\mathbb{Z}_2$ 1-cochain. Therefore $B_2$ defines a cohomology class in $H^2(\mathcal{M}_4, \mathbb{Z}_2)$.

In the presence of $B_2$, the functional integral over the dynamical $SU(2)$ gauge fields is modified: instead of summing over $SU(2)$ bundles on $\mathcal{M}_4$, we sum over $SO(3)$ bundles with fixed 't Hooft flux $w_2(SO(3)) = B_2$, where $w_2(SO(3))$ is the second Stiefel-Whitney class. It is known that $\mathbb{Z}_2^{(1)}$ does not have 't Hooft anomalies that prevent us from gauging it. It can therefore be gauged by summing over all $B^{(2)} \in H^2(\mathcal{M}_4, \mathbb{Z}_2)$. In the resulting theory we sum over all $SO(3)$ gauge bundles, without a constraint on $w_2(SO(3))$. Therefore gauging $B^{(2)}$ has the effect of converting the $SU(2)$ theory to a theory with gauge group $SO(3)$.

This perspective is useful when discussing mixed anomalies involving the $\mathbb{Z}_2^{(1)}$ symmetry: Any symmetry that is present if the gauge group is $SU(2)$, but absent if we replace the gauge group by $SO(3)$ necessarily has a mixed anomaly with $\mathbb{Z}_2^{(1)}$ in the original $SU(2)$ gauge theory. Precisely this happens for the $\mathbb{Z}_8$ flavor symmetry.

Indeed, as discussed in section 2.2.2, this symmetry is the remnant of a classical $\mathsf{U}(1)_r$ symmetry, which is broken to $\mathbb{Z}_8$ by an ABJ anomaly. In $SU(2)$ gauge theory, this breaking pattern follows from the fact that the $\theta$-angle was $2\pi$ periodic, together with its $\mathsf{U}(1)_r$ transformation

---

[28] We do not use the notation $B^{(2)}$ because $B_2$ is not a differential 2-form.

rule (25). As was discussed around (20), replacing the gauge group by $SO(3)$ extends the periodicity of $\theta$. On spin manifolds this leads to $\theta \sim \theta + 4\pi$. Therefore only $U(1)_r$ rotations by integer multiples of $\frac{\pi}{2}$ preserve $\theta = 0$. In $SO(3)$ gauge theory the ABJ anomaly therefore breaks $U(1)_r \rightarrow Z_4$. It follows that the original $SU(2)$ gauge theory must have a mixed 't Hooft anomaly between $Z_8$ and the $\mathbb{Z}_2^{(1)}$ center symmetry, while the $Z_4 \subset Z_8$ subgroup does not have such an anomaly on spin manifolds.

We can make this more explicit in terms of background fields as follows. The effect of an $r$ transformation is to modify the action by a $\theta$-term for the dynamical gauge field with coefficient $2\pi$,

$$\delta S = 2\pi n_v = \frac{1}{4\pi} \int_{\mathcal{M}_4} \mathrm{tr}\left(v^{(2)} \wedge v^{(2)}\right). \tag{53}$$

However, the fractional part of the instanton number can be expressed in terms of the background field $B_2$. Specifically, let $\mathcal{P}(B_2 \cup B_2)$ denote the Pontryagin square of $B_2$. This is $\mathbb{Z}_4$ cohomology class which is even on spin manifolds. Then we have (see e.g. [21])

$$n_v = \frac{1}{4} \int_{\mathcal{M}_4} \mathcal{P}(B_2 \cup B_2) \quad (\mathrm{mod}\ \mathbb{Z}). \tag{54}$$

From the above we deduce that after a $Z_8$ gauge transformation with parameter $k_r$ the partition function is modified as follows,

$$Z[B_2] \quad \longrightarrow \quad Z[B_2] \exp\left(\frac{i\pi k_r}{2} \int_{\mathcal{M}_4} \mathcal{P}(B_2 \cup B_2)\right). \tag{55}$$

Since the Pontryagin square is even, this anomalous variation is $\pm 1$ depending on the parity of $k_r$. In particular this means that the $Z_4$ subgroup generated by $r^2$ does not have an anomaly with the $\mathbb{Z}_2^{(1)}$ symmetry on spin manifolds. (Below we will see that this conclusion is modified on non-spin manifolds.)

It is also straightforward to write a five-dimensional partition function that produces the anomalous variation (55) by inflow,

$$\exp\left(\frac{i\pi}{2} \int_{\mathcal{M}_5} z \cup \mathcal{P}(B_2 \cup B_2)\right), \tag{56}$$

where $z$ is the $Z_8$ gauge field discussed in (50).

### 2.4.2 Background fields on non-spin manifolds

In the previous subsection we considered mass-deformed deformed $\mathcal{N} = 2$ SYM theories (including adjoint QCD) on spin manifolds $\mathcal{M}_4$. On such manifolds spinors are well defined, and we were free to turn on additional background fields associated with the global $SU(2)_R$, $Z_8$, and $\mathbb{Z}_2^{(1)}$ symmetries. In fact, the deformed $\mathcal{N} = 2$ theories we are discussing can be placed on more general manifolds, and coupled to more general background fields. In particular, they can be placed on manifolds that are not spin. As we will discuss in more detail below, this is due to the fact that all fields in the theory satisfy a nonabelian analog of the spin-charge relation [18], which correlates their $SU(2)_R$ representation with their Lorentz spin. This correlation is expressed by the identification (21) between the central element $-\mathbb{1} \in SU(2)_R$ and fermion parity,

$$-\mathbb{1} = (-1)^F. \tag{57}$$

Explicitly, this means that all fields whose $SU(2)_R$ spin $j_R$ is half-integer are fermions, while all fields for which $j_R \in \mathbb{Z}$ are bosons. (More precisely, this relation holds for all gauge-invariant

local operators.) The ability to place the theory on non-spin manifolds allows us to probe more of its properties; in particular, more 't Hooft anomalies.

Let us describe the class of backgrounds compatible with (57). This relation means that the symmetry group in question is a quotient[29]

$$\frac{\text{Spin}(4) \times SU(2)_R}{\mathbb{Z}_2}. \tag{58}$$

Where in the above Spin(4) is the spacetime symmetry group. One class of allowed backgrounds are products of bundles for the groups in the numerator. These are spin manifolds and $SU(2)_R$ gauge bundles. However, the most general class of backgrounds, where the quotient is relevant, can be expressed as a pair consisting of an $SO(4)$ bundle (describing the geometrty) and $SO(3)_R$ bundle (denoted by $\mathcal{B}_R$) subject to a constraint

$$w_2(\mathcal{M}_4) = w_2(\mathcal{B}_R). \tag{59}$$

The relation (59) can be thought of as the $SU(2)$ analogue of a spin$^c$ structure. For instance, (59) implies that the background gauge field associated with the Cartan subgroup $U(1)_R \subset SU(2)_R$ is a conventional spin$^c$ connection.

To see why such backgrounds are consistent consider for instance the fermions $\lambda_\alpha^{iA}$. If $\mathcal{M}_4$ is a spin manifold then $\lambda_\alpha^{iA}$ is a well-defined section of both the left-handed spinor bundle $\mathcal{S}_+$ and the $SU(2)_R$ bundle $\mathcal{B}_R$ acting via the fundamental representation. On non-spin manifolds the bundle $\mathcal{S}_+$ is not well-defined: the fact that $w_2(\mathcal{M}_4) \neq 0$ implies that it fails to be consistent on some triple overlaps of patches on $\mathcal{M}_4$. However whenever this happens there is similarly an obstruction to defining the action of $\mathcal{B}_R$ in the fundamental representation (i.e. defining a lift to $SU(2)_R$). As a result $\lambda_\alpha^{iA}$ can always be defined.

**Relationship to topologically twisted $\mathcal{N} = 2$ SYM theory** An important and well studied case of backgrounds satisfying (59) occurs in the supersymmetric theory in the context of topological twisting [20]. In that construction one takes the $SU(2)_R$ symmetry and the $\text{Spin}(4) = SU(2)_+ \times SU(2)_-$ Lorentz symmetry of the theory, and replaces the product $SU(2)_R \times SU(2)_+$ by its diagonal subgroup $\widetilde{SU(2)_+}$,

$$SU(2)_R \times SU(2)_+ \times SU(2)_- \quad \longrightarrow \quad \widetilde{SU(2)_+} \times SU(2)_-. \tag{60}$$

The symmetry $\widetilde{SU(2)_+} \times SU(2)_-$ is then interpreted as a twisted Lorentz group.

Twisting therefore modifies the Lorentz quantum numbers of fields that carry $SU(2)_R$ charge. Explicitly, a field that carries spins $(j_R, j_+, j_-)$ (with $j_{R,\pm} \in \frac{1}{2}\mathbb{Z}$) under $SU(2)_R \times SU(2)_+ \times SU(2)_-$ turns into a field that transforms in the $(\widetilde{j_+} = j_R \otimes j_+, j_-)$ representation of the twisted Lorentz group. The gauge field $v_\mu^A$ and the scalars $\phi^A$ are therefore not modified, while the fermions and the auxiliary field turn in to differential forms. Explicitly, $\lambda_\alpha^{iA}$ transforms as $\left(\frac{1}{2}, \frac{1}{2}, 0\right)$, and thus gives rise to a scalar and a self-dual 2-form in the $(\widetilde{0}, 0)$ and $(\widetilde{1}, 0)$ representations of the twisted Lorentz group. Similarly, $\overline{\lambda}_{\dot\alpha i}^A$ transforms as $\left(\frac{1}{2}, 0, \frac{1}{2}\right)$ and turns into a 1-form $(\widetilde{\frac{1}{2}}, \frac{1}{2})$ after twisting, while the auxiliary field $D^{ijA}$ (which transforms as $(1, 0, 0)$) gives rise to another self-dual 2-form $(\widetilde{1}, 0)$.

Since all fields have turned into differential forms, the twisted theory can be formulated on an arbitrary 4-manifold $\mathcal{M}_4$: A spin structure is not needed. The (suitably decorated) supersymmetric partition function $Z_{DW}$ of the twisted theory coincides with the Donaldson invariants (see for instance [21]) of the smooth 4-manifold $\mathcal{M}_4$. We refer to $Z_{DW}$ as the Donaldson-Witten partition function. Due to this relation, the global properties of twisted $\mathcal{N} = 2$ SYM

---

[29]The fact that the $\mathbb{Z}_8$ generator $r$ satisfies $r^4 = -\mathbb{1} = (-1)^F$ implies that this discussion can be further generalized to also include the $\mathbb{Z}_8$ background gauge field, but we will not do so here.

on 4-manifolds are quite well understood. This includes several subtle effects that will be important below.

Rather than thinking of the twisted $\mathcal{N} = 2$ theory in terms of fields that are differential forms on $\mathcal{M}_4$, we can equivalently describe it by coupling the physical, untwisted theory to a special configuration of the $SU(2)_R$ background gauge fields on $\mathcal{M}_4$.[30] In order to describe the twisted $\mathcal{N} = 2$ theory in this language, we decompose the spin connection into its self-dual part, which is valued in $SU(2)_+$, and its anti-self-dual part, which is valued in $SU(2)_-$. The topologically twisted theory is obtained by embedding the self-dual part of the curvature in the $SU(2)_R$ background gauge field $A_R^{(1)}$, i.e. the $SU(2)_R$ field strength $F_R^{(2)}$ is chosen so that

$$F_R^{(2)} = R_+^{(2)}, \tag{61}$$

where $R_+^{(2)}$ is the self-dual part of the Riemann curvature 2-form. This special choice of background fields ensures that there is a suitably covariantly constant spinor on $\mathcal{M}_4$, so that a conserved supercharge can be defined. When interpreted on a non-spin manifold. This relationship (61) between the curvature tensors implies the constraint (59).

### 2.4.3  't Hooft anomalies on non-spin manifolds

We now discuss refinements of the 't Hooft anomalies that are present on non-spin manifolds. Additionally we will discuss a mixed anomaly between the $\mathbb{Z}_2^{(1)}$ 1-form symmetry and geometry that does not have an analog on spin manifolds. Throughout we take $\mathcal{M}_4$ to be an oriented 4-manifold with $w_2(\mathcal{M}_4) \neq 0$.

**Anomalies involving $SU(2)_R$ and geometry**  We begin by discussing the 't Hooft anomalies associated with $SU(2)_R$ and the geometry of $\mathcal{M}_4$. As discussed in section 2.4.1, on spin manifolds the only such anomaly is the $\mathbb{Z}_2$-valued global anomaly of [71], which counts $SU(2)_R$ doublets modulo 2. In our case there are three such doublets, so the anomaly is present. We would like to know how to describe the anomaly when $\mathcal{M}_4$ is not spin, and also whether it becomes more refined on such manifolds. Both questions are answered in [70], where the authors classify candidate 't Hooft anomalies for fermionic theories with different symmetries. In their notation, we are discussing theories with symmetry $\mathrm{Pin}^- \times_{\{\pm 1\}} SU(2)_R$, or symmetry class CII. This means that we have a time-reversal symmetry that satisfies $\mathsf{T}^2 = 1$, as in (40), and an $SU(2)_R$ flavor symmetry whose $-\mathbb{1}$ element is identified with with $(-1)^F$ as in (57), so that all fields satisfy the nonabelian spin-charge relation discussed in section 2.4.2.

The analysis in [70] (see in particular corollary 9.95 on p.94) shows that candidate 't Hooft anomalies for four-dimensional theories (corresponding to $n = 5$ in table (9.96) of [70]) have a $\mathbb{Z}_2 \times \mathbb{Z}_2$ classification. Only one $\mathbb{Z}_2$ anomaly is present in free fermion theories, or theories that are continuously connected to such theories (as is the case for us, since our theories are asymptotically free).[31] This is precisely the global $SU(2)_R$ anomaly already discussed above, suitably generalized to non-spin manifolds. For instance, the partition function of the five-dimensional anomaly-inflow theory on a closed 5-manifold $\mathcal{M}_5$ now involves the mod 2 index of a certain $SU(2)_R$-twisted Dirac operator, which is also described in [70].

---

[30]The approach to supersymmetric field theories on curved manifolds based on supergravity background fields (such as the $SU(2)_R$ background field strength $F_R^{(2)}$) was pioneered in [76]. See [77] for a pedagogical introduction, and [78] for a detailed discussion of the relation between twisted and untwisted theories.

[31]The additional $\mathbb{Z}_2$ anomaly that does not arise in free fermion theories is related to a five-dimensional invertible TQFT with partition function $\exp\left(i\pi \int_{\mathcal{M}_5} w_2 \cup w_3\right)$, which was discussed in [79,80]. An example of a theory that carries this anomaly is a version of Maxwell theory in which all three fundamental line operators (with electric and magnetic charges $(1,0)$, $(0,1)$, and $(1,1)$) are fermions [80,81]. This theory can be engineered by coupling ordinary Maxwell theory (where the $(1,0)$ and $(0,1)$ lines are bosonic, while the $(1,1)$ line is a fermion) to $w_2(\mathcal{M}_4)$ via its electric and magnetic 2-form background gauge fields.

Thus we see that there are no new anomalies that involve $SU(2)_R$ and gravity. As was the case on spin manifolds, the anomaly trivializes if we only consider the $U(1)_R \subset SU(2)_R$ Cartan subgroup.

**Mixed anomalies involving $Z_8$**    We will now reexamine the mixed anomalies involving the $Z_8$ and $\mathbb{Z}_2^{(1)}$ symmetries, which were discussed on spin manifolds in section 2.4.1. We first reconsider the mixed anomalies between $Z_8$ and $SU(2)_R$ or gravity in (48). We substitute for the anomaly coefficients in (45), and we also write the $SU(2)_R$ instanton number as $n_R = \frac{1}{4} p_1(\mathcal{B}_R)$, where $p_1(\mathcal{B}_R)$ is the Pontryagin class, and the signature is expressed as $\sigma = \frac{1}{3} p_1(\mathcal{M}_4)$. Then the anomalous transformation rule (48) becomes

$$Z \longrightarrow Z \exp\left( \frac{i\pi \mathsf{k}_r}{16} \left( 3 p_1(\mathcal{B}_R) - p_1(\mathcal{M}_4) \right) \right), \quad \mathsf{k}_r \in Z_8. \tag{62}$$

Here it is understood that the bundles satisfy the constraint in (59). It is instructive to verify that (62) trivializes when $\mathsf{k}_r \in 8\mathbb{Z}$. This amounts to showing that

$$3 p_1(\mathcal{B}_R) - p_1(\mathcal{M}_4) = 0 \ (\mathrm{mod}\ 4). \tag{63}$$

Now we use the relations

$$\begin{aligned} p_1(\mathcal{B}_R) &= \mathcal{P}(w_2(\mathcal{B}_R) \cup w_2(\mathcal{B}_R)) \ (\mathrm{mod}\ 4), \\ p_1(\mathcal{M}_4) &= \mathcal{P}(w_2(\mathcal{M}_4) \cup w_2(\mathcal{M}_4)) + 2 w_4(\mathcal{M}_4) \ (\mathrm{mod}\ 4). \end{aligned} \tag{64}$$

Together with the identification in (59), the statement (63) reduces to

$$w_2(\mathcal{M}_4) \cup w_2(\mathcal{M}_4) + w_4(\mathcal{M}_4) = 0 \ (\mathrm{mod}\ 2), \tag{65}$$

which is indeed true on any oriented 4-manifold.[32]

The mixed anomaly between $Z_8$ and $\mathbb{Z}_2^{(1)}$ discussed around (66) is essentially the same:

$$Z[B_2] \to Z[B_2] \exp\left( \frac{i\pi \mathsf{k}_r}{2} \int_{\mathcal{M}_4} \mathcal{P}(B_2 \cup B_2) \right). \tag{66}$$

The only difference is that on a non-spin manifold the Pontryagin square of $B_2$ is not in general even. Therefore only the $Z_2 \subset Z_8$ subgroup generated by $r^4 = (-1)^F$ does not have a mixed 't Hooft anomaly with the $\mathbb{Z}_2^{(1)}$ center symmetry.

**A mixed anomaly involving the 1-form symmetry $\mathbb{Z}_2^{(1)}$ and geometry**    Finally, we will now discuss an 't Hooft anomaly that has no analogue on a spin manifold. To start the discussion, we briefly return to the twisted $\mathcal{N} = 2$ SYM theory on a 4-manifold discussed in section 2.4.2, whose partition function $Z_{DW}$ coincides with the Donaldson invariants. It is known that that the definition of $Z_{DW}$ depends on certain non-canonical choices. (See [21–25] for background and additional details.) The effect we will focus on here is absent for $SU(2)$ gauge bundles, but present for $SO(3)$ bundles. When phrased solely in terms of the $SU(2)$ theory, this means that the effect is only present when the 2-form background field $B_2$ that couples to the $\mathbb{Z}_2^{(1)}$ center symmetry is activated.

In this case, it is known that the Donaldson-Witten partition function requires a choice of spin$^c$ structure on $\mathcal{M}_4$. (Such a structure exists on every orientable 4-manifold.) A choice of

---

[32]On an oriented 4-manifold, the combination $w_2 \cup w_2 + w_4$ coincides with the top Wu class $\nu_4$, which always vanishes. This follows from the fact that the top Wu class is equal to the top Steenrod square operation on the degree zero cohomology class of 1.

spin$^c$ structure is equivalent to a choice of lift of $w_2$ to an integral cohomology class $\widetilde{w}_2$. (Such a lift always exists in four dimensions because $\beta(w_2)$ vanishes on any 4-manifold, see below.) If we change the integral lift by shifting $\widetilde{w}_2 \rightarrow \widetilde{w}_2 + 2y$ with $y$ an integral cohomology class, the Donaldson-Witten partition function shifts as follows,

$$Z_{DW} \quad \longrightarrow \quad Z_{DW}(-1)^{B_2 \cup y}. \tag{67}$$

Note that this expression is invariant under $B_2$ gauge transformations, because $y$ is a cohomology class. The transformation in (67) was derived in [24], by examining the path integral measure of the gaugino zero-modes on $\mathcal{M}_4$.

Alternatively, it is possible to define $Z_{DW}$ so that it does not depend on an integral lift of $w_2$, but rather an integral lift $\widetilde{B}_2$ of the $\mathbb{Z}_2^{(1)}$ background gauge field $B_2$. As before, changing lifts amounts to shifting $\widetilde{B}_2 \rightarrow \widetilde{B}_2 + 2x$, but now $x$ is an arbitrary integer 2-cochain. This leads to the following shift of the partition function,

$$Z_{DW} \quad \longrightarrow \quad Z_{DW}(-1)^{x \cup w_2(\mathcal{M}_4)}. \tag{68}$$

We will now interpret (67) and (68) as an 't Hooft anomaly of adjoint QCD. More precisely, it is a mixed anomaly that involves the $\mathbb{Z}_2^{(1)}$ center symmetry background field $B_2$ and the topology of $\mathcal{M}_4$. First, note that the two presentations of the anomaly are related by a local counterterm on $\mathcal{M}_4$, which depends the integral lifts of both $B_2$ and $w_2$,

$$S_{\text{c.t.}} = \frac{i\pi}{2} \int_{\mathcal{M}_4} \widetilde{B}_2 \cup \widetilde{w}_2(\mathcal{M}_4). \tag{69}$$

If we shift $\widetilde{w}_2(\mathcal{M}_4)$ by $2y$, the partition function is multiplied by $(-1)^{\widetilde{B}_2 \cup y} = (-1)^{B_2 \cup y}$, and shifting $\widetilde{B}_2$ by $2x$ multiplies the partition function by $(-1)^{x \cup \widetilde{w}_2(\mathcal{M}_4)} = (-1)^{x \cup w_2(\mathcal{M}_4)}$. Therefore the counterterm in (69) relates the two shifts in (67) and (68), but it cannot give rise to either one of them in isolation. This is typical of mixed 't Hooft anomalies.

We claim that the anomaly arises from the following five-dimensional action via inflow,[33]

$$S_5 = i\pi \int_{\mathcal{M}_5} B_2 \cup w_3(\mathcal{M}_5). \tag{70}$$

Note that this is invariant under 1-form gauge transformations of $B_2$,

$$B_2 \quad \rightarrow \quad B_2 + \delta\Lambda_1, \qquad S_5 \quad \rightarrow \quad S_5 = i\pi \int_{\mathcal{M}_4} \Lambda_1 \cup w_3 = 0, \tag{71}$$

because $w_3(\mathcal{M}_4)$ vanishes on any 4-manifold. We will interpret (70) by choosing lifts $\widetilde{B}_2$ and $\widetilde{w}_3(\mathcal{M}_5)$ to integral 2- and 3-cochains, respectively. Note that in five dimensions neither $\widetilde{B}_2$ nor $\widetilde{w}_3$ are, in general, closed. We also need a lift of $w_2$ to an integral 2-cochain $\widetilde{w}_2$. Since $B_2$ and $w_2(\mathcal{M}_4)$ are $\mathbb{Z}_2$ cohomology classes, it follows that $\delta\widetilde{B}_2 = \delta\widetilde{w}_2 = 0 \pmod{2}$. Therefore $\delta\widetilde{B}_2$ and $\delta\widetilde{w}_2$ are even integral 3-chains.

In particular, the discussion above implies that $\frac{1}{2}\delta\widetilde{w}_2$ is an integral 3-cochain which is closed. Moreover if the integral lift of $w_2$ is changed then $\frac{1}{2}\delta\widetilde{w}_2$ shifts by an exact element. Therefore $\frac{1}{2}\delta\widetilde{w}_2$ defines a well-defined cohomology class in $H^3(\mathbb{Z}, \mathcal{M}_5)$. This class is the image of $w_2$ under the Bockstein homomorphsim $\beta : H^2(\mathbb{Z}_2, \mathcal{M}_5) \rightarrow H^3(\mathbb{Z}, \mathcal{M}_5)$ associated with the

---

[33]In a different context, this anomaly was recently discussed in [75].

short exact coefficient sequence $\mathbb{Z} \to \mathbb{Z} \to \mathbb{Z}_2$. The cohomology class $\beta(w_2)$ is an uplift of $w_3$ to integer cohomology

$$\frac{1}{2}\delta\widetilde{w}_2 = \beta(w_2) = \widetilde{w}_3\,. \tag{72}$$

Using the identification of $\widetilde{w}_3$ above we now determine the anomalous variation implied by the action (70). We change the integral lifts as $\widetilde{w}_2 \to \widetilde{w}_2 + 2y$ and $\widetilde{B}_2 \to \widetilde{B}_2 + 2x$. Substituting into (70) and working modulo $2\pi i\mathbb{Z}$, we find that

$$\delta S_5 = i\pi \int_{\mathcal{M}_5} \widetilde{B}_2 \cup \delta y = i\pi \int_{\mathcal{M}_5} \delta\widetilde{B}_2 \cup y + i\pi \int_{\mathcal{M}_4} \widetilde{B}_2 \cup y\,. \tag{73}$$

The term $i\pi \int_{\mathcal{M}_5} \delta\widetilde{B}_2 \cup y$ vanishes modulo $2\pi i\mathbb{Z}$, because $\delta\widetilde{B}_2$ is an even integer 3-chain. Since $\widetilde{B}_2 = B_2 \pmod 2$, the last term gives rise to the anomalous shift in (67).

Note that $\beta(w_2) = \widetilde{w}_3$ is the obstruction to the existence of a spin$^c$ structure. The anomaly therefore trivializes if we are only allowed to consider spin$^c$ 5-manifolds $\mathcal{M}_5$. From the perspective of four-dimensional 't Hooft anomalies, this means that the anomaly (70) cannot be detected if we only activate spin$^c$ background gauge fields. Instead, we must allow the more general class of background fields discussed in section (2.4.2). This observation is important to resolve an apparent paradox once we add fermion masses in section 2.5. These preserve the $U(1)_R \subset SU(2)_R$ Cartan subgroup, but render all fermions massive, hence there should be no 't Hooft anomalies for this $U(1)_R$ subgroup.

## 2.5 Adding fermion masses

As discussed in section 1.1, a general constraint on possible phases for adjoint QCD is that after giving large masses to the fermions it should reproduce the expected behavior of theories with smaller $N_f$. In this section, we examine the possible fermion mass terms and the symmetries that they preserve.

The most general masses for the fermions $\lambda_\alpha^{iA}$ take the form

$$\Delta V = \frac{1}{2}m_{ij}\mathcal{O}^{ij} + \frac{1}{2}\overline{m}_{ij}\overline{\mathcal{O}}^{ij}\,, \qquad m_{ij} = m_{(ij)}\,, \qquad \left(m_{ij}\right)^* = \overline{m}^{ij}\,. \tag{74}$$

Here $\mathcal{O}^{ij}$ is the following fermion bilinear,

$$\mathcal{O}^{ij} = \mathcal{O}^{(ij)} = \lambda^{\alpha iA}\lambda_\alpha^{jA}\,. \tag{75}$$

This operator is also an order parameter for chiral symmetry breaking (see section 3). In $\mathcal{N} = 2$ SYM theory, supersymmetry relates $\mathcal{O}^{ij}$ to the chiral operator $u = \mathrm{tr}\left(\phi^2\right)$ that parametrizes the Coulomb branch. Using the supersymmetry transformations (B.11) in appendix B.1, we find that

$$-\frac{1}{2}Q^{\alpha i}Q_\alpha^j u = \lambda^{\alpha iA}\lambda_\alpha^{jA} - i\sqrt{2}\phi^A D^{ijA} = \mathcal{O}^{ij} - i\sqrt{2}\phi^A D^{ijA}\,. \tag{76}$$

If we are only interested in correlation functions or expectation values of $\mathcal{O}^{ij}$, we can set the $D$-term in (76) to zero using its equation of motion. This can be used to reliably track the operator $\mathcal{O}^{ij}$ to the deep IR (see section 5.3). If we deform the potential by $\frac{1}{2}m_{ij}Q^{\alpha i}Q_\alpha^j u + (\text{h.c.})$, we obtain the $\mathcal{O}(m)$ fermion mass in (74), as well as an $\mathcal{O}(m^2)$ scalar mass that arises by integrating out the $D$-term.

In order to analyze the mass deformation (74), it is helpful to diagonalize the complex symmetric mass matrix $m_{ij}$ using an $SU(2)_R$ rotation,[34]

$$m_{ij} = e^{i\chi} \begin{pmatrix} \alpha & 0 \\ 0 & \beta \end{pmatrix}, \qquad \alpha \geq 0, \quad \beta \geq 0. \tag{78}$$

The $\mathsf{Z}_8$ symmetry can then be used to restrict the range of the angle $\chi$ to a single quadrant, for instance $\chi \in \left[0, \frac{\pi}{2}\right)$, although it is not always convenient to do so. If $\alpha, \beta$ are sufficiently large, the fermions $\lambda_\alpha^{1A}$, $\lambda_\alpha^{2A}$ are very heavy and can be integrated out. In this way we can flow from adjoint QCD with $N_f = 2$ flavors to the theories with $N_f = 0, 1$.

The anomalous $\mathsf{U}(1)_r$ transformation in (25) can be used to set the phase $\chi$ in (78) to zero, at the expense of introducing a $\theta$-angle $\theta = 4\chi$ for the dynamical $SU(2)$ gauge field. When $\theta = 0$, the measure that appears in the euclidean functional integral is real and positive, as in [48,82]. This happens when $\chi$ is an integer multiple of $\frac{\pi}{4}$, so that $m_{11}$ and $m_{22}$ in (78) are real or purely imaginary. Symmetries that remain unbroken in the presence of such mass terms should therefore be free of 't Hooft anomalies.[35]

We would like to make some additional comments that will be useful in later sections:

1.) For generic choices of $\alpha$ and $\beta$, the $SU(2)_R$ and $\mathsf{Z}_8$ symmetries are explicitly broken. The $\mathsf{Z}_8$ generator $\mathsf{r}$ and time-reversal $\mathsf{T}$ act on the gaugino via $\left(\lambda_\alpha^{iA}\right) = e^{\frac{i\pi}{4}} \lambda_\alpha^{iA}$ (see (27)) and $\mathsf{T}\left(\lambda_\alpha^{iA}\right) = i\lambda_i^{\alpha A}$ (see (32)). The fermion bilinear $\mathcal{O}^{ij}$ in (75) then transforms as follows,

$$\mathsf{r}\left(\mathcal{O}^{ij}\right) = i\mathcal{O}^{ij}, \qquad \mathsf{T}\left(\mathcal{O}^{ij}\right) = -\mathcal{O}_{ij}. \tag{79}$$

For special choices of the phase $\chi$, it is therefore possible to combine $\mathsf{T}$ with a power of $\mathsf{r}$ to define a preserved time-reversal symmetry,

- When $\chi = 0, \pi$ the masses $m_{11}, m_{22}$ are real and preserve $\widetilde{\mathsf{T}} = \mathsf{r}^2\mathsf{T}$. Since these values of $\chi$ correspond to $\theta = 0$, it follows from the discussion above that $\widetilde{\mathsf{T}}$ should be free of 't Hooft anomalies.[36]

- When $\chi = \pm\frac{\pi}{2}$, the masses $m_{11}, m_{22}$ are purely imaginary and preserve $\mathsf{T}$, which should therefore be free of 't Hooft anomalies.

- When $\chi = \frac{\pi}{4}$, the complex masses $m_{11} = e^{\frac{i\pi}{4}}\alpha$ and $m_{22} = e^{\frac{i\pi}{4}}\beta$ can be traded for an $SU(2)$ $\theta$-angle with $\theta = \pi$. This choice of masses is preserved by $\mathsf{r}\mathsf{T}$, but since $\theta = \pi$, this symmetry may have an 't Hooft anomaly. Indeed, it follows from (55) that the partition function transforms as

$$\mathsf{r}\mathsf{T}(Z) = Z e^{\frac{i\pi}{2}\mathcal{P}(B_2 \cup B_2)}. \tag{80}$$

This matches the mixed 't Hooft anomaly between time-reversal and the $\mathbb{Z}_2^{(1)}$ center symmetry of pure $SU(2)$ YM theory at $\theta = \pi$ uncovered in [4]. As was explained

---

[34]In triplet notation (see appendix A), this amounts to

$$m_1 = \frac{i}{2} e^{i\chi}(\alpha - \beta), \qquad m_2 = -\frac{1}{2} e^{i\chi}(\alpha + \beta), \qquad m_3 = 0. \tag{77}$$

[35]With the exception of Weyl anomalies, 't Hooft anomalies manifest as phases in euclidean signature.

[36] This is not manifest, because $\mathsf{r}^2$ has a variety of 't Hooft anomalies, while $\mathsf{T}$ does not (see below). However, it is possible to adjust the local counterterms so that $\widetilde{\mathsf{T}} = \mathsf{r}^2\mathsf{T}$ is free of 't Hooft anomalies. For instance, recall from (55) that there is a mixed anomaly between $\mathsf{Z}_8$ and $\mathbb{Z}_2^{(1)}$, which leads to the transformation rule $\mathsf{r}^2(Z) = Z e^{i\pi B_2 \cup B_2}$ for the partition function $Z$. (Here $B_2$ is the $\mathbb{Z}_2^{(1)}$ background gauge field.) This transformation can be absorbed by the local counterterm $e^{\frac{i\pi}{2}\mathcal{P}(B_2 \cup B_2)}$, which is well defined because $\mathcal{P}(B_2 \cup B_2) \in \mathbb{Z}_4$. This counterterm transforms by the right amount under $\mathsf{T}$ to render $\widetilde{\mathsf{T}} = \mathsf{r}^2\mathsf{T}$ anomaly free. Similar comments apply to mixed anomalies with $SU(2)_R$ and gravity.

there (and unlike the discussion in footnote 36), it is not possible to remove this anomaly using well-defined local counterterms.[37]

2.) When $m_{22} = \beta = 0$, the fermion $\lambda_\alpha^{2A}$ remains massless, while $\lambda_\alpha^{1A}$ acquires a complex mass $m_{11} = e^{i\chi}\alpha$. For large $\alpha$, the theory flows to adjoint QCD with $N_f = 1$, i.e. pure $\mathcal{N} = 1$ SYM theory. In this case the fermion masses break $SU(2)_R$ and $\mathsf{Z}_8$ to the expected $\mathsf{Z}_4$ symmetry of the $\mathcal{N} = 1$ SYM theory (see section 1.1). Moreover, there is a preserved time-reversal symmetry for any value of $m_{11}$. Note that the operator $m_{11}Q^{\alpha 1}Q_\alpha^1 u + (\text{h.c.})$, which preserves the supercharge $Q_\alpha = Q_\alpha^1$ and its hermitian conjugate (see appendix B), is precisely the $\mathcal{N} = 1$ preserving superpotential deformation studied in [9].

3.) If $\alpha = \beta$, the $U(1)_R \subset SU(2)_R$ Cartan subgroup is preserved. While r remains broken, we can combine $\mathsf{r}^2$ with an $SU(2)_R$ rotation $\mathcal{U} = \left(\begin{smallmatrix} 0 & i \\ i & 0 \end{smallmatrix}\right)$ by $\pi$ around the 1-axis to define a preserved $\widetilde{\mathbb{Z}}_2$ symmetry generated by $\mathsf{r}^2\mathcal{U}$. In order to verify that $\mathsf{r}^2\mathcal{U}$ has order two, note that $\mathcal{U}^2 = -\mathbb{1}$ is a $2\pi$ rotation in $SU(2)_R$. Together with (28), we find

$$\left(\mathsf{r}^2\mathcal{U}\right)^2 = \mathsf{r}^4\mathcal{U}^2 = (-1)^F(-1)^F = 1. \tag{81}$$

Since $\mathsf{r}^2\mathcal{U}$ acts as reflection across the 2-3 plane and $U(1)_R$ describes rotations around the 3-axis, the two symmetries do not commute. Rather, they combine into

$$O(2)_R = \widetilde{\mathbb{Z}}_2 \ltimes U(1)_R. \tag{82}$$

This symmetry is unbroken when $m_{11}, m_{22}$ are real or purely imaginary (corresponding to $\theta = 0$), and hence it should be free of 't Hooft anomalies.[38]

In section 2.4.2 we explained how to place the theories under consideration on non-spin manifolds using $SU(2)_R$ background gauge fields that satisfy the spin$^c$-like condition (59). Once we add fermion masses that break the $SU(2)_R$ symmetry this is no longer possible. An exception occurs when $\alpha = \beta$ (see point 3.) above), since this preserves the $U(1)_R \subset SU(2)_R$ Cartan subgroup. The $U(1)_R$ background gauge field is a conventional spin$^c$ connection, which can be used to place the theory on non-spin manifolds. This necessarily involves a choice of spin$^c$ structure on $\mathcal{M}_4$, because the conventionally normalized $U(1)_R$ field strength $G_R^{(2)}$ defines an integral lift $\frac{2}{2\pi}G_R^{(2)}$ of $w_2(\mathcal{M}_4)$.

As was already pointed out in section 2.4.3, the 't Hooft anomaly associated with the five-dimensional anomaly inflow action $\exp\left(i\pi\int_{\mathcal{M}_5} B_2 \cup w_3(\mathcal{M}_5)\right)$ trivializes when we restrict to $U(1)_R$ spin$^c$ background fields, because $w_3(\mathcal{M}_5)$ vanishes on 5-manifolds with a spin$^c$ structure. More prosaically, this also follows from an extension of the discussion around (82): both the $U(1)_R$ symmetry (which is needed to access 4-manifolds with non-vanishing $w_2(\mathcal{M}_4)$) and the $\mathbb{Z}_2^{(1)}$ center symmetry are compatible with real (or purely imaginary) fermion masses. Hence there cannot be a mixed 't Hooft anomaly involving these symmetries.

---

[37]By contrast, rT does not have mixed 't Hooft anomalies with the $SU(2)_R$ symmetry or gravity. As in footnote 36, they can be eliminated by adding suitable local counterterms.

[38]This is easy to see for $U(1)_R$, because the doublet gauginos $\lambda_\alpha^i$ decompose into fields of $U(1)_R$ charge $\pm 1$. On spin manifolds $\mathsf{r}^2$ does not have a mixed 't Hooft anomaly with gravity (see the discussion around (55)), and hence the same is true for $\mathsf{r}^2\mathcal{U}$. Showing that $\mathsf{r}^2\mathcal{U}$ does not have mixed 't Hooft anomalies with $SU(2)_R$ is complicated by the fact that the $SU(2)_R$ symmetry has a Witten anomaly (see section 2.4.1).

# 3 A $\mathbb{CP}^1$ phase with confinement and chiral symmetry breaking

As was discussed around (1), the most familiar scenario for the IR behavior of adjoint QCD is that it confines and spontaneously breaks the $SU(2)_R$ chiral symmetry. This is expected to happen via the condensation of the following fermion bilinear,

$$\mathcal{O}^{ij} = \mathcal{O}^{(ij)} = \lambda^{\alpha i A}\lambda_\alpha^{jA} \quad \Longleftrightarrow \quad \mathcal{O}^I = \frac{i}{2}\tau_{ij}^I \lambda^{\alpha i A}\lambda_\alpha^{jA}. \tag{83}$$

In this section we explore some consequences of this assumption, without referring to a more complete microscopic description. (Such a description will be discussed in section 5.3.) As we will see, the long-distance physics is described by two copies of a $\mathbb{CP}^1$ sigma model.

## 3.1 Broken and unbroken symmetries

We begin by discussing the symmetries that are preserved and broken by a condensate for the operator $\mathcal{O}^{ij}$ in (83). (Since this operator also appears in the mass deformation (74), there is some overlap with the discussion in section 2.5.) The operator $\mathcal{O}^{ij}$ is a complex triplet of $SU(2)_R$. Equivalently, $\mathcal{O}^I$ is a complex $SO(3)_R$ vector. If $\text{Re}\,\mathcal{O}^I$ and $\text{Im}\,\mathcal{O}^I$ were to acquire generic expectation values, the $SU(2)_R$ symmetry would be completely broken. This possibility is ruled out by the general arguments of [82], which imply that the $U(1)_R \subset SU(2)_R$ Cartan subgroup is necessarily unbroken. To see this, note that the fermions $\lambda_\alpha^{iA}$ can be assembled into a single Dirac spinor $\Psi = \left(\lambda_\alpha^{1A}, \overline{\lambda}_{\dot\alpha}^{1A}\right)$, which has charge 1 under $U(1)_R$. We can therefore add a Dirac mass $m_\Psi$ for $\Psi$ that preserves the $U(1)_R$ symmetry.[39] The arguments of [82] then imply that $U(1)_R$ is not spontaneously broken if we take $m_\Psi \to 0$. Note that these arguments apply to adjoint QCD, but not in general to the (deformed or undeformed) $\mathcal{N} = 2$ SYM theory, due to the Yukawa couplings in (37) (see e.g. [36] for a discussion of this point).

The upshot is that $\text{Re}\,\mathcal{O}^I$ and $\text{Im}\,\mathcal{O}^I$ must have parallel vevs, which spontaneously break $SU(2)_R$ to its $U(1)_R$ Cartan subgroup. This leads to two NG bosons (which we will sometimes refer to as pions) that parametrize the coset space $SU(2)/U(1) = \mathbb{CP}^1$ via a real $SO(3)_R$ unit vector $n^I$. At low energies, the operator $\mathcal{O}^I$ therefore flows to

$$\mathcal{O}^I \to C\,n^I, \qquad \langle n^I \rangle = \delta^{I3}, \qquad n^I n^I = 1, \qquad C \in \mathbb{C}. \tag{84}$$

Here we have oriented the vev of $n^I$ along the positive 3-direction. At the end of section 3.4 we will show that the constant $C$ in (84), which determines the vev of $\mathcal{O}^I$, must in fact be real or purely imaginary,

$$C \in \mathbb{R}, \quad \text{or} \quad C \in i\mathbb{R}. \tag{85}$$

Let us consider the fate of the other global symmetries of adjoint QCD that were discussed in section 2.2:

- The generator $\mathsf{r}$ of the $\mathbb{Z}_8$ symmetry acts as (see (27))

$$\mathsf{r}\left(\lambda_\alpha^{iA}\right) = e^{\frac{i\pi}{4}}, \qquad \mathsf{r}\left(\mathcal{O}^I\right) = i\mathcal{O}^I. \tag{86}$$

Therefore $\mathsf{r}$ is always spontaneously broken. Moreover, it necessarily sends the $\mathbb{CP}^1$ we started with (denoted by $\mathbb{CP}^1_+$), characterized by a constant $C = C_+$ in (84), to a second, disconnected $\mathbb{CP}^1$ (denoted by $\mathbb{CP}^1_-$), characterized by $C = C_- = iC_+$. Together with (85) this implies that we are free to choose

$$C_+ = f_\pi^3, \qquad C_- = iC_+ = if_\pi^3, \qquad f_\pi > 0. \tag{87}$$

---

[39] In the language of section 2.5 this amounts to the statement that the $U(1)_R$ symmetry is preserved by the addition of fermion masses (74) with real or purely imaginary $m_{11}, m_{22}$. As was noted there, this symmetry is free of 't Hooft anomalies.

Here $f_\pi$ is the pion decay constant, which has units of energy.

The symmetry $\mathsf{r}^2$ acts as

$$\mathsf{r}^2(\mathcal{O}^I) = -\mathcal{O}^I\,, \qquad \mathsf{r}^2(n^I) = -n^I\,. \tag{88}$$

It is therefore spontaneously broken, but maps $\mathbb{CP}^1_+$ to itself. (Similarly, $\mathsf{r}^2$ also maps $\mathbb{CP}^1_-$ to itself.) We can combine $\mathsf{r}^2$ with a broken $SU(2)_R$ rotation $\mathcal{U}$ by $\pi$ around the 1-axis,[40] so that

$$\mathsf{r}^2\mathcal{U}(n_1) = -n_1\,, \qquad \mathsf{r}^2\mathcal{U}(n_{2,3}) = n_{2,3}\,. \tag{89}$$

Thus $\mathsf{r}^2\mathcal{U}$ generates an unbroken $\widetilde{\mathbb{Z}}_2$ symmetry, which acts as a reflection across the 2-3 plane. As such it does not commute with the $U(1)_R$ symmetry, which acts by rotations around the 3-axis. Together, the two symmetries assemble into the group $O(2)_R = \widetilde{\mathbb{Z}}_2 \ltimes U(1)_R$.

The preceding discussion shows that the symmetry-breaking pattern induced by the vev (84) for the fermion bilinear $\mathcal{O}^I$ is given by

$$\frac{\mathsf{Z}_8 \times SU(2)_R}{\mathbb{Z}_2} \qquad \longrightarrow \qquad \widetilde{\mathbb{Z}}_2 \ltimes U(1)_R = O(2)_R\,. \tag{90}$$

As discussed above, this implies that there are two disconnected, physically equivalent copies $\mathbb{CP}^1_\pm$ of the $\mathbb{CP}^1$ model, which are exchanged by the spontaneously broken $\mathsf{Z}_8$ generator $\mathsf{r}$. For this reason we will mostly focus on $\mathbb{CP}^1_+$.

- The assumption of confinement implies that the $\mathbb{Z}_2^{(1)}$ 1-form symmetry is unbroken. In section 3.2 we will discuss its embedding into the symmetries of the $\mathbb{CP}^1$ model.

- Time-reversal $\mathsf{T}$ acts on the fermions via $\mathsf{T}(\lambda_\alpha^{iA}) = i\lambda_i^{\alpha A}$ (see (32)), so that the fermion bilinear in (83) transforms as follows,

$$\mathsf{T}(\mathcal{O}^I) = -\mathcal{O}^I\,. \tag{91}$$

Together with (87), this implies that $\mathsf{T}$ is preserved on $\mathbb{CP}^1_-$, where $\langle\mathcal{O}^I\rangle = C_- = if_\pi^3$ is purely imaginary, while $\widetilde{\mathsf{T}} = \mathsf{r}^2\mathsf{T}$ is preserved on $\mathbb{CP}^1_+$, where $\langle\mathcal{O}^I\rangle = C_+ = f_\pi^3$ is real. Note that $\mathsf{T}$ and $\widetilde{\mathsf{T}}$ are related via conjugation by the broken $\mathsf{Z}_8$ generator $\mathsf{r}$,

$$\mathsf{r}\,\mathsf{T}\,\mathsf{r}^{-1} = \mathsf{r}^2\,\mathsf{T} = \widetilde{\mathsf{T}}\,. \tag{92}$$

## 3.2 The $\mathbb{CP}^1$ model

As discussed above, chiral symmetry breaking via (83) leads to two copies of a $\mathbb{CP}^1$ sigma model. We will now describe some aspects of this model in more detail.

### 3.2.1 Discrete $\theta$-angle and Hopf solitons

The low-energy pions are described by a real $SO(3)_R$ unit vector $n_I$ (see (84)). Even though the $SU(2)_R$ symmetry is spontaneously broken, we must match the associated Witten anomaly

---

[40]We can take $\mathcal{U}$ to be a rotation by $\pi$ around any axis in the 1-2 plane. Different choices of $\mathcal{U}$ are related by the unbroken $U(1)_R$ symmetry.

discussed in section 2.4.1.[41] As in [53,54] this is achieved by including in the definition of the $\mathbb{CP}^1$ model a discrete $\theta$-angle associated with

$$\pi_4(\mathbb{CP}^1) = \mathbb{Z}_2 \,. \tag{93}$$

More explicitly, the model is defined by a functional integral over maps $n^I$ from euclidean spacetime $S^4$ (the compactification from $\mathbb{R}^4$ to $S^4$ arises because the pions must approach the vacuum at infinity) to the target $\mathbb{CP}^1$. Due to (93) this function space has two disconnected components. We are free to weight the contributions from the topologically non-trivial sector by a minus sign. This is accomplished by the discrete $\theta$-term for the $\mathbb{CP}^1$ model. (In section 3.2.3 below, we will also discuss conventional continuous $\theta$-terms.)

As in [53], adding the discrete $\theta$-term affects the quantum numbers of solitons, which makes it possible to identify them with the baryons of the underlying microscopic model. In adjoint QCD, the analogue of a baryonic operator takes the following schematic form,

$$\mathrm{tr}\left(\lambda_\alpha^i \cdots v_{\mu\nu} \cdots D_\mu \lambda^j \cdots\right) \,. \tag{94}$$

These operators satisfy the nonabelian spin-charge relation (21), according to which the central element $-\mathbb{1} \in SU(2)_R$ is identified with fermion parity $(-1)^F$. In other words, the operators in (94) are either bosons in integer-spin representations of $SU(2)_R$, or fermions in half-integer spin representations of $SU(2)_R$. In a confining phase we expect all excitations to arise by acting with such gauge-invariant local operators on the vacuum, and hence these excitations should also satisfy the spin-charge relation. Note that this is the case for the perturbative sector of the $\mathbb{CP}^1$ model, described by small fluctuations of the pion field $n^I$, which is a boson in the vector representation of $SO(3)_R$.[42]

The solitons of the $\mathbb{CP}^1$ model are characterized by a topological charge given by the Hopf invariant (for this reason we refer to them as Hopf solitons),

$$\pi_3(\mathbb{CP}^1) = \mathbb{Z} \,. \tag{95}$$

As discussed above, matching the $SU(2)_R$ Witten anomaly requires a discrete $\theta$-angle. This has the effect of turning the minimal Hopf soliton of topological charge 1 into a fermion, while maintaining consistency with the generalized spin-charge relation (21).[43] There are three related aspects to this claim:

- If we consider a sufficiently simple spatial manifold, such as $S^3$, the Hopf invariant (95) appears to define a meaningful integer charge $Q_H$ and suggests the existence of a (perhaps accidental) $U(1)_H$ symmetry of the $\mathbb{CP}^1$ model. However, it was shown [56] in the closely related context of the three-dimensional $\mathbb{CP}^1$ model, that the Hopf charge $Q_H$ is

---

[41]By contrast, the mixed 't Hooft anomalies that involve the spontaneously broken discrete symmetries r and $r^2$ with $SU(2)_R$, $\mathbb{Z}_2^{(1)}$ or gravity are matched because the vacua that are related by these broken symmetries have different counterterms for the corresponding background fields. (This leads to non-trivial 't Hooft anomalies on domain walls interpolating between such vacua.) Recall from the discussion in section (2.5) that the unbroken symmetries T or $\widetilde{T} = r^2 T$ (depending on the $\mathbb{CP}^1$) and $r^2\mathcal{U}$ are free of 't Hooft anomalies.

[42]More precisely, we should use the unbroken $U(1)_R$ charge to label dynamical excitations, in terms of which the nonabelian spin-charge relation (21) reduces to the conventional spin-charge relation for an abelian symmetry. By contrast, local operators always transform in full $SU(2)_R$ representations and thus continue to satisfy (21) in the spontaneously broken phase.

[43]Aspects of this problem were discussed in [55]. For the $N_f = N_c = 2$ adjoint QCD theory studied here, we do not encounter the exotic soliton states discussed there. Note that if we study adjoint QCD with $N_f = 2$ and general $N_c$, there is an $SU(2)_R$ Witten anomaly if and only if $N_c$ is even. Therefore the $\mathbb{CP}^1$ model only has a discrete $\theta$-angle for even $N_c$. Since this coupling is also responsible for turning the Hopf solitons of the $\mathbb{CP}^1$ model into fermions, a possible confining and chiral-symmetry breaking phase for adjoint QCD with odd $N_c$ may be qualitatively different than the scenario discussed here.

at best meaningful modulo 2. The basic issue is that, unlike conventional topological charges, $Q_H$ cannot be expressed as the integral of a well-defined local density. In other words, there is no good notion of a conserved current for a putative $U(1)_H$ symmetry.[44] As a result, the charge $Q_H$ cannot be consistently defined if space is chosen to be a more complicated 3-manifold $\mathcal{M}_3$. However, it was shown in [56] that a $\mathbb{Z}_2$-valued charge can be defined on any 3-manifold $\mathcal{M}_3$ with a $\mathrm{spin}^c$ structure, and that this charge reduces to $Q_H$ (mod 2) when $\mathcal{M}_3 = S^3$.

- The presence of the discrete $\theta$-angle implies that the topological charge $Q_H$ (mod 2) coincides with $(-1)^F$. To see that the elementary Hopf soliton with $Q_H = 1$ is a fermion, one can follow [53] and consider a process that involves the nucleation and annihilation of a soliton-antisoliton pair that also involve a spatial rotation by $2\pi$. This process is described by a map from spacetime $S^4$ to the $\mathbb{CP}^1$ target space that is a non-trivial element of $\pi_4(\mathbb{CP}^1)$. Due to the discrete $\theta$-angle, it is weighted by a minus sign in the path integral, which implies that the soliton is a fermion. Alternatively, we can study the disorder operators of the $\mathbb{CP}^1$ model (as was done in [83]), which are defined by removing a small euclidean ball from spacetime and demanding that the configuration of the pions on the $S^3$ boundary of this ball has non-trivial Hopf number. (Alternatively, we can study these operators by considering states on $S^3$.) The discrete $\theta$-angle then implies that the operators with odd Hopf number are fermions. They furnish the IR description of the baryon operators (94).

- In order to see why the Hopf solitons, and the corresponding disorder operators, satisfy the generalized spin-charge relation $(-1)^F = -\mathbb{1} \in SU(2)_R$, we can follow [84] and explicitly examine the simplest Hopf map from $S^3$ to $\mathbb{CP}^1$, where we think of $S^3$ as space. In suitable conventions, this map identifies the $-\mathbb{1}$ element of the left $SU(2)$ isometry of the spatial $S^3$ with the $-\mathbb{1}$ element of the $SU(2)_R$ isometry of the $\mathbb{CP}^1$ target space. Since the former is a spatial rotation by $2\pi$, it coincides with $(-1)^F$.

### 3.2.2 Unbroken 1-form symmetry and confining strings

As was explained in [3], the $\mathbb{CP}^1$ sigma model has a $U(1)^{(1)}$ 1-form symmetry. The corresponding 2-form current $J^{(2)}$ is pullback to spacetime (via the pion field $n^I$) of the $\mathbb{CP}^1$ Kähler form $\omega$,

$$*J^{(2)} = n^*(\omega). \tag{96}$$

Since $\omega$ is closed, it follows that $J^{(2)}$ is conserved. We normalize $\omega$ such that the target space has unit volume, $\int_{\mathbb{CP}^1} \omega = 1$. It follows that $J^{(2)}$ associates integer charges to 2-cycles $\Sigma_2$ in spactime,

$$\int_{\Sigma_2} *J^{(2)} = \int_{\Sigma_2} n^*(\omega) \in \mathbb{Z}. \tag{97}$$

The $U(1)^{(1)}$ symmetry is unbroken, and there are solitonic strings, associated with $\pi_2(\mathbb{CP}^1) = \mathbb{Z}$, that are charged under $U(1)^{(1)}$. The charge in (97) measures the number of strings piercing the spatial 2-plane $\Sigma_2$. There are also disorder line operators charged under $U(1)^{(1)}$, which therefore create solitonic strings. These operators can be defined by excising a small euclidean tubular neighborhood (with boundary $\sim S^2 \times \mathbb{R}$) of the line and demanding that the pion fields on the $S^2$ have non-trivial winding number.

---

[44]In section 5.3 we will describe the $\mathbb{CP}^1$ model using a gauged linear sigma model that arises by deforming $\mathcal{N} = 2$ SQED. In this description, the $U(1)_H$ symmetry is conserved classically, but explicitly broken to $(-1)^F$ by an ABJ anomaly involving the $U(1)$ gauge field.

The $\mathbb{Z}_2^{(1)}$ center symmetry of UV theory is a subgroup of $U(1)^{(1)}$, which is therefore an accidental symmetry of the low-energy theory. Higher-energy process can violate the $U(1)^{(1)}$ symmetry, so that the 1-form charge of strings and line operators is only meaningful modulo 2. The remaining $\mathbb{Z}_2^{(1)}$, which is neither explicitly nor spontaneously broken, is consistent with the assumption of confinement. As in the examples discussed in [53], this means that the elementary solitonic string of the $\mathbb{CP}^1$ model can be interpreted as the confining string of adjoint QCD.

For future use, we present the coupling of $J^{(2)}$ to a $U(1)^{(1)}$ background 2-form gauge field $B^{(2)}$ (normalized so that $\int_{\Sigma_2} B^{(2)}$ is gauge invariant modulo $2\pi$),

$$\exp\left(i\int_{\mathcal{M}_4} B^{(2)} \wedge *J^{(2)}\right) = \exp\left(i\int_{\mathcal{M}_4} B^{(2)} \wedge n^*(\omega)\right). \tag{98}$$

We can then embed the $\mathbb{Z}_2^{(1)}$ background gauge field $B_2 \in H^2(\mathcal{M}_4, \mathbb{Z}_2)$ via $B^{(2)} = \pi B_2$, so that the coupling (98) takes the form

$$\exp\left(i\pi\int_{\mathcal{M}_4} B_2 \cup n^*(\omega)\right). \tag{99}$$

### 3.2.3 Possibility of a continuous $\theta$-angle

The $\mathbb{CP}^1$ sigma-model admits a topological coupling that is analogous to a conventional, continuous $\theta$-term. It modifies the functional integral by the following phase factor,

$$\exp\left(\frac{i\theta}{2}\int_{\mathcal{M}_4} n^*(\omega) \wedge n^*(\omega)\right). \tag{100}$$

Here the normalization follows from (97). Despite the factor of $\frac{1}{2}$ in (100), $\theta$ has standard periodicity $\theta \sim \theta + 2\pi$ if $\mathcal{M}_4$ is a spin manifold.[45] The necessary modification on non-spin manifolds is discussed in section 3.3.

As discussed around (91), both $\mathbb{CP}^1_\pm$ sigma models that arise in the context of adjoint QCD preserve a time-reversal symmetry. ($\mathbb{CP}^1_+$ preserves $\widetilde{\mathsf{T}} = r^2\mathsf{T}$, while $\mathbb{CP}^1_-$ preserves $\mathsf{T}$.) It follows that the $\theta$-angle in (100) can only take the time-reversal invariant values

$$\theta = 0, \quad \text{or} \quad \theta = \pi. \tag{101}$$

Even if the chiral-symmetry breaking scenario discussed here occurs in adjoint QCD, we cannot reliably determine which possibility in (101) is realized. However, we will see in section 5.3 that the value $\theta = \pi$ appears more natural from the point of view of deformed $\mathcal{N} = 2$ SYM theory. In section 3.5 we will show that (if present) this value of $\theta$ has interesting implications for the physics of the confining strings.

## 3.3 Aspects of the $\mathbb{CP}^1$ model on non-spin manifolds

As was discussed in sections 2.4.2 and 2.4.3, certain features of adjoint QCD, and in particular certain 't Hooft anomalies, only become visible if we place the theory on a non-spin manifold $\mathcal{M}_4$. We must therefore also explain how to place the $\mathbb{CP}^1$ model on non-spin manifolds,

---

[45]This is particularly transparent in the abelian Higgs model description discussed in section 5.3. There $n^*(\omega) = \frac{f^{(2)}}{2\pi}$, where $f^{(2)}$ the field strength of a dynamical $U(1)$ gauge field.

so that the anomalies match.[46] This may seem trivial, because the low-energy pion fields are bosons. (On the other hand, the solitons of the model are fermions.) To proceed, we must remember that the coupling to non-spin manifolds proceeds via background fields for the $SU(2)_R$ symmetry that are connections on an $SO(3)_R$ bundle $\mathcal{B}_R$ that satisfies the condition $w_2(\mathcal{B}_R) = w_2(\mathcal{M}_4)$ in (59). We must therefore couple the model to $SU(2)_R$ background gauge fields so that this constraint is satisfied.

Consider the Kähler form $n^*(\omega)$. In the absence of background fields, it has the following explicit expression in terms of the pion fields

$$n^*(\omega) = \frac{1}{8\pi}\, \varepsilon_{IJK}\, n^I dn^J \wedge dn^K\,. \tag{102}$$

In the presence of $SU(2)_R$ background fields, we must replace the exterior derivative $d$ by its $SU(2)_R$ covariant version $d_R$.[47] However, the resulting $n^*(\omega)(A_R^{(1)})$ is not closed and must be corrected by adding an $SU(2)_R$-invariant term $\sim n^I F_R^{(2)I}$, where $F_R^{(2)I}$ is the $SU(2)_R$ background field strength 2-form,[48]

$$n^*(\omega)(A_R^{(1)}) = \frac{1}{8\pi}\left(\varepsilon_{IJK}\, n^I d_R n^J \wedge d_R n^K - 2n^I F_R^{(2)I}\right)\,. \tag{103}$$

This expression is closed, $SU(2)_R$-invariant, and reduces to (102) in the absence of background fields.

If the dynamical field $n_I$ resides in the vacuum configuration $n_I = \delta^{I3}$, the expression in (103) reduces to

$$n^*(\omega)(A_R^{(1)}) = -\frac{1}{4\pi} F_R^{(2)3} = -\frac{1}{2\pi}\, G_R^{(2)}\,. \tag{104}$$

Here $G_R^{(2)}$ is the conventionally normalized background field strength 2-form corresponding to the unbroken $U(1)_R \subset SU(2)_R$ Cartan subgroup. Recall that $G_R^{(2)}$ is the field strength of a spin$^c$ connection. This means that $\frac{2}{2\pi}\, G_R^{(2)}$ is an integral cohomology class congruent to $w_2(\mathcal{M}_4)$ modulo 2 (i.e. it defines an integral lift of $w_2(\mathcal{M}_4)$). We would like to turn this into an $SU(2)_R$ covariant statement that is valid for all configurations of the pion field. We are therefore led to demand that the expression $n^*(\omega)\left(A_R^{(1)}\right)$ in (103), which involves both the dynamical pion fields $n^I$ and the $SU(2)_R$ background fields, is such that $2n^*(\omega)\left(A_R^{(1)}\right)$ defines an integral cohomology class that is congruent to $w_2(\mathcal{M}_4)$ modulo 2,

$$2n^*(\omega)\left(A_R^{(1)}\right) = w_2 \;(\mathrm{mod}\; 2)\,. \tag{105}$$

We will perform the functional integral over the dynamical pions so that this constraint is satisfied. Not coincidentally, the constraint (105) is analogous to the constraint satisfied by the field strength of a dynamical spin$^c$ gauge field (see section 5.3). One a spin manifold, where $w_2(\mathcal{M}_4) = 0$, (105) implies that $n^*(\omega)$ is an integral cohomology class, as in (97).

Having modified the definition of the functional integral on non-spin manifolds, we must reexamine the couplings (99) and (100):

1.) Since $n^*(\omega)$ is no longer an integral cohomology class when $\mathcal{M}_4$ is not spin, the coupling (99) to the $\mathbb{Z}_2^{(1)}$ background field $B_2$ is no longer automatically invariant under $B_2$ gauge transformations. It is therefore natural to choose an integral lift $\widetilde{w}_2(\mathcal{M}_4)$

---

[46]Much of the discussion in this section becomes very natural when viewed through the lens of the abelian Higgs model described in section 5.3.

[47]Here $d_R n^I = dn^I + \epsilon_{IJK} A_R^{(1)J} n^K$.

[48]Recall that $F_R^{(2)I} = dA_R^{(1)I} + \frac{1}{2}\varepsilon_{IJK} A_R^{(1)J} \wedge A_R^{(1)K}$.

of $w_2(\mathcal{M}_4)$ and modify the coupling as follows,

$$\exp\left(i\pi \int_{\mathcal{M}_4} B_2 \cup n^*(\omega)\left(A_R^{(1)}\right) + \frac{i\pi}{2} \int_{\mathcal{M}_4} B_2 \cup \widetilde{w}_2(\mathcal{M}_4)\right). \tag{106}$$

Using (105), we can check that (106) is invariant under $B_2$ background gauge transformations. This is because $B_2$ only couples to $n^*(\omega)\left(A_R^{(1)}\right) + \frac{1}{2}\widetilde{w}_2(\mathcal{M}_4)$, which is an integral cohomology class.

2.) Due to (105), the $\theta$ term (100) is no longer $2\pi$-periodic if $\mathcal{M}_4$ is not a spin manifold. This can be cured by including a purely gravitational coupling that involves the signature $\sigma(\mathcal{M}_4)$ of the spacetime 4-manifold,

$$\exp\left(\frac{i\theta}{2} \int_{\mathcal{M}_4} n^*(\omega)\left(A_R^{(1)}\right) \cup n^*(\omega)\left(\left(A_R^{(1)}\right)\right) - \frac{i\theta}{8}\sigma(\mathcal{M}_4)\right). \tag{107}$$

With this definition, $\theta$ still has periodicity $2\pi$ (see for instance appendix A of [18]).[49] In particular, the value $\theta = \pi$ remains invariant under time-reversal.

We can now verify that the definition of the $\mathbb{CP}^1$ model given above reproduces the mixed 't Hooft anomaly (70) between the $\mathbb{Z}_2^{(1)}$ center symmetry and $w_2(\mathcal{M}_4)$ that was discussed in section 2.4.3. As we did there, we choose lifts $\widetilde{B}_2$, $\widetilde{w}_2(\mathcal{M}_4)$ of $B_2$, $w_2(\mathcal{M}_4)$ to integral 2-cochains and examine the behavior of the model under changes of these lifts,

$$\widetilde{B}_2 \to \widetilde{B}_2 + 2x\,, \qquad \widetilde{w}_2(\mathcal{M}_4) \to \widetilde{w}_2(\mathcal{M}_4) + 2y\,. \tag{109}$$

Here $x, y$ are integral 2-cochains. Since $n^*(\omega)\left(A_R^{(1)}\right)$ is summed over all configurations subject to the constraint (105), it does not shift under (109). The coupling in (106) then produces the expected anomalous variation (67) of the partition function,

$$Z \to Z \exp\left(i\pi \int_{\mathcal{M}_4} B_2 \cup y\right). \tag{110}$$

By contrast, the $\theta$-term (107), and all other terms constructed using only $n^*(\omega)(A_R^{(1)})$, are invariant under (109).

## 3.4 Adding fermion masses

We will now follow the discussion in section (2.5) and deform the model by adding masses for the fermions $\lambda_\alpha^{iA}$. When the fermion masses are large, the theory flows to adjoint QCD with $N_f = 0, 1$. In fact, we will reproduce the expected vacuum structure of these theories even when the fermion masses are small.[50] This makes the $\mathbb{CP}^1$ phase very economical – no phase transitions are required to occur as we increase the fermion masses.

---

[49]Note that any lift $\widetilde{w}_2 \in H^2(\mathcal{M}_4, \mathbb{Z})$ of $w_2(\mathcal{M}_4)$ to an integral cohomology class satisfies

$$\widetilde{w}_2 \cup \widetilde{w}_2 = \sigma(\mathcal{M}_4) \,(\text{mod } 8)\,. \tag{108}$$

This is because $w_2(\mathcal{M}_4) \cup x = x \cup x \,(\text{mod } 2)$ for any $x \in H^2(\mathcal{M}_4, \mathbb{Z})$, so that $\widetilde{w}_2$ is a characteristic vector of the intersection form on $H^2(\mathcal{M}_4, \mathbb{Z})$ (see e.g. section 1.1.3 of [21]).

[50]A similar phenomenon generically occurs in the chiral lagrangian for conventional QCD with massless fundamental flavors, see e.g. [5] for a recent discussion.

We consider the mass deformation $\Delta V$ in (74) in the parametrization (78). Together with (84) and (87), we can then express the resulting potential $\Delta V_\pm$ in the two disconnected $\mathbb{CP}^1_\pm$ sigma models as

$$\Delta V_\pm = -x_\pm^I \cdot n^I,\tag{111}$$

where the $SO(3)_R$ vectors $x_\pm^I$ are functions of the mass parameters $\chi, \alpha, \beta$ in (78),

$$\begin{aligned}
x_+^I &= \Big((\alpha-\beta)\sin\chi,\,(\alpha+\beta)\cos\chi,\,0\Big),\\
x_-^I &= \Big((\alpha-\beta)\cos\chi,\,-(\alpha+\beta)\sin\chi,\,0\Big).
\end{aligned}\tag{112}$$

To start, assume that the mass parameters are sufficiently generic so that both $x_+^I$ and $x_-^I$ are non-vanishing. (We will subsequently discuss degenerate cases.) The ferromagnetic form of the potential (111) then implies that there is a unique minimum on $\mathbb{CP}^1_+$ and $\mathbb{CP}^1_-$ in which the pion field $n^I$ aligns with the vectors $x_+^I$ and $x_-^I$, respectively. This leads to two local minima with energy $\Delta V_\pm = -|x_\pm^I|$, where

$$|x_\pm^I|^2 = \alpha^2 + \beta^2 \pm 2\alpha\beta\cos 2\chi.\tag{113}$$

The minimum that is energetically favored, and hence the global minimum, is determined by the vector with the larger norm (113).

We can now explore the implications of the preceding discussion for different choices of the mass parameters listed in section 2.5:

- Consider first the special case $m_{22} = \beta = 0$. As explained in point 2.) of section 2.5, this leads to $N_f = 1$ adjoint QCD, i.e. pure $\mathcal{N} = 1$ SYM. We see from (113) that the two minima on $\mathbb{CP}^1_+$ and $\mathbb{CP}^1_-$ are exactly degenerate. This is in perfect agreement with the two vacua that are expected in $\mathcal{N} = 1$ SYM with gauge group $SU(2)$.

- If $\alpha$ and $\beta$ are both non-zero and generic, both fermions acquire masses. As explained in point 1.) of section 2.5, taking these masses to be large leads to pure YM theory with $\theta = 4\chi$. Instead we will consider what happens when they are small, as predicted by (113):

  i) For $0 \leq \theta < \pi$ (corresponding to $0 \leq \chi < \frac{\pi}{4}$) we see that $|x_+^I| > |x_-^I|$, so that there is a unique minimum on $\mathbb{CP}^1_+$. Similarly, for $\pi < \theta < 2\pi$ (corresponding to $\frac{\pi}{4} < \chi < \frac{\pi}{2}$) we find $|x_+^I| < |x_-^I|$ and there is a unique minimum on $\mathbb{CP}^1_-$. This is consistent with the standard expectation that pure $SU(2)$ YM theory has a unique vacuum for all $\theta \neq \pi$.

  ii) At the special value $\theta = \pi$ (which corresponds to $\chi = \frac{\pi}{4}$) we find that $|x_+^I| = |x_-^I|$, so that the minima on $\mathbb{CP}^1_+$ and $\mathbb{CP}^1_-$ are exactly degenerate. As was explained below (80), this is consistent with the fact that pure YM theory at $\theta = \pi$ has a mixed 't Hooft anomaly that obstructs a single, trivial gapped vacuum [4]. This simplest option is that $\mathsf{T}$ is spontaneously broken, leading to two degenerate vacua, and this is the option that is realized here in the small mass limit.

Let us comment on the case where the masses are such that one of the vectors $x_\pm^I$ vanishes, e.g. $x_+^I = 0$. (The case $x_-^I = 0$ is similar.) In this case the potential $\Delta V_+$ on $\mathbb{CP}^1_+$ vanishes at leading $\mathcal{O}(m)$ order in the mass deformation. However, it follows from (113) that $x_\pm^I$ cannot simultaneously vanish (unless $\alpha = \beta = 0$). Consequently, the potential $\Delta V_-$ on $\mathbb{CP}^1_-$ is non-trivial at $\mathcal{O}(m)$ and leads to a negative $\mathcal{O}(m)$ shift in the vacuum energy. It follows that the unique $\mathbb{CP}^1_-$ minimum is also the global minimum. The conclusions described above therefore remain valid.

We can extend the preceding discussion to justify the claim in (85) that the constant $C$ determining the vev of the chiral condensate must either be real or purely imaginary. To see this, assume that $C \in \mathbb{C}$ is complex and turn on real masses $m_{11} = m_{22} = \alpha > 0$ (this corresponds to $\chi = 0$ and $\alpha = \beta$ in (78)). As was discussed around (79), these masses preserve the time-reversal symmetry $\widetilde{\mathsf{T}} = r^2 \mathsf{T}$. Moreover, they lead to a euclidean functional integral with strictly positive measure. Therefore the arguments in [48] apply and we can conclude that $\widetilde{\mathsf{T}}$ should not be spontaneously broken.[51] If we assume that $\alpha$ is small, we can compute the $\mathcal{O}(\alpha)$ potential in the $\mathbb{CP}^1$ description,

$$\Delta V = -\alpha \left( C + \overline{C} \right) n_2 \,. \tag{114}$$

This leaves only two possibilities: either $C \in i\mathbb{R}$ is purely imaginary, or $n_2 = \pm 1$. In the latter case the chiral condensate is given by $\langle \mathcal{O}^I \rangle = \pm C \delta^{I2}$, which only preserves $\widetilde{\mathsf{T}}$ if $C \in \mathbb{R}$ is real (see the discussion around (91)). Therefore $C$ can only be real or purely imaginary.

## 3.5 Possibility of a topological insulator on the confining string

We now describe some additional aspects of the worldsheet theory of the confining strings discussed in section 3.2.2. Note that the presence of the massless $\mathbb{CP}^1$ pion fields means that the worldsheet theory does not make sense as a genuine two-dimensional quantum field theory, but only as a coupled 2d-4d system.[52] This changes once we turn on small fermion masses, as in section 3.4 above. The bulk dynamics is then gapped, and hence there is a meaningful two-dimensional effective field theory on the string worldsheet.

Here we focus on masses that preserve the $U(1)_R \subset SU(2)_R$ Cartan subgroup, as well as the $\widetilde{\mathsf{T}} = r^2 \mathsf{T}$ time-reversal symmetry. In the notation of (78), this corresponds to $\chi = 0$ and $\alpha = \beta$, so that

$$m_{11} = m_{22} = \alpha > 0 \,. \tag{115}$$

It follows from the discussion in section 3.4 that the vacuum for the pion fields is at the point $(n_1, n_2, n_3) = (0, 1, 0)$. If the masses in (115) are taken to be large, we flow to pure $SU(2)$ YM theory with $\theta = 0$, which is expected to be in a trivial gapped phase.

Recall that the $\mathbb{CP}^1$ model admits a $\theta$-term of the form (100), which can take the time-reversal invariant values $\theta = 0, \pi$ in (101). We now show that this induces a $\theta$-angle of the same value on the worldsheet of the minimal confining string. We parametrize the pion field in the presence of the string worldsheet $\Sigma_2$ as follows,

$$n^*(\omega) = \mathrm{PD}(\Sigma_2) + n^*(\omega)_{\mathrm{fluct.}} \,. \tag{116}$$

Here $\mathrm{PD}(\Sigma_2)$ is the Poincaré dual to the two-dimensional string worldsheet. Its presence in (116) guarantees that the string charge (97) integrates to 1. The term $n^*(\omega)_{\mathrm{fluct.}}$ represents the fluctuating pion fields in the presence of the strings. Substituting into (100), we find that the bulk $\theta$-term gives rise to the following coupling on the string worldsheet,[53]

$$\exp\left( \frac{i\theta}{2} \int_{\mathcal{M}_4} n^*(\omega) \wedge n^*(\omega) \right) \quad \supset \quad \exp\left( i\theta \int_{\Sigma_2} n^*(\omega)_{\mathrm{fluct.}} \right) \,. \tag{117}$$

In order to uncover the significance of the term in (117) we turn on a background gauge field for the unbroken $U(1)_R \subset SU(2)_R$ symmetry. It is a spin$^c$ connection, with field

---

[51]Since we are discussing $SU(2)$ gauge theory, which does not admit a charge-conjugation symmetry, what is called $\mathsf{T}$ here corresponds to $\mathsf{CT}$ in [48].

[52]A consequence of this fact is that the minimal solitonic string of the $\mathbb{CP}^1$ model has non-normalizable zero modes, which label different superselection sectors for the soft pions [85].

[53]Here we use the fact that $\int_{\mathcal{M}_4} \mathrm{PD}(\Sigma_2) \cup x^{(2)} = \int_{\Sigma_2} x^{(2)}$ for any closed 2-form $x^{(2)}$.

strength $G_R^{(2)}$. In the presence of $SU(2)_R$ background fields, we must replace $n^*(\omega)$ by the quantity $n^*(\omega)(A^{(1)})$ in (103). As in (104), this leads to a non-trivial expression involving the background field strength $G_R^{(2)}$, even when the pion fields reside in their vacuum configuration,

$$n^*(\omega)\left(A_R^{(1)}\right) \quad \longrightarrow \quad -\frac{1}{2\pi}\, G_R^{(2)}. \tag{118}$$

Substituting back into the worldsheet coupling (117), we find a two-dimensional $\theta$-term for the $U(1)_R$ background gauge field,

$$\exp\left(-\frac{i\theta}{2\pi}\int_{\Sigma_2} G_R^{(2)}\right). \tag{119}$$

This term is trivial when $\theta = 0$. However, if $\theta$ takes the non-trivial $\widetilde{\mathsf{T}}$-invariant value $\theta = \pi$, then the coupling (119) implies that the worldsheet of the confining string is a topological insulator protected by the unbroken $\widetilde{\mathsf{T}}$ and $U(1)_R$ symmetries.

Let us contemplate the fate of this topological insulator as we increase the bulk fermion mass $\alpha$ in (115). Since the topological insulator is protected by symmetries, it should persist for sufficiently small masses. As we make the masses larger, we eventually flow to pure YM theory at $\theta = 0$, which is expected to a trivial gapped theory. Its confining strings are therefore also expected to be in a trivial gapped phase. If this is the case, there must be some critical value $\alpha_*$ of the bulk fermion mass at which the theory on the string worldsheet undergoes a phase transition, while the bulk remains gapped.

The simplest possibility for this transition is that a two-dimensional Dirac fermion becomes massless on the string worldsheet when the bulk mass is tuned to the critical value $\alpha = \alpha_*$. (Here it is important that $G_R^{(1)}$ is a spin$^c$ background gauge field, which can couple to the Dirac fermion with unit charge.) It is not possible to study this transition within the $\mathbb{CP}^1$ model. However, precisely this scenario is realized in the abelian Higgs model completion of the $\mathbb{CP}^1$ model analyzed in section 5.3.

# 4 The Seiberg-Witten solution of the $\mathcal{N} = 2$ SYM theory

In this section we review some additional details about the IR solution of $SU(2)$ $\mathcal{N} = 2$ SYM found in [9]. In particular, we spell out how some of the symmetries and 't Hooft anomalies of the UV theory are realized in the low-energy description.

## 4.1 The Coulomb branch sigma model

As was reviewed around (6), the low-energy dynamics at every point $u \in \mathbb{C}$ on the Coulomb branch is described by an abelian $\mathcal{N} = 2$ vector multiplet $\left(\varphi, \rho_\alpha^i, f^{(2)}\right)$. (An exception are the monopole and dyon points, where there are additional massless particles.) The leading (irrelevant) interactions of these fields are described by an $\mathcal{N} = 2$ special Kähler sigma model characterized by a holomorphic prepotential $\mathcal{F}(\varphi)$. The lagrangian schematically takes the form

$$\mathcal{L} \sim Q^4 \mathcal{F}(\varphi) + (\text{h.c.}). \tag{120}$$

The fact that $\mathcal{F}$ is a function of $\varphi$ implies that the kinetic terms for $\varphi$ and $\rho_\alpha^i$ have a non-trivial $\varphi$-dependent metric. Similarly, the complexified $U(1)$ gauge coupling $\tau$ depends on $\varphi$,

and moreover this dependence is holomorphic,[54]

$$\tau(\varphi) = \frac{\theta(\varphi)}{2\pi} + \frac{2\pi i}{e^2(\varphi)} = \mathcal{F}''(\varphi). \tag{122}$$

In a free theory, where $\tau$ is constant, $\mathcal{F} = \frac{1}{2}\tau\varphi^2$.

In order to discuss the action of electric-magnetic duality, it is convenient to trade the prepotential $\mathcal{F}$ (which does not transform in a simple way under duality) for the quantity

$$\varphi_D = \mathcal{F}'(\varphi). \tag{123}$$

Note that $\tau = \mathcal{F}'' = \frac{d\varphi_D}{d\varphi}$. The pair $(\varphi_D, \varphi)$ defines a set of special coordinates that transforms as a doublet of the $SL(2,\mathbb{Z})$ duality group. The solution of [9] is expressed in terms of a particular set of special coordinates denoted by $a$ and $a_D$. In general, $\varphi$ is an integer linear combination of $a, a_D$ that depends on the duality frame. An electric-magnetic duality transformation is characterized by an $SL(2,\mathbb{Z})$ matrix. Under such a transformation, the special coordinates $(a_D, a)$ and the gauge coupling $\tau$ transform as follows,

$$\begin{pmatrix} a_D' \\ a' \end{pmatrix} = \begin{pmatrix} m & n \\ p & q \end{pmatrix} \begin{pmatrix} a_D \\ a \end{pmatrix}, \qquad \tau' = \frac{m\tau + n}{p\tau + q}, \qquad \begin{pmatrix} m & n \\ p & q \end{pmatrix} \in SL(2,\mathbb{Z}). \tag{124}$$

In order to express the mapping from the variable $u$ that parametrizes the Coulomb branch to the low-energy field $\varphi$, we must express the special coordinates as functions of $u$. This is achieved by the following definite integrals,[55]

$$a(u) = \frac{1}{\pi\sqrt{2}} \int_{-\Lambda^2}^{\Lambda^2} dx \, \frac{\sqrt{u-x}}{\sqrt{\Lambda^4 - x^2}}, \qquad a_D(u) = \frac{\sqrt{2}i}{\pi} \int_{\Lambda^2}^{u} dx \, \frac{\sqrt{u-x}}{\sqrt{x^2 - \Lambda^4}}. \tag{125}$$

This presentation of $a(u)$ and $a_D(u)$ as definite integrals is valid when $\text{Re}(u) > \Lambda^2$. When $|u|$ is large, they behave as

$$a(u) \approx \sqrt{u/2}, \qquad a_D(u) \approx \frac{i\sqrt{2u}}{\pi} \log(u/\Lambda^2), \quad |u| \gg \Lambda^2. \tag{126}$$

The behavior of $a(u)$ reflects the fact that in this weakly-coupled large-$u$ region the $SU(2)$ adjoint scalar field $\phi$ of the UV theory acquires a vev $\phi = \begin{pmatrix} a & 0 \\ 0 & -a \end{pmatrix}$ that higgses $SU(2) \to U(1)$. Then $u = \text{tr}\,\phi^2 = 2a^2$. Meanwhile the asymptotic behavior of $a_D(u)$ in (126) encodes the logarithmic running of the asymptotically free gauge coupling,

$$\tau = \frac{da_D}{da} \approx \frac{2i}{\pi} \log(u/\Lambda^2) + \mathcal{O}(1), \quad |u| \gg \Lambda^2. \tag{127}$$

---

[54]We define $\tau = \frac{\theta}{2\pi} + \frac{2\pi i}{e^2}$, rather than the more common $\frac{\theta}{2\pi} + \frac{4\pi i}{e^2}$, to ensure that the $U(1)$ gauge theory has conventionally normalized Maxwell kinetic terms, and a $\theta$-term with periodicity $2\pi$ on spin manifolds and $4\pi$ on non-spin manifolds, assuming that $\frac{f^{(2)}}{2\pi}$ has integral fluxes. In lorentzian signature, the lagrangian is

$$\mathscr{L} = \frac{i}{16\pi} \left( \tau f_{\mu\nu}^+ f^{+\mu\nu} - \overline{\tau} f_{\mu\nu}^- f^{-\mu\nu} \right) = -\frac{1}{4e^2} f_{\mu\nu} f^{\mu\nu} - \frac{\theta}{32\pi^2} \varepsilon^{\mu\nu\rho\sigma} f_{\mu\nu} f_{\rho\sigma}. \tag{121}$$

Here $f_{\mu\nu}^\pm = \frac{1}{2} \left( f_{\mu\nu} \pm \frac{i}{2} \varepsilon_{\mu\nu\rho\sigma} f^{\rho\sigma} \right)$. Since we use the conventions of [1] (with $\varepsilon^{0123} = 1$), a Wick rotation $x^0 = -ix^4$ to euclidean signature (with $\varepsilon_{1234} = 1$) maps $f_{\mu\nu}^+$ and $f_{\mu\nu}^-$ to the standard self-dual and anti-self-dual components of $f_{\mu\nu}$. The euclidean lagrangian takes the form $\mathscr{L}_E = \frac{1}{4e^2} f_{\mu\nu} f^{\mu\nu} + \frac{i\theta}{32\pi^2} \varepsilon^{\mu\nu\rho\sigma} f_{\mu\nu} f_{\rho\sigma}$.

[55]Our conventions for the special coordinates $a$ and $a_D$ are those of [10]. The resulting monodromies therefore lie in the subgroup $\Gamma^0(4) \subset SL(2,\mathbb{Z})$.

Note that $\tau$ is purely imaginary as long as $u$ is real, so that the effective $\theta$-angle vanishes. This is consistent with the fact that time-reversal is unbroken on the real $u$-axis. As we will see below, this behavior persists until we reach the monopole point $u = \Lambda^2$.

The values of $a(u)$ and $a_D(u)$ outside the region $\text{Re}(u) > \Lambda^2$ can be obtained by analytic continuation. We will use the following useful representation in terms of hypergeometric functions (see for instance [86]),

$$
\begin{aligned}
a(u) &= \sqrt{\frac{(\Lambda^2 + u)}{2}} \, {}_2F_1\left(-\frac{1}{2}, \frac{1}{2}, 1 \, ; \, \frac{2\Lambda^2}{\Lambda^2 + u}\right), \\
a_D(u) &= \frac{i\left(u - \Lambda^2\right)}{2\Lambda} \, {}_2F_1\left(\frac{1}{2}, \frac{1}{2}, 2 \, ; \, \frac{\Lambda^2 - u}{2\Lambda^2}\right).
\end{aligned}
\tag{128}
$$

These are branched functions. Specifically, $a(u)$ has a branch cut just below the real axis, which runs from $u = -\infty$ to the monopole point $u = \Lambda^2$. Similarly, $a_D(u)$ has a branch cut just below the real axis that runs from $u = -\infty$ to the dyon point $u = -\Lambda^2$. Across these cuts, the functions (128) jump by $SL(2, \mathbb{Z})$ duality transformations.[56] Below we will often work on the real $u$-axis. When we evaluate the functions (128) there we will implicitly evaluate them slightly above the real axis, i.e. we choose an $i\varepsilon$ prescription of the form $a(u + i\varepsilon)$, $a_D(u + i\varepsilon)$ with $u \in \mathbb{R}$. For future reference, we note that the formulas (128) can be inverted to give a formula for $u$ in terms of $a$, $a_D$ and the prepotential [87],

$$
u = 2\pi i \left(\mathcal{F}(a) - \frac{1}{2} a a_D\right).
\tag{129}
$$

This formula can also be derived by promoting the strong-coupling scale $\Lambda$ to an $\mathcal{N} = 2$ background chiral superfield [38]. Unlike $u$, the expression on the right-hand side of (129) is not obviously duality invariant, but the prepotential can be defined in such a way that this is in fact the case.

We would like to use the formulas (128) to clarify the behavior of the effective $\theta$-angle on the real $u$-axis, where time-reversal symmetry is preserved. Therefore $\theta$ necessarily takes one of the two time-reversal invariant values $\theta = 0$ or $\theta = \pi$, but importantly this statement need only hold up to an $SL(2, \mathbb{Z})$ duality transformation. As was explained below (127), the $\theta$ angle vanishes when $u \gg \Lambda^2$. Using (128) it can be checked that this behavior persists until the monopole point $u = \Lambda^2$. If we continue to $u < \Lambda^2$ in this duality frame, we find that the $\theta$ angle continuously evolves from $\theta = 0$ to $\theta = -2\pi$ at $u = 0$ and ultimately to $\theta = -4\pi$ at the dyon point $u = -\Lambda^2$, beyond which it remains constant.

In order to see that this behavior is consistent with time-reversal, we can perform an $S$-duality transformation to go to the duality frame that is appropriate near the monopole point, where $\varphi = a_D$ and $\tau_D = -\frac{1}{\tau}$. The dual $\theta$-angle $\theta_D = 2\pi \, \text{Re} \, \tau_D$ vanishes when $u > \Lambda^2$ and jumps discontinuously as we cross the monopole point,

$$
\theta_D = \begin{cases} 0, & u > \Lambda^2, \\ \pi, & -\Lambda^2 < u < \Lambda^2. \end{cases}
\tag{130}
$$

Therefore $\theta_D$ always takes a value consistent with time-reversal symmetry. The physical mechanism behind the jump in (130) is that the monopole hypermultiplet $\left(h_i, \psi_{+\alpha}, \psi_{-\alpha}\right)$ becomes massless when $u = \Lambda^2$. As we pass through that point along the real axis, the mass of the Dirac fermion comprised of $\psi_{+\alpha}$ and $\psi_{-\alpha}$ passes through zero, and this leads to a jump by $\pi$ in the $\theta$-angle.

---

[56]The presentation in (128) is useful in computer software programs such as `Mathematica`. Other presentations in terms of elliptic functions in general do not implement the same choice of branch cuts.

## 4.2 Symmetries and 't Hooft anomalies

The realization of the global symmetries on the Coulomb branch was already discussed in section 1.4. Here we present some additional details. In particular, we match some of the 't Hooft anomalies of the UV theory (see section 2.4) on the Coulomb branch.

### 4.2.1 Generic points on the Coulomb branch

We begin by discussing points $u \neq 0, \pm \Lambda^2$ on the Coulomb branch. The $SU(2)_R$ symmetry is unbroken and acts on the gaugino $\rho_\alpha^i$ of the low-energy abelian vector multiplet. This matches the Witten anomaly for the $SU(2)_R$ symmetry.

The $\mathbb{Z}_2^{(1)}$ 1-form global symmetry associated with the center of the UV gauge group is embedded into the accidental $U(1)_{\text{electric}}^{(1)} \times U(1)_{\text{magnetic}}^{(1)}$ 1-form symmetries of the low-energy $U(1)$ gauge theory. Massive charged particles explicitly break it to $\mathbb{Z}_2^{(1)}$. Whether this $\mathbb{Z}_2^{(1)}$ is embedded into the electric or magnetic 1-form symmetry depends on the duality frame. For instance, if we use the duality frame associated with the special coordinate $a(u)$ and examine the weak-coupling region $|u| \gg \Lambda^2$, we find a massive W-boson of electric charge 2, which breaks $U(1)_{\text{electric}}^{(1)} \to \mathbb{Z}_2^{(1)}$, and a magnetic monopole of magnetic charge 1, which completely breaks $U(1)_{\text{magnetic}}^{(1)}$.

Recall from (5) that the $\mathsf{Z}_8$ symmetry generated by $\mathsf{r}$ acts via $\mathsf{r}(\phi^A) = i\varphi^A$, and hence $\mathsf{r}(u) = -u$. As long as $u \neq 0$, this symmetry is therefore spontaneously broken to its $\mathsf{Z}_4 \subset \mathsf{Z}_8$ subgroup. The unbroken generator $\mathsf{r}^2$ acts on the abelian vector multiplet fields on the Coulomb branch as follows,

$$\mathsf{r}^2(\varphi) = \varphi, \qquad \mathsf{r}^2(\rho_\alpha^i) = -i\rho_\alpha^i, \qquad \mathsf{r}^2(f^{(2)}) = -f^{(2)}. \tag{131}$$

This a discrete $R$-symmetry, under which $\mathsf{r}^2(Q_\alpha^i) = -iQ_\alpha^i$.[57] If we track the symmetry $\mathsf{r}^2$ to the weak-coupling region of the Coulomb branch, we find that the IR action in (131) differs from the action (39) on the UV fields by an overall sign. This sign corresponds to the action of charge-conjugation symmetry $\mathsf{C}$, which is an exact symmetry of the low-energy theory. It corresponds to the non-trivial central element $\mathsf{C} = -\mathbb{1} \in SL(2, \mathbb{Z})$, which is a symmetry for every value of $\tau$. In the nonabelian gauge theory description that is appropriate when $|u| \gg \Lambda^2$, $\mathsf{C}$ becomes the Weyl element of the $SU(2)$ gauge group. We are therefore free to mix the global symmetry generated by $\mathsf{r}^2$ with $\mathsf{C}$ to obtain (131).

On spin manifolds, the only non-trivial 't Hooft anomaly that involves the unbroken $\mathsf{r}^2$ symmetry is a mixed anomaly with the $SU(2)_R$ symmetry, which was discussed around (48).[58] (The 't Hooft anomalies of the Coulomb-branch theory on non-spin manifolds are discussed in section 4.3.) If we turn on an $SU(2)_R$ background field with instanton number $n_R = 1$, the partition functions shifts as follows under the action of $\mathsf{r}^2$,

$$\mathsf{r}^2(Z) = \exp\left(\frac{i\pi}{2}\kappa_{\mathsf{r}R^2}\right). \tag{132}$$

In the UV, there are three gauginos of $\mathsf{U}(1)_\mathsf{r}$ charge 1, so that $\kappa_{\mathsf{r}R^2}^{\text{UV}} = 3$ (see (45)). In the IR there is only one gaugino, and the transformation rule (131) implies that it effectively has $\mathsf{U}(1)_\mathsf{r}$ charge $-1$, so that $\kappa_{\mathsf{r}R^2}^{\text{IR}} = -1$. (Recall that $\mathsf{U}(1)_\mathsf{r}$ is the classical symmetry that is broken to $\mathsf{Z}_8$ by an ABJ anomaly.) The anomaly (132) is then matched because $3 = -1 \pmod 4$.

We would like to also identify the action of the time-reversal symmetry $\mathsf{T}$, whose action on the UV fields was defined in (40). By going to the weak-coupling region, where $a$ arises

---

[57]Note that (131) is a symmetry of any special Kähler sigma model (120).

[58]The 't Hooft anomalies for the broken $\mathsf{Z}_8$ symmetry are matched because the vacua related by the broken generator $\mathsf{r}$ have different local counterterms for the background fields.

from the UV scalar field $\phi^A$ by adjoint higgsing (see the discussion below (126)), we find that $\mathsf{T}(a) = a$. The action on the other fields in the abelian vector multiplet then follows from supersymmetry (recall from (41) that $\mathsf{T}(Q_\alpha^i) = -iQ_i^\alpha$),

$$\mathsf{T}(a) = a\,, \qquad \mathsf{T}(\rho_\alpha^i) = i\rho_i^\alpha\,, \qquad \mathsf{T}(f^{(2)}) = f^{(2)}\,. \tag{133}$$

Here the last equation means that each tensor index of $f_{\mu\nu}$ is acted on by the time-reversal Lorentz transformation $\mathcal{T}_\mu{}^\nu = \mathrm{diag}(-1, 1, 1, 1)$. It follows from (133) that the magnetic and electric charges $(n_m, n_e)$ transform as $\mathsf{T}(n_m, n_e) = (n_m, -n_e)$. Note that $\mathsf{T}^2 = 1$ when acting on the fields in (133).[59]

Under an $S$-duality transformation, the $\mathsf{T}$-operation (133) combines with charge conjugation $\mathsf{C}$, so that

$$\mathsf{T}(a_D) = -a_D\,, \qquad \mathsf{T}(\rho_{D\alpha}^i) = -i\rho_{Di}^\alpha\,, \qquad \mathsf{T}(f_D^{(2)}) = -f_D^{(2)}\,. \tag{134}$$

Again we find that $\mathsf{T}^2 = 1$. In this duality frame, electric and magnetic charges transform as $\mathsf{T}(n_m, n_e) = (-n_m, n_e)$. The action of $\mathsf{T}$ in a general duality frame can be deduced by writing $\varphi$ as a linear combination of $a$, $a_D$ and using (133), (134).

### 4.2.2 The origin of the Coulomb branch

At the origin $u = 0$ of the Coulomb branch, the $\mathbb{Z}_8$ symmetry is unbroken. So far we have so far only identified the action (131) of the $\mathbb{Z}_4 \subset \mathbb{Z}_8$ subgroup generated by $\mathsf{r}^2$. In order to determine the action of $\mathsf{r}$, we begin with the observation that $\mathsf{r}^2(f^{(2)}) = -f^{(2)} = \mathsf{C}(f^{(2)})$. Here $\mathsf{C} = -1 \in SL(2, \mathbb{Z})$ is the charge conjugation operation discussed above. Since $\mathsf{r}$ must be a square root of this action, it must (roughly speaking) act on the gauge field $f^{(2)}$ by phase. As we will now explain, this is possible if $\mathsf{r}$ acts via the electric-magnetic $S$-duality transformation.

In general, an $SL(2, \mathbb{Z})$ duality transformation maps a given $U(1)$ gauge theory (with coupling $\tau$) to a physically equivalent abelian gauge theory with a different coupling. However, there are special values of $\tau$ that are stabilized by an element of $SL(2, \mathbb{Z})$. This particular duality transformation then maps the original theory back to itself and therefore defines a global symmetry. Precisely this phenomenon occurs at the origin $u = 0$ if the Coulomb branch. If we work in the duality frame

$$\varphi = a\,, \qquad \varphi_D = a + a_D\,, \tag{135}$$

we can use (128) to evaluate[60]

$$\varphi(u = 0) = ix\,, \qquad \varphi_D(u = 0) = x\,, \qquad \tau(u = 0) = i\,. \tag{136}$$

Here $x \neq 0$ is a complex constant, whose precise value will not be important for us. The coupling $\tau = i$ is invariant under the $S$-duality transformation, which sends $\tau \to -\frac{1}{\tau}$. In lorentzian signature, it acts on the self-dual and anti-self-dual parts of $f^{(2)}$ via $S(f_\pm^{(2)}) = \pm if_\pm^{(2)}$, so that $S^2 = \mathsf{C}$. It also sends $S(\varphi) = \varphi_D = -i\varphi$ and $S(\varphi_D) = -\varphi = -i\varphi_D$.

Although $S$ is a global symmetry of the $U(1)$ gauge field at $\tau = i$, it does not leave the scalars $\varphi$ and $\varphi_D$ invariant. In order to define a global symmetry of the full $\mathcal{N} = 2$ abelian gauge theory, we must therefore accompany $S$ by a discrete $R$-symmetry transformation $X$ that acts as follows,

$$X(\varphi) = i\varphi\,, \qquad X(\rho_\alpha^i) = e^{\frac{i\pi}{4}}\rho_\alpha^i\,, \qquad X(f^{(2)}) = f^{(2)}\,, \qquad X(Q_\alpha^i) = e^{-\frac{i\pi}{4}}Q_\alpha^i\,. \tag{137}$$

---

[59]This statement is not meaningful on states that carry electric charge, since we can change the value of $\mathsf{T}^2$ on these states by mixing $\mathsf{T}$ with a gauge transformation.

[60]We note that (136) as well as (130) can also be derived using the biholomorphism between the $u$-plane and the $\tau$ upper half-plane revisited recently in [88, 89].

We can now define the action of the $Z_8$ generator as follows,

$$r = XS. \tag{138}$$

Note that this leaves the scalars vevs in (136) invariant and acts via $S$-duality on $f^{(2)}$.

We can use (138) to determine the action of the $Z_8$ generator r on the IR gaugino $\rho_\alpha^i$ at the origin. Using the supersymmetry transformations in appendix B.2 and the fact that $\tau = i$, we can write

$$S(\rho_\alpha^i) = \rho_{D\alpha}^i = -\frac{i}{\sqrt{2}} Q_\alpha^i \varphi_D = -\frac{i}{\sqrt{2}} \mathcal{F}''(\varphi) Q_\alpha^i \varphi = \tau(\varphi) \rho_\alpha^i = i \rho_\alpha^i. \tag{139}$$

Combining this with the transformation rule of $\rho_\alpha^i$ under $X$ in (137), we find that

$$r(\rho_\alpha^i) = e^{\frac{3i\pi}{4}} \rho_\alpha^i. \tag{140}$$

This is indeed a square root of the phase found for the action of $r^2$ in (131).

It follows from (140) that the IR gaugino effectively has $U(1)_r$ charge 3. This leads to exactly the same mixed anomaly coefficients $\kappa_{rR^2} = 3$ and $\kappa_r = 6$ with $SU(2)_R$ and gravity as the three UV gauginos of $U(1)_r$ charge 1. Therefore these mixed 't Hooft anomalies are matched at $u = 0$ on both spin and non-spin manifolds.

The interplay between the $Z_8$ symmetry and electric-magnetic duality is also responsible for matching the mixed anomaly between $Z_8$ and the $\mathbb{Z}_2^{(1)}$ 1-form symmetry. As was discussed around (55), acting with the $Z_8$ generator r can change the partition function by at most a sign, i.e. the anomaly is $\mathbb{Z}_2$-valued.[61] By contrast, the generator $r^2$ does not have a mixed anomaly with the $\mathbb{Z}_2^{(1)}$ symmetry on spin manifolds.

To see that the anomaly is present, we work in a duality frame where $\mathbb{Z}_2^{(1)} \subset U(1)_{\text{electric}}^{(1)}$. This symmetry multiplies odd-charge Wilson lines by $-1$, but leaves even-charge Wilson lines invariant. Gauging the $\mathbb{Z}_2^{(1)}$ symmetry therefore removes the odd-charge Wilson lines from the spectrum. If we would like the new $U(1)$ gauge field to be conventionally normalized, we must rescale it by a factor of 2, i.e. $f_{\text{old}}^{(2)} = 2 f_{\text{new}}^{(2)}$. If $f_{\text{old}}^{(2)}$ was at the self-dual coupling $\tau = i$, this will not be the case for $f_{\text{new}}^{(2)}$. In other words, gauging the $\mathbb{Z}_2^{(1)}$ 1-form symmetry ruins the $S$-duality symmetry and therefore breaks the $Z_8$ symmetry at the origin of the Coulomb branch. This demonstrates the presence of a mixed 't Hooft anomaly between $Z_8$ and $\mathbb{Z}_2^{(1)}$ in the original theory, where the $\mathbb{Z}_2^{(1)}$ symmetry is not gauged.

As a final remark on the theory at the origin we note that the realization of r via electric-magnetic duality implies that the spectrum of massive charged dyons enjoys some degeneracy. In the duality frame (135) the action of r maps a dyon with charges $(n_m, n_e)$ to an exactly degenerate dyon with charges $(n_e, -n_m)$. Of course there is also the degeneracy $(n_m, n_e) \leftrightarrow (-n_m, -n_e)$ implied by charge conjugation. In particular this implies that the two hypermultiplets that become massless at the special points $u = \pm \Lambda^2$ are degenerate at the origin. In the duality frame (135) their charges are $(1, -1)$ and $(1, 1)$.

### 4.2.3 Monopole and dyon points

We now discuss the global symmetries and 't Hooft anomalies at the monopole and dyon points at $u = \Lambda^2$ and $u = -\Lambda^2$. Since they are related by the spontaneously broken $Z_8$ symmetry, we will focus on the monopole point. We work in the duality frame where $\varphi = a_D$, where the low-energy theory in the vicinity of the monopole point is described by $\mathcal{N} = 2$ SQED (see appendix B.2).

---

[61]This is true on spin manifolds. If $\mathcal{M}_4$ is not spin, the anomaly is valued in $\mathbb{Z}_4$.

The unbroken $Z_4$ symmetry acts on the fields in the abelian vector multiplet as in (131),

$$r^2(\varphi) = \varphi, \qquad r^2(\rho_\alpha^i) = -i\rho_\alpha^i, \qquad r^2(f^{(2)}) = -f^{(2)}. \tag{141}$$

This can be extended to a symmetry of SQED by defining the action on the hypermultiplet fields as follows,[62]

$$r^2(h_i) = \overline{h}_i, \qquad r^2(\overline{h}_i) = -h_i, \qquad r^2(\psi_{+\alpha}) = i\psi_{-\alpha}, \qquad r^2(\psi_{-\alpha}) = -i\psi_{+\alpha}. \tag{142}$$

Note that this exchanges fields of opposite charge, which is consistent with the fact that the action of $r^2$ on $f^{(2)}$ in (141) implements charge conjugation. For instance, the transformations (141) and (142) preserve the Yukawa couplings of the SQED lagrangian (see (B.18)),

$$\mathscr{L} \supset \sqrt{2}\left(\overline{h}_i\rho^i\psi_+ - h_i\rho^i\psi_-\right) + (\text{h.c.}). \tag{143}$$

Note that $r^4(\psi_{\alpha\pm}) = \psi_{\alpha\pm}$. Moreover, the fermions $\psi_{\alpha\pm}$ are neutral under $SU(2)_R$. The identification $r^4 = (-1)^F = -\mathbb{1}$ in (28) therefore does not hold. In particular, the fields $\psi_{\alpha\pm}$ do not satisfy the nonabelian spin-charge relation. This is not a contradiction, because they are charged under the dynamical $U(1)$ gauge field, i.e. they are not gauge invariant. (Moreover, as we will review below, this gauge field is actually a spin$^c$ connection.) By contrast, it is straightforward to check that $r^4 = (-1)^F = -\mathbb{1}$ does hold on all gauge-invariant local operators.

With the definitions above, the hypermultiplet fermions $\psi_{\pm\alpha}$ do not contribute to any 't Hooft anomalies. Therefore the anomaly matching at generic points of the Coulomb branch (see section 4.2.1) persists at the monopole and dyon points.

For future reference, we spell out the action of time-reversal $\mathsf{T}$ on the hypermultiplets. Since $\varphi = a_D$, time-reversal acts on the vector multiplet at the monopole point as in (134),

$$\mathsf{T}(\varphi) = -\varphi, \qquad \mathsf{T}(\rho_\alpha^i) = -i\rho_i^\alpha, \qquad \mathsf{T}(f^{(2)}) = -f^{(2)}. \tag{144}$$

Together with the form of the SQED lagrangian (B.18), (in particular the Yukawa couplings (143)) this fixes the action of $\mathsf{T}$ on the hypermultiplet fields,

$$\mathsf{T}(h_i) = h^i, \qquad \mathsf{T}(\overline{h}_i) = \overline{h}^i, \qquad \mathsf{T}(\psi_{+\alpha}) = -i\psi_+^\alpha, \qquad \mathsf{T}(\psi_{-\alpha}) = -i\psi_-^\alpha. \tag{145}$$

Observe that $\mathsf{T}^2 = -1$ on the fields in the hypermultiplet, while $\mathsf{T}^2 = 1$ on the fields in the vector multiplet.[63] Therefore the hypermultiplet fields (which are identified with a magnetic monopole at weak coupling) are Kramers doublets.

## 4.3 Considerations on non-spin manifolds

When $\mathcal{M}_4$ is not a spin manifold, there are additional interaction terms in the low-energy effective action on the Coulomb branch that involve $w_2(\mathcal{M}_4)$ [24]. Consider first the duality frame $\varphi = a$, which can loosely be thought of as arising from adjoint higgsing. It was shown in [24] that the action for the corresponding gauge field $f^{(2)}$ must be supplemented by the following term,

$$\exp\left(i\pi \int_{\mathcal{M}_4} \frac{f}{2\pi} \cup \widetilde{w}_2\right). \tag{146}$$

---

[62]The $\mathcal{N} = 2$ SQED lagrangian can be found in (B.18). Recall that hermitian conjugation acts on the hypermultiplet scalars via $(h_i)^\dagger = \overline{h}^i$ and $(h^i)^\dagger = -\overline{h}_i$.

[63]This sign is gauge invariant, because $\mathsf{T}$ preserves electric charge in this duality frame, so that mixing $\mathsf{T}$ with gauge transformations does not modify $\mathsf{T}^2$.

Here $\widetilde{w}_2$ is an integral lift of the second Stiefel-Whitney class $w_2(\mathcal{M}_4)$.

One consequence of the interaction (146) is that it turns 't Hooft lines of odd magnetic charge into fermions (see for instance [80]).[64] It is also plays a crucial role in matching the 't Hooft anomalies that involve the $\mathbb{Z}_2^{(1)}$ 1-form symmetry on non-spin manifolds. To describe these effects, we must first explain how the background gauge field $B_2$ corresponding to $\mathbb{Z}_2^{(1)}$ appears at low energies. Since activating $B_2$ turns $SU(2)$ gauge bundles into $SO(3)$ bundles with $w_2(SO(3)) = B_2$ and $f^{(2)}$ is obtained from $SU(2)$ by adjoint higgsing, it follows that $f^{(2)}$ satisfies the following flux quantization condition,

$$\frac{2f^{(2)}}{2\pi} = B_2 \ (\mathrm{mod}\ 2). \tag{148}$$

If $B_2 = 0$ then $f^{(2)}$ is conventionally normalized, but when $B_2$ is activated then $f^{(2)}$ admits magnetic monopoles of half-integer charge.

We can now describe how the interaction (146) depends on the lift $\widetilde{w}_2$. If we choose a different lift by shifting $\widetilde{w}_2 \rightarrow \widetilde{w}_2 + 2y$ (with $y$ an integral 2-cochain), we find that the partition function is multiplied by the following factor,

$$\exp\left( i\pi \int_{\mathcal{M}_4} B \cup y \right). \tag{149}$$

This is precisely the anomalous variation of the partition under the mixed anomaly discussed around (73).

We can also match the 't Hooft anomaly between the $\mathsf{Z}_4$ generator $\mathsf{r}^2$ and the $\mathbb{Z}_2^{(1)}$ 1-form symmetry discussed around (66). According to (131), $\mathsf{r}^2$ acts on the gauge field $f^{(2)}$ by charge conjugation. Applying such a transformation to the interaction (146), we find that the functional integral is modified by the following factor,

$$\exp\left( 2i\pi \int_{\mathcal{M}_4} \frac{f}{2\pi} \cup \widetilde{w}_2 \right) = \exp\left( i\pi \int_{\mathcal{M}_4} B_2 \cup w_2(\mathcal{M}_4) \right) = \exp\left( i\pi \int_{\mathcal{M}_4} B_2 \cup B_2 \right), \tag{150}$$

which matches (66) with $\mathsf{k}_r = 2$.

The effects described above in the duality frame defined by $\varphi = a$ propagate to other duality frames. For our purposes it will be sufficient to know what happens when $\varphi = a_D$. This was analyzed in [24]. Since $S$-duality transforms the 't Hooft line to a Wilson line and the interaction (146) implies that the 't Hooft line before duality is a fermion, it follows that the Wilson line after duality is a fermion. Consequently, the dual gauge field $f_D^{(2)}$ should be a spin$^c$ connection, which satisfies

$$\frac{2f_D^{(2)}}{2\pi} = \widetilde{w}_2 \ (\mathrm{mod}\ 2). \tag{151}$$

---

[64]To see this, consider an 't Hooft line of magnetic charge 1, so that $\frac{f^{(2)}}{2\pi}$ integrates to one over a small $S^2$ surrounding the line. The interaction then reduces to

$$\exp\left( i\pi \int_{\Sigma_2} w_2(N) \right), \tag{147}$$

where $\Sigma_2$ is a surface ending on the line and $w_2(N)$ is the second Stiefel-Whitney class of the normal bundle to $\Sigma$. Here we have used adjunction to reduce the bulk class $w_2(\mathcal{M}_4)$ on $\Sigma_2$. (We have $0 = w_1(\mathcal{M}_4)|_{\Sigma_2} = w_1(\Sigma_2) + w_1(N)$ and $0 = w_2(\mathcal{M}_4)|_{\Sigma_2} = w_2(\Sigma_2) + w_2(N) + w_1(N) \cup w_1(\mathcal{M}_4)$. We also use the fact that $w_2(\Sigma_2) + w_1(\Sigma_2) \cup w_1(\Sigma_2)$ always vanishes. Since (147) defines an SPT for the $SO(2)$ rotation symmetry of the normal bundle, the edge modes are in a projective representation of this $SO(2)$. This means that a $2\pi$ rotation acts as $-1$, i.e. the edge modes are fermions. Since the boundary of $\Sigma_2$ is the 't Hooft line, we conclude that the 't Hooft line is a fermion.

Recall that $\widetilde{w}_2$ is an integral lift of $w_2(\mathcal{M}_4)$.

Since $B_2$ couples to $f^{(2)}$ electrically (i.e. before duality), it must couple to $f_D^{(2)}$ magnetically (i.e. after duality). This coupling takes the following form,

$$\exp\left( i\pi \int_{\mathcal{M}_4} \frac{f_D^{(2)}}{2\pi} \cup \widetilde{B}_2 + \frac{i\pi}{2} \int_{\mathcal{M}_4} \widetilde{w}_2 \cup \widetilde{B}_2 \right). \tag{152}$$

The pure counterterm in this expression is fixed by the duality transformation, as was explicitly shown in [24].

As in the $\varphi = a$ duality frame discussed above, we can use (151) and (152) to match the 't Hooft anomalies that involve $B_2$ in the $\varphi = a_D$ duality frame. It is not an accident (see section 5.3) that the matching involves exactly the same manipulations as those carried out for the $\mathbb{CP}^1$ model in section 3.3.

# 5 Candidate phases for adjoint QCD from deformed $\mathcal{N} = 2$ SYM

In this section we will track the soft scalar mass deformation

$$\Delta V = \frac{M^2}{g^2} \overline{\phi}^A \phi^A, \tag{153}$$

from the $\mathcal{N} = 2$ SYM theory in the UV to the IR description on the Coulomb branch. As in [40], this is possible even though the operator (153) is not holomorphic, because this operator resides in the $\mathcal{N} = 2$ stress tensor supermultiplet. In fact, it is the bottom component of that multiplet. This leads to a streamlined derivation of some formulas in [38]. Note that the factor of $\frac{1}{g^2}$ is due to the non-canonical kinetic terms for $\phi^A$ in the $\mathcal{N} = 2$ lagrangian (37), so that $M$ is the classical pole mass of $\phi^A$.

We then analyze the effects of the mass deformation (153) on the Coulomb branch, expanding on the discussion in sections 1.5 and 1.6. The analysis is reliable when $M \ll \Lambda$ (where $\Lambda$ is the strong-coupling scale of the $\mathcal{N} = 2$ SYM theory) and leads to a unique vacuum at the origin $u = 0$ of the Coulomb branch. (This vacuum was also found in [38].) The massless fields in this vacuum are the IR gaugino $\rho_\alpha^i$ and a $U(1)$ gauge field at the $S$-duality invariant value $\tau = i$ of the gauge coupling. The physics of this vacuum was already discussed in sections 1.6 and 4.2.2.

We will also examine the mass deformation (153) near the monopole and dyon points. As was discussed below (13), a naive classical analysis suggests the existence of a meta-stable vacuum near these points. However, this analysis does not correctly capture the small-$M$ behavior of the deformed $\mathcal{N} = 2$ theory, because the low-energy $U(1)$ gauge coupling $e$ in the $\mathcal{N} = 2$ SQED theories that describe the monopole and dyon points is not sufficiently weak. As long as $M \ll \Lambda$, the only vacuum is therefore the one $u = 0$.

In the spirit of exploring consistent candidate phases for adjoint QCD that could emerge from the deformed $\mathcal{N} = 2$ theory when $M \gg \Lambda$, we nevertheless permit ourselves to explore the semiclassical regime $e \ll 1$ of the $\mathcal{N} = 2$ SQED theories at the monopole and dyon points. In this regime, the classical analysis is reliable, as long as $M$ is not too small (see below). Moreover, as we stressed throughout the paper, any phase obtained in this fashion automatically matches all 't Hooft anomalies of the UV theory.

In the $e \ll 1$ regime, the hypermultiplet scalars $h_i$ of the SQED theory condense, so that the $U(1)$ gauge group is higgsed. As in [9], this means that the UV theory confines via monopole condensation. Moreover, the monopole condensate spontaneously breaks the $SU(2)_R$ symmetry. The resulting low-energy theory consists of two copies of a $\mathbb{CP}^1$ sigma

model and was already discussed in section 3. It is remarkable that the massless degrees of freedom of the $\mathcal{N} = 2$ theory at the monopole and dyon points are sufficient to realize this chiral-symmetry breaking phase.

The embedding into the supersymmetric theory provides several additional handles. For instance, we will show that the monopole condensate in the IR in fact corresponds to a condensate of the gaugino bilinear $\mathcal{O}^{ij} = \lambda^{\alpha i A} \lambda_\alpha^{j A}$. Moreover, the deformed $\mathcal{N} = 2$ SQED theory provides a rigorous definition of the $\mathbb{CP}^1$ model that takes into account all global subtleties. Finally, we use the deformed SQED theory to explicitly exhibit the phase transition on the confining string that was discussed at the end of section (3.5).

## 5.1 The $\mathcal{N} = 2$ stress tensor multiplet

As was already mentioned above, the scalar $\mathcal{T} = \frac{1}{g^2} \overline{\phi}^A \phi^A$ that appears in the mass deformation (153) is the bottom component of the $\mathcal{N} = 2$ stress tensor supermultiplet. This multiplet was introduced in [47] (see also [90–93], as well as section 4.3 of [94] and section 5.5.2 of [95]). Here we will only briefly summarize its basic properties, since this will be sufficient for our purposes below.

In general, the $\mathcal{N} = 2$ stress tensor multiplet is a short multiplet of supersymmetry that is characterized by the following constraint,[65]

$$Q^{\alpha i} Q_\alpha^j \mathcal{T} = X^{ij}, \qquad X^{ij} = X^{(ij)}, \qquad Q_\alpha^{(i} X^{jk)} = \overline{Q}_{\dot\alpha}^{(i} X^{jk)} = 0. \tag{154}$$

The real scalar $\mathcal{T}$, which furnishes the bottom component of the multiplet, was already mentioned above. The complex $SU(2)_R$ triplet $X^{ij}$ is the bottom component of a non-trivial submultiplet that contains a complex abelian flavor current associated with the central charge in the $\mathcal{N} = 2$ supersymmetry algebra,[66] as well as the traces of the supersymmetry currents and the stress tensor. Therefore, the operator $X^{ij}$ vanishes if and only if the $\mathcal{N} = 2$ theory is superconformal.

Since pure $\mathcal{N} = 2$ SYM theory is classically conformal, the operator $X^{ij}$ vanishes at tree level. Using the supersymmetry transformations in (B.11) it can be checked that

$$\mathcal{T} = \frac{1}{g^2} \overline{\phi}^A \phi^A, \tag{155}$$

indeed satisfies the shortening condition (154) with $X^{ij} = 0$. Quantum mechanically, the theory is not conformal and the gauge coupling $g$ runs logarithmically. As a result, the operator $X^{ij} \sim \overline{Q}_{\dot\alpha}^i \overline{Q}^{\dot\alpha j} \overline{u}$ is generated at one-loop. (The coefficient is proportional to the one-loop $\beta$-function.) The details of this will not be important below.

A given supersymmetric field theory may admit distinct stress tensor multiplets that differ by improvement terms. Such improvements change the operators in the stress tensor multiplet while preserving the shortening condition (154). In particular, the conserved currents in the multiplet (including the stress tensor) are only modified by suitable total derivative terms that do not affect the corresponding conserved charges. The $\mathcal{N} = 2$ stress tensor multiplet (154) admits the following improvements (see for instance [93]),

$$\mathcal{T} \longrightarrow \mathcal{T} + \Phi + \overline{\Phi}, \qquad X^{ij} \longrightarrow X^{ij} + Q^{\alpha i} Q_\alpha^j \Phi, \qquad \overline{Q}_{\dot\alpha}^i \Phi = 0. \tag{156}$$

---

[65]This stress tensor multiplet is appropriate for conventional $\mathcal{N} = 2$ gauge theories, or deformed superconformal theories, which possess an $SU(2)_R$ symmetry. It does not, for instance, exist in theories where the $SU(2)_R$ symmetry is explicitly broken, such as $\mathcal{N} = 2$ SQED with a Fayet-Iliopoulos term.

[66]On the Coulomb branch, the central charge is also sensitive to certain boundary terms that arise from total derivatives in the supersymmetry current algebra. These terms are responsible for the familiar electromagnetic contributions to the central charge [96].

Here $\Phi$ is an $\mathcal{N} = 2$ chiral multiplet. In the UV, such a chiral multiplet can only be a holomorphic function $f(u)$ of the gauge-invariant chiral operator $u = \text{tr}(\phi)^2$ that parametrizes the Coulomb branch.[67] In the UV we are free to choose the stress tensor multiplet so that its bottom component is given by (155), without any improvements of the form $f(u) + \overline{f(u)}$. However, such improvements may well be generated when we flow to the deep IR. Below we will determine these improvements on the Coulomb branch of the $\mathcal{N} = 2$ SYM theory (see [38,40] for related discussions).

## 5.2 The stress tensor multiplet on the Coulomb branch

We would like to analyze the effect of the mass deformation (153) on the Coulomb branch of the $\mathcal{N} = 2$ theory. Thanks to (155), it suffices to track the bottom component $\mathcal{T} = \frac{1}{g^2}\overline{\phi}^A \phi^A$ of the $\mathcal{N} = 2$ stress tensor multiplet along the RG flow. In the deep IR, we would like to express $\mathcal{T}$ in terms of the special Coulomb branch coordinates $a$ and $a_D$. We claim that, away from the monopole and dyon points (see below), the operator $\mathcal{T}$ is given by the following expression,

$$\mathcal{T} = \frac{i}{4\pi}\left(a\overline{a}_D - \overline{a}a_D\right). \tag{157}$$

To justify this formula, we argue as follows:

1.) The right-hand side of (157) defines a valid stress tensor multiplet of the low-energy theory. This can be checked explicitly: if we use $a_D = \mathcal{F}'(a)$ (where $\mathcal{F}(a)$ is the prepotential), as well as the equations of motion that follow from the low-energy lagrangian (120), and the supersymmetry transformations in (B.20), we can verify that (157) satisfies the shortening condition (154) for a suitable choice for $X^{ij}$. This is particularly simple in a free $\mathcal{N} = 2$ $U(1)$ gauge theory (which is supeconformal), where $\mathcal{F}(a) = \frac{1}{2}\tau a^2$, with constant $\tau = \frac{\theta}{2\pi} + \frac{2\pi i}{e^2}$. In this theory $a_D = \tau a$, so that

$$\mathcal{T}_{\text{free}} = \frac{\text{Im}\,\tau}{2\pi}|a|^2 = \frac{1}{e^2}|a|^2, \tag{158}$$

which satisfies (154) with $X^{ij} = 0$, as expected for a superconformal theory.

Since (157) defines a valid stress-tensor multiplet for the IR theory, it must agree with the UV stress tensor multiplet up to an improvement term. We will now argue that no such improvement terms are needed to match the UV operator (155).

2.) Since the UV operator (155) is globally well-defined on the entire Coulomb branch, the same must be true of the corresponding IR operator. In particular, this operator must be invariant under $SL(2,\mathbb{Z})$ duality transformations. This is indeed the case for the expression in (157), because $(a_D, a)$ transform as an $SL(2,\mathbb{Z})$ doublet. Any possible improvements are therefore also globally well defined, and therefore determined by a globally holomorphic function $f(u)$ of the Coulomb-branch coordinate $u$ (see the UV discussion below (156)).

2.) Using the asymptotic expressions (126) and (127), we can compare the UV and IR operators in the region $|u| \gg \Lambda^2$. Up to subleading corrections, we find that[68]

$$\frac{1}{g^2}\overline{\phi}^A \phi^A \approx \frac{|u|}{2\pi^2}\log\frac{|u|}{\Lambda^2} \approx \frac{i}{4\pi}\left(a\overline{a}_D - \overline{a}a_D\right). \tag{159}$$

---

[67]In pure $\mathcal{N} = 2$ SYM theory, we can use such an improvement to make $X^{ij}$ real, so that it satisfies $\left(X^{ij}\right)^\dagger = X_{ij}$. This will not play a role in our discussion below.

[68]The $SU(2)$ gauge theory is higgsed to $U(1)$ by the adjoint scalar $\phi^3 = 2a$. Then $u = \text{tr}(\phi)^2 = \frac{1}{2}(\phi^3)^2 = 2a^2$. Since we use a normalization in which the $U(1)$ gauge field $f^{(2)}$ on the Coulomb branch has integral fluxes, $\frac{1}{2\pi}\int_{\Sigma_2} f^{(2)} \in \mathbb{Z}$, the $SU(2)$ gauge coupling $g$ and the $U(1)$ gauge coupling $e$ are related by $g = 2e$.

This implies that any improvement term that may be present must be subleading to $|u|\log|u|$ when $|u| \gg \Lambda^2$. Therefore the globally holomorphic function $f(u)$ that governs the possible improvements can at most be linear in $u$.

3.) To rule out the possibility of an improvement term of the form $f(u) \sim u$, we use the fact that the UV operator (155) is invariant under the global $\mathbb{Z}_8$ symmetry, while $u$ changes by a sign. To prove that such an improvement is absent in the IR, it therefore suffices to check that the IR expression (157) is also $\mathbb{Z}_8$ invariant. As discussed in section 4.2.2, the $\mathbb{Z}_8$ symmetry acts on the moduli space through a combination of $SL(2,\mathbb{Z})$ duality transformations and the discrete $R$-symmetry transformation (137), under which $a$ and $a_D$ rotate by the same phase (see also [97]). The expression in (157) is manifestly invariant under both of these transformations.

The preceding discussion shows that the UV operator (155) flows to the IR operator (157), up to an overall c-number constant that can be set to any value using a UV counterterm. (This corresponds to an improvement for which the function $f(u)$ is constant.) Since such a constant will play no role below, we set it to zero.

Having identified the IR expression (157) for the operator $\mathcal{T}$, and hence the soft scalar mass deformation (153), we have now justified the starting point for the discussion in sections 1.5 and 1.6. Indeed, it is straightforward to check that $u = 0$ is in fact a critical point of the mass deformation (153). We compute

$$\frac{d}{du}\big(a\bar{a}_D - \bar{a}a_D\big) = \frac{da}{du}\big(\bar{a}_D - \bar{a}\tau(a)\big). \tag{160}$$

Using the values $a = ix$, $a_D = x$, and $\tau = i$ from (136), we find that the right-hand side of (160) vanishes at the origin $u = 0$ of the Coulomb branch. Moreover examining Figure 1, it is clear that the origin is a global minimum and that the potential slopes rapidly upwards in all directions away from that point.

As discussed in section 1.6, the vacuum of the mass-deformed theory at $u = 0$ involves a massless gaugino $\rho_\alpha^i$ and a $U(1)$ gauge field with an $S$-duality invariant coupling constant $\tau = i$.

## 5.3 A candidate $\mathbb{CP}^1$ phase from the monopole and dyon points

As was already discussed at the beginning of this section, we will now repeat the analysis of the previous subsection near the monopole and dyon points. (Due to the $\mathbb{Z}_8$ symmetry it suffices to focus on the monopole point $u = \Lambda^2$.) In the spirit of generating a consistent candidate phase for adjoint QCD, we will dial the IR gauge coupling $e$ of the $\mathcal{N} = 2$ SQED theories at these points to very small values $e \ll 1$. In this regime we find a $\mathbb{CP}^1$ phase with confinement and chiral symmetry breaking, as described in section 3.

### 5.3.1 Matching the stress tensor multiplet

At the monopole point $u = \Lambda^2$ we work in the duality frame where $\varphi = a_D$, which vanishes at the monopole point. As we did on the Coulomb branch, we must first match the UV operator $\mathcal{T}$ in (155) in the low-energy SQED theory at the monopole point. Unlike in section (5.2), where we considered the full Coulomb-branch sigma model, we will here only discuss the immediate vicinity of the monopole point. As we will see, this is justified because we assume that $e \ll 1$. This will enable us to drop most higher-order terms in $\varphi$ (see however below). In this approximation, the $\mathcal{N} = 2$ theory at the monopole point is described by the renormalizable SQED lagrangian (B.18).

This theory is classically superconformal, so that the tree-level stress tensor multiplet bottom component $\mathcal{T}_0$ satisfies (154) with $X^{ij} = 0$. Using the supersymmetry transformations in (B.20) and (B.21), it can be checked that $\mathcal{T}_0$ is given by

$$\mathcal{T}_0 = \frac{1}{e^2} |\varphi|^2 - \frac{1}{2} \overline{h}^i h_i \,. \tag{161}$$

As was explained below (155), the operator $X^{ij}$ is generated at one-loop, because the SQED theory is not conformal and the $U(1)$ gauge coupling $e$ runs to zero in the deep IR.

We claim that the operator $\mathcal{T}_0$ in (161) is related to the UV operator $\mathcal{T}$ in (155) via

$$\mathcal{T} = \mathcal{T}_0 - \frac{i\Lambda}{2\pi^2} (\varphi - \overline{\varphi}) + \cdots = \frac{1}{e^2} |\varphi|^2 - \frac{1}{2} \overline{h}^i h_i - \frac{i\Lambda}{2\pi^2} (\varphi - \overline{\varphi}) + \cdots \,. \tag{162}$$

Here the term $\sim i(\varphi - \overline{\varphi})$ is an improvement term, and the ellipsis denotes higher-order improvement terms that vanish more rapidly at the monopole point. Unlike these higher-order terms, we can reliably match the linear term in (162) by only considering field excursions of $\varphi$ in the immediate vicinity of the monopole point, where our approximations are valid. In order to justify (162), we turn on a small non-zero vev for $\varphi$, so that the hypermultiplet becomes massive and can be integrated out. This turns the gauge coupling $e$ into a logarithmically running function of $\varphi$ that is captured by the behavior of the gauge coupling $\tau_D(a_D)$ near the monopole point. Importantly, integrating out the hypermultiplet does not generate any improvement terms.[69] We can then match the resulting expression to the Coulomb-branch formula (157). Near the monopole point, this leads to

$$\frac{i}{4\pi} \left( a \overline{a}_D - \overline{a} a_D \right) - \mathcal{T}_0 = \frac{i}{4\pi} \left( a \overline{a}_D - \overline{a} a_D \right) - \frac{\mathrm{Im}(\tau_D)}{2\pi} |a_D|^2 = -\frac{i\Lambda}{2\pi^2} (\varphi - \overline{\varphi}) + \cdots \,. \tag{163}$$

Note that the non-analytic terms that capture the logarithmic running of $e$ have canceled between $\frac{i}{4\pi} \left( a \overline{a}_D - \overline{a} a_D \right)$ and $\mathcal{T}_0$. This explains why $\mathcal{T}_0$ appears in (162) with unit coefficient. The term on the right-hand side of (163) is the improvement term in (162).

### 5.3.2 Analyzing the scalar potential

The full scalar potential $V$ of the $\mathcal{N} = 2$ SQED theory at the monopole point in the presence of the mass deformation $\Delta V$ in (153) is given by adding the supersymmetric potential (B.19) to $\Delta V = M^2 \mathcal{T}$, where $\mathcal{T}$ is the bottom component of the $\mathcal{N} = 2$ stress-tensor multiplet. Near the monopole point $\mathcal{T}$ is given by (162).

Before writing down an explicit formula for $V$ and analyzing its consequences, we choose the following parametrization for the complex scalar $\varphi$,

$$\varphi = \frac{-y + ix}{2} \,, \qquad x, y \in \mathbb{R} \,. \tag{164}$$

This parameterization is motivated by the fact that near the monopole point,[70]

$$\varphi = a_D(u) \approx \frac{i}{2\Lambda} \left( u - \Lambda^2 \right) \,, \tag{165}$$

so that (164) amounts to

$$u \approx \Lambda^2 + \Lambda (x + iy) \,. \tag{166}$$

---

[69]As we will discuss below, $\mathcal{N} = 2$ SQED has an accidental $U(1)_X$ superconformal $R$-symmetry that assigns charge 2 to $\varphi$. This symmetry is broken by an ABJ anomaly, but remains valid in perturbation theory. Since $\mathcal{T}_0$ is neutral under $U(1)_X$, while any improvements are necessarily charged, integrating out the massive hypermultiplet cannot generate such improvement terms.

[70]This can seen by expanding the exact formula for $a_D(u)$ in (128).

Therefore $x$ and $y$ parametrize real and imaginary motions in the complex $u$-pane, away from the monopole point.

In terms of the variables $x$, $y$ and $h_i$, the scalar potential $V$ is given by

$$V = \frac{e^2}{2}\left(\bar{h}^i h_i\right)^2 + \frac{1}{2}\left(x^2 + y^2\right)\bar{h}^i h_i + M^2\left(\frac{1}{4e^2}(x^2 + y^2) - \frac{1}{2}\bar{h}^i h_i + \frac{\Lambda x}{2\pi^2}\right). \quad (167)$$

As was explained above, we will analyze this potential in the regime $e \ll 1$ and $|x|, |y| \ll \Lambda$, i.e. close to the monopole point. Classically, there are two distinct possibilities:

a) There is an extremum of the potential in which the vev of $h_i$ vanishes,

$$h_i = 0, \qquad x = -\frac{\Lambda e^2}{\pi^2}, \qquad y = 0, \qquad V = -\frac{M^2 \Lambda^2 e^2}{4\pi^4}. \quad (168)$$

Note that $x$ is very close to the monopole point, because $e \ll 1$. The scalars $x$ and $y$ have large positive masses $m_{x,y}^2 \sim \frac{M^2}{e^2}$, while the mass of $h_i$ satisfies $m_h^2 \sim \frac{e^4 \Lambda^2}{\pi^4} - M^2$. This vacuum is therefore only stable if $M$ is sufficiently small,

$$M < \frac{e^2 \Lambda}{\pi^2}. \quad (169)$$

For larger values of $M$ the scalar $h_i$ becomes tachyonic.

b) There is another extremum where $h_i$ has a vev. The equations governing this extremum take the following form,

$$\bar{h}^i h_i = \frac{M^2 - x^2}{2e^2}, \qquad x^3 - 2M^2 x - \frac{e^2 M^2 \Lambda}{\pi^2} = 0, \qquad y = 0. \quad (170)$$

Since $\bar{h}^i h_i > 0$, we must restrict $|x| < M$. When $M \ll e^2 \Lambda$ these constrained equations do not have a solution. By contrast, when $M \gg e^2 \Lambda$ we find a minimum where

$$\bar{h}^i h_i = \frac{M^2}{2e^2}, \qquad x = -\frac{\Lambda e^2}{2\pi^2}, \qquad y = 0, \qquad V = -\frac{M^4}{8e^2}. \quad (171)$$

Here we have only retained the leading form of the solution when $M \gg e^2 \Lambda$. Note that $M$ may still be well below the scale $\Lambda$, since we assume that $e \ll 1$. As in (168) the vev of $x$ is very close to the monopole point, and $m_{x,y}^2 \sim \frac{M^2}{e^2}$. The mass of the radial mode of $h_i$ (i.e. the Higgs particle) is $m_h^2 \sim M^2$. It is also straightforward to analyze (170) in the intermediate region $M \sim e^2 \Lambda$, but this will not be necessary for the qualitative considerations below.

The preceding discussion shows that the vacuum in (168), where $h_i = 0$, is realized when $M$ is small. As we increase $M$ we eventually encounter a transition to the vacuum in (171), where $h_i$ condenses. When $M$ is sufficiently small, the nature of the vacua, as well as the transition between them, can be affected by the detailed structure of the potential (167), or by Coleman-Weinberg effects [98]. However, as long as $e \ll 1$, the Higgs vacuum (171) is reached for sufficiently large $M$. This is simply because the tachyonic mass term for $h_i$ in (167) eventually becomes dominant and forces $h_i$ to condense.

### 5.3.3   The $\mathbb{CP}^1$ model from deformed $\mathcal{N} = 2$ SQED

We now analyze the low-energy physics of the deformed $\mathcal{N} = 2$ SQED theory in the Higgs vacuum (171), which is realized for sufficiently large $M$. As we will see, this vacuum is described by the $\mathbb{CP}^1$ model analyzed in section 3. To see this, consider the masses of the various fields in the vacuum (171):

- The fact that $\overline{h}^i h_i \sim \frac{M^2}{e^2}$ implies that the $U(1)$ photon is higgsed and acquires a mass $m_\gamma^2 \sim M^2$.

- As was already mentioned below (171), the fields $x, y$ acquire masses $m_{x,y}^2 \sim \frac{M^2}{e^2}$, while the mass of the radial mode of the Higgs field $h_i$ is $m_h^2 \sim M^2$. The remaining three components of $h_i$ are massless and constrained to lie on the $S^3$ defined by the vev $\overline{h}^i h_i \sim \frac{M^2}{e^2}$. However, the Hopf fiber of the $S^3$ is acted on by $U(1)$ gauge transformations, and the corresponding massless scalar is eaten when the photon acquires a mass (see above). The remaining two massless scalars parametrize the space $S^3/U(1) = \mathbb{CP}^1$. The radius of this $\mathbb{CP}^1$ sigma model sets the pion decay constant $f_\pi^2 \sim \overline{h}^i h_i \sim \frac{M^2}{e^2}$ (see also section 5.3.4 below).

- The Yukawa couplings of $\mathcal{N} = 2$ SQED are given by (143) (see also (B.18)),

$$\mathscr{L} \supset \sqrt{2}\left(\overline{h}_i \rho^i \psi_+ - h_i \rho^i \psi_-\right) + (\text{h.c.}). \tag{172}$$

Since $h_i$ has a vev $\sim \frac{M}{e}$, all fermions acquire masses $\sim M$. (Recall that the gauginos $\rho_\alpha^i$ are not canonically normalized, while this is the case for $\psi_{\pm\alpha}$, see (B.18).)

We see that the long-distance physics is indeed described by a $\mathbb{CP}^1$ model. We can use the embedding of this model into the deformed $\mathcal{N} = 2$ SQED theory at the monopole point to motivate and clarify some of the more subtle points discussed in section 3:

1.) After higgsing, the gauginos $\rho_\alpha^i$ become massive. Since they carry the $SU(2)_R$ Witten anomaly, integrating them out generates a discrete theta angle for the $\mathbb{CP}^1$ model.

2.) After higgsing, the dynamical $U(1)$ gauge field $f^{(2)}$ of SQED flows to the Kähler form of the $\mathbb{CP}^1$ model,

$$\frac{f^{(2)}}{2\pi} \longrightarrow n^*(\omega). \tag{173}$$

We can also couple the SQED theory to $SU(2)_R$ background gauge fields $A_R^{(1)}$. In this case the dynamical $U(1)$ field strength $f^{(2)}$ flows to the Kähler form $n^*(\omega)(A_R^{(1)})$ coupled to $SU(2)_R$ background fields, which was defined in (103).

As we reviewed in section 4.3, the dynamical $U(1)$ gauge field $f^{(2)}$ at the monopole point is in fact a spin$^c$ connection. Therefore $\frac{2}{2\pi} f^{(2)}$ defines an integral cohomology class that satisfies the usual spin$^c$ constraint,

$$\frac{2}{2\pi} f^{(2)} = w_2 \,(\text{mod } 2). \tag{174}$$

Since $f^{(2)}$ flows to $n^*(\omega)(A_R^{(1)})$ after higgsing, we find the spin$^c$-like constraint on the pion fields in (105).

3.) The presence of the linear improvement term $\sim i\left(\varphi - \overline{\varphi}\right)$ in (163) leads to the tadpole term $\sim x$ in the scalar potential (167). This term is responsible for the negative expectation value of $x$ in the Higgs vacuum (171). It follows from (166) that $u$ is displaced away from the monopole point, and towards the origin $u = 0$ of the Coulomb branch. (More generally, we expect the scalar potential in Figure 1 to push us into the strong-coupling region.) As was explained around (130), the fact that $x < 0$ (i.e. that $u < \Lambda^2$) implies that there is a dynamical $\theta$-angle for the $U(1)$ gauge field $f^{(2)}$ that takes the non-trivial time-reversal invariant value $\theta = \pi$. Using the identification (173), we see that the $\mathbb{CP}^1$ model similarly acquires a $\theta$-angle (100) with $\theta = \pi$. We will further explore the implications of this fact in section 5.3.5 below.

4.) The $\mathcal{N} = 2$ SQED theory has an (accidental) superconformal $U(1)_X$ superconformal $R$-symmetry that assigns the following charges to the various fields,

$$q_X(\varphi) = 2, \qquad q_X\left(\rho_\alpha^i\right) = 1, \qquad q_X\left(f^{(2)}\right) = 0, \qquad q_X\left(h_i\right) = 0, \qquad q_X\left(\psi_{\pm\alpha}\right) = -1. \tag{175}$$

This symmetry suffers from an ABJ anomaly with the $U(1)$ gauge field that breaks it to its $\mathbb{Z}_2$ subgroup, which can be identified with fermion parity $(-1)^F$. The classically conserved $U(1)_X$ current $j_X^{(1)}$ therefore satisfies the following non-conservation equation,

$$d * j_X^{(1)} = \frac{1}{4\pi^2} f^{(2)} \wedge f^{(2)}. \tag{176}$$

We can locally express the right-hand side as $d * j_H^{(1)}$, where $j_H^{(1)} = \frac{1}{4\pi^2} b^{(1)} \wedge d b^{(2)}$ (here $f^{(2)} = d b^{(1)}$), but of course $j_H^{(1)}$ is not gauge invariant. After higgsing, $f^{(2)}$ flows to the $\mathbb{CP}^1$ Kähler form according to (173). In this case the charge $Q_H = \int_{S^3} * j_H^{(1)}$ is precisely the Hopf invariant of the map $S^3 \to \mathbb{CP}^1$ defined by the pion field. The fact that $j_H^{(1)}$ is not gauge invariant obstructs a satisfactory definition of $Q_H$ on general 3-manifolds. However, the $\mathbb{Z}_2 \subset U(1)_X$ subgroup is unbroken and identified with $(-1)^F$. We can therefore use (176) to conclude that $Q_H$ (mod 2) is also meaningful and equal to $(-1)^F$. This is consistent with the discussion around (95), as well as in [56].

### 5.3.4 Chiral condensate

Even though the deformed $\mathcal{N} = 2$ SQED theory we are discussing is not strictly speaking the one that arises from the UV $\mathcal{N} = 2$ SYM theory, we can nevertheless use supersymmetry to motivate relations between UV and IR operators. (These relations would be exact if we studied the SQED theory in the parameter regime dictated by the Seiberg-Witten solution.) Here we will do this for the gaugino condensate and verify that it has the expected properties discussed in section 3.

Recall from (76) that the gaugino condensate satisfies

$$\mathcal{O}^{ij} = \lambda^{\alpha i A} \lambda_\alpha^{jA} = -\frac{1}{2} Q^{\alpha i} Q_\alpha^j u + i \sqrt{2} \phi^A D^{ijA}. \tag{177}$$

If we are only interested in the expectation value of $\mathcal{O}^{ij}$ we can set the $D$-term to zero on-shell, since it appears quadratically in (37). As was reviewed around (129), the chiral operator $u$ can be expressed as follows on the Coulomb branch,

$$u = 2\pi i \left( \mathcal{F}(a) - \frac{1}{2} a a_D \right). \tag{178}$$

If we act on this expression with the supercharges (see (B.20)) and use $\mathcal{F}'(a) = a_D$, we can obtain an exact expression for $-\frac{1}{2} Q^{\alpha i} Q_\alpha^j u$ that involves the functions $a(u)$, $a_D(u)$ and $\tau(u)$.

Using (128) to expand these functions near the monopole point $u \approx \Lambda^2 + \Lambda(x + iy)$ (see (166)), we find that

$$-\frac{1}{2} Q^{\alpha i} Q_\alpha^j u = \left(-2\sqrt{2}\Lambda - \frac{1}{2\sqrt{2}}(x+iy)\right) D^{ij} - \left(\frac{1}{2} - \frac{3}{16\Lambda}(x+iy)\right) \rho^{\alpha i} \rho_\alpha^j + \cdots. \quad (179)$$

Here the ellipsis denotes terms that are subleading at the monopole point and $D^{ij}$ is the auxiliary $D$-term of the $\mathcal{N} = 2$ abelian vector multiplet in the SQED theory.

In the Higgs phase $x, y$, and $\rho_\alpha^i$ are massive and can be integrated out. This only leaves the $D$-term. Integrating it out (see (B.18)) sets $D^{ij} = 2ie^2 \bar{h}^{(i} h^{j)}$. Comparing with (177), we find a non-vanishing gaugino condensate (recall from (171) that $x$ is very close to the monopole point and can therefore be neglected),

$$\mathcal{O}_{ij} \approx -2\sqrt{2}\Lambda D_{ij} = -4\sqrt{2}\Lambda ie^2 h_{(i} \bar{h}_{j)} \sim M^2 \Lambda. \quad (180)$$

Note that since the $D$-term is real, the the same is true for the chiral condensate $\mathcal{O}^{ij}$, i.e. $\mathcal{O}^I$ is a real $SO(3)_R$ vector. This precisely matches our expectations from section 3.

### 5.3.5 A phase transition on the confining string

One virtue of embedding the $\mathbb{CP}^1$ phase into the deformed $\mathcal{N} = 2$ SQED theory is that the latter is UV complete. We can therefore use it to explore phenomena that are not captured by the low-energy $\mathbb{CP}^1$ model analyzed in section 3. We will now use the SQED description to exhibit the phase transition on the confining string discussed in section 3.5.

We begin by adding small fermion masses $m_{ij}$ in the UV, as in (74) and (78),

$$\Delta V = \frac{1}{2} m_{ij} \mathcal{O}^{ij} + \text{(h.c.)}, \qquad m_{ij} = e^{i\chi} \begin{pmatrix} \alpha & 0 \\ 0 & \beta \end{pmatrix}, \qquad \alpha, \beta \geq 0. \quad (181)$$

If we use the identification (177) and continue to drop the UV $D$-term,[71] the mass deformation reduces to

$$\Delta V = -\frac{1}{4} m_{ij} Q^{\alpha i} Q_\alpha^j u + \text{(h.c.)}. \quad (182)$$

We can now use (179) to analyze the effect of this mass deformation on the SQED theory in the IR. Working to linear order in the masses $m_{ij}$ and the field $\varphi = \frac{1}{2}(-y + ix)$, we can solve for the $D$-term of the abelian vector multiplet,

$$D_{ij} = 2ie^2 \left[ h_{(i} \bar{h}_{j)} + i\sqrt{2}\Lambda(m_{ij} + \overline{m}_{ij}) + \frac{1}{2\sqrt{2}}(m_{ij}\varphi - \overline{m}_{ij}\overline{\varphi}) \right]. \quad (183)$$

Using this $D$-term, we can recompute the scalar potential, which now takes the form

$$V = \frac{e^2}{2} \left(\bar{h}^i h_i\right)^2 + \frac{1}{2}\left(x^2 + y^2\right) \bar{h}^i h_i + M^2 \left(\frac{1}{4e^2}\left(x^2 + y^2\right) - \frac{1}{2}\bar{h}^i h_i + \frac{\Lambda x}{2\pi^2}\right)$$
$$+ 4\sqrt{2}\,\Lambda\,e^2\,\bar{h}^i h_i\,\text{Re}(m_I)\,n_I + \frac{e^2}{\sqrt{2}}\,\bar{h}^i h_i\,\left(x\,\text{Re}(m_I) - y\,\text{Im}(m_I)\right) n_I. \quad (184)$$

In this formula we have introduced the pion field $n_I$, which satisfies $n^I n^I = 1$, to describe the direction of the $SO(3)_R$ vector $h_{(i} \bar{h}_{j)}$. The components $m_I$ of the mass vector are related to the parameters in (181) as follows (see appendix A),

$$m_1 = \frac{i}{2} e^{i\chi}(\alpha - \beta), \qquad m_2 = -\frac{1}{2} e^{i\chi}(\alpha + \beta), \qquad m_3 = 0. \quad (185)$$

---

[71]When added to the lagrangian, a term $\sim \phi^A D^{ij} A$ gives rise to a mass term for the scalar $\phi^A$. Since we assume that $M$ is sufficiently large to lift all the scalars in the UV, we can neglect this term.

As in section 3.4, the term $\sim \mathrm{Re}(m_I)\,n_I$ in (184) gives the pion fields a mass. The additional information we get from (184) is how the UV fermion masses affect the critical point (171) that defines the $\mathbb{CP}^1$ phase. We will be particular interested in the expectation value of $\varphi$.

Let us choose the mass parameters so that the $U(1)_R \subset SU(2)_R$ symmetry is preserved. Recall from the discussion around (82) that this requires $\alpha = \beta$ in (185), so that $m_1 = 0$ and $m_2 = -\alpha e^{i\chi}$. Working to $\mathcal{O}(\alpha)$ in the mass parameters, we find that the critical point in (171) is perturbed as follows,

$$\bar{h}^i h_i = \frac{M^2}{2e^2} + 4\sqrt{2}\alpha |\cos\chi|\,, \qquad x = -\frac{\Lambda e^2}{2\pi^2} + \frac{\alpha e^2}{2\sqrt{2}}|\cos\chi|\,, \qquad y = -\frac{\alpha e^2}{2\sqrt{2}}\sin\chi\,. \qquad (186)$$

Here the absolute value on the cosine arises because we must take into account the correct minimum $n_2 = \pm 1$ for the pion fields. If the phase $\chi$ is generic, then $y$ acquires an expectation value, so that time-reversal symmetry is broken. However, when $\chi = 0$ or $\chi = \pi$, we find that $y = 0$ so that the time-reversal symmetry $\widetilde{\mathsf{T}} = r^2 \mathsf{T}$ is preserved.

As was discussed in section 3.5, the unbroken symmetries $U(1)_R$ and $\widetilde{\mathsf{T}}$ protect the topological insulator on the worldsheet of the confining string. Since the strings of pure YM theory with $\theta = 0$ (which is reached when the mass parameter $\alpha$ is large) are expected to be trivial, we were led to expect a phase transition on the string worldsheet at some critical value $\alpha = \alpha_*$ of the mass deformation.

We can now test this picture using the SQED description. As the fermion mass $\alpha$ is increased, it follows from (186) that $x$ (which starts out to the left of the monopole point) starts moving to the right (i.e. toward the monopole point). If we naively extrapolate the formulas (186), we find that at the critical value $\alpha_* \sim \Lambda$ the minimum reaches the monopole point and continues towards $x > 0$.

At the critical value $\alpha = \alpha_*$ of the mass parameter, there is a phase transition on the worldsheet of the confining string that involves a single massless two-dimensional Dirac fermion. To see this, note that when $x = y = 0$, the only mass term for the charged fermions $\psi_{\pm\alpha}$ of the SQED theory arises from the Yukawa couplings (172). (When $\varphi = \frac{1}{2}(-y + ix)$ is nonzero, there is another mass term $\sim \varphi\psi_+\psi_- + (\text{h.c.})$, see (B.18).) Although the Yukawa couplings keep all fermions massive in the four-dimensional bulk, there can be fermion zero modes on the ANO strings of the SQED theory in the Higgs phase. As was shown in [99, 100], the charged fermions $\psi_{\pm\alpha}$ give rise to chiral zero modes on the string worldsheet. Since the chiral is correlated with the electric charge of the fermions, we find both a left-moving and a right-moving chiral fermion in two dimensions. Together, they assemble into a single massless Dirac fermion.

This massless Dirac fermion on the string worldsheet is exactly what is needed to describe the transition between a two-dimensional topological insulator protected by $U(1)_R$ and $\widetilde{\mathsf{T}}$, and a trivial gapped phase. For $\alpha < \alpha_*$ the string worldsheet $\Sigma_2$ supports a topological insulator characterized by the background-field action

$$\exp\left(i\pi \int_{\Sigma_2} \frac{G_R^{(2)}}{2\pi}\right)\,. \qquad (187)$$

Here $G_R^{(2)}$ is the field strength of the background spin$^c$ gauge field for the $U(1)_R$ symmetry. Since the Higgs field $h_i$ is charged under both $U(1)_R \subset SU(2)_R$ and the $U(1)$ gauge symmetry, the unbroken $U(1)_R$ symmetry after higgsing mixes with the gauge symmetry. Therefore the fermions $\psi_{\pm\alpha}$ acquire $U(1)_R$ charges $\pm 1$ after higgsing, and hence the same is true of the two-dimensional Dirac fermion on the string worldsheet that becomes massless when $\alpha = \alpha_*$. Thus, dialing the mass of this Dirac fermion through zero causes the $U(1)_R$ background $\theta$-angle (187) to jump from $\theta = \pi$ to $\theta = 0$, so that the topological insulator gives way to a trivial gapped phase. This entire process takes place while the bulk theory remains gapped.

# Acknowledgments

We are grateful to G. Festuccia, K. Intriligator, J. Maldacena, P. Putrov, C. Vafa, and A. Vishwanath for useful discussions.

**Funding information** C.C. is supported by the Marvin L. Goldberger Membership at the Institute for Advanced Study and by DOE grant de-sc0009988. The work of TD is supported by the National Science Foundation under grant number PHY-1719924, and by the John Templeton Foundation under award number 52476.

# A  Conventions

In lorentzian signature, we follow the conventions of [1] for two-component spinor indices. In this appendix we adapt these conventions to $SU(2)_R$ doublet indices. (The resulting conventions are similar to those used for euclidean spacetime spinors in [101].) We consider $SU(2)_R$ doublets $v^i, w_i$ ($i = 1, 2$) with raised and lowered indices. Their $SU(2)_R$ transformations are such that $v^i w_i$ is invariant. Hermitian conjugation exchanges raised and lowered indices,

$$\left(v^i\right)^\dagger = \overline{v}_i, \qquad \left(w_i\right)^\dagger = \overline{w}^i. \tag{A.1}$$

$SU(2)_R$ doublet indices can be raised and lowered from the left using the antisymmetric symbols $\varepsilon^{ij}, \varepsilon_{ij}$, which are normalized as follows,

$$\varepsilon^{12} = -\varepsilon_{12} = 1. \tag{A.2}$$

Raising and lowering indices on both sides of the equations in (A.1) leads to a sign,

$$\left(v_i\right)^\dagger = -\overline{v}^i, \qquad \left(w^i\right)^\dagger = -\overline{w}_i. \tag{A.3}$$

A symmetric $SU(2)_R$ tensor $V^{ij} = V^{(ij)}$ transforms as a triplet. It is sometimes useful to convert $V^{(ij)}$ to an $SO(3)_R$ vector $V^I$ ($I = 1, 2, 3$). For this we use the $SU(2)_R$ Pauli matrices $\tau^I$, which we define with the following index structure,

$$\left(\tau^1\right)_i^{\ j} = \begin{pmatrix} 0 & 1 \\ 1 & 0 \end{pmatrix}, \qquad \left(\tau^2\right)_i^{\ j} = \begin{pmatrix} 0 & -i \\ i & 0 \end{pmatrix}, \qquad \left(\tau^3\right)_i^{\ j} = \begin{pmatrix} 1 & 0 \\ 0 & -1 \end{pmatrix}. \tag{A.4}$$

Since all $\tau^I$ are traceless, $\left(\tau^I\right)_i^{\ i} = 0$, it follows that $\tau^{Iij} = \tau^{I(ij)}$ and $\tau^I_{ij} = \tau^I_{(ij)}$ are symmetric. We can therefore use them to convert between the symmetric tensor $V^{ij}$ and the vector $V^I$,

$$V_{ij} = i\tau^I_{ij} V_I, \qquad V^I = \frac{i}{2}\tau^I_{ij} V^{ij}. \tag{A.5}$$

Explicitly,

$$\tau^I_{ij} = \left(-\tau^3, i\mathbb{1}, \tau^1\right), \qquad \tau^{Iij} = \left(\tau^3, i\mathbb{1}, -\tau^1\right), \tag{A.6}$$

so that (A.5) gives,

$$V^1 = -\frac{i}{2}\left(V^{11} - V^{22}\right), \qquad V^2 = -\frac{1}{2}\left(V^{11} + V^{22}\right), \qquad V^3 = iV^{12}. \tag{A.7}$$

From (A.6) we also see that

$$\left(i\tau^I_{ij}\right)^* = i\tau^{Iij}. \tag{A.8}$$

Therefore $\left(V_{ij}\right)^\dagger = i\tau^{Iij}\left(V_I\right)^\dagger$, so that $V_I$ is real if and only if $\left(V_{ij}\right)^\dagger = V^{ij}$.

# B   Review of $\mathcal{N} = 2$ supersymmetry

The $\mathcal{N} = 2$ supercharges and their hermitian conjugates are given by

$$Q_\alpha^i, \qquad \overline{Q}_{\dot\alpha i} = \left(Q_\alpha^i\right)^\dagger, \qquad i = 1, 2. \tag{B.1}$$

The $\mathcal{N} = 2$ supersymmetry algebra takes the following form,

$$\left\{Q_\alpha^i, \overline{Q}_{\dot\alpha j}\right\} = 2\delta^i{}_j \sigma^\mu_{\alpha\dot\alpha} P_\mu, \qquad \left\{Q_\alpha^i, Q_\beta^j\right\} = 2\varepsilon_{\alpha\beta}\varepsilon^{ij}\overline{Z}. \tag{B.2}$$

Here $Z$ is the complex central charge and $\overline{Z} = Z^\dagger$ is its hermitian conjugate. The algebra admits the following *R*-automorphism,

$$\frac{SU(2)_R \times U(1)_r}{\mathbb{Z}_2}. \tag{B.3}$$

Here the $\mathbb{Z}_2$ quotient identifies the central element $-\mathbb{1} \in SU(2)_R$ with $-1 \in U(1)_r$. The supercharges $Q_\alpha^i$ are $SU(2)_R$ doublets, while the central charge $Z$ is $SU(2)_R$ invariant. Under $U(1)_r$, the supercharges $Q_\alpha^i$ carry charge $-1$, while $Z$ has charge 2.

We can embed an $\mathcal{N} = 1$ supersymmetry algebra into (B.2) by choosing $Q_\alpha = Q_\alpha^1$ to be the $\mathcal{N} = 1$ supercharge, with hermitian conjugate $\overline{Q}_{\dot\alpha} = \overline{Q}_{\dot\alpha 1}$. When we discuss representations of (B.2) on fields or local operators $\mathcal{O}$, we take the supercharges to act from the left via nested graded (anti-) commutators, while $P_\mu \mathcal{O} = i\partial_\mu \mathcal{O}$.

## B.1   $SU(2)$ supersymmetric Yang-Mills theory

Pure $\mathcal{N} = 2$ supersymmetric Yang-Mills (SYM) theory with gauge group $SU(2)$ is based on a single, nonabelian $\mathcal{N} = 2$ vector multiplet $\mathcal{V}_{\mathcal{N}=2}^A$, which transforms in the adjoint representation of the $SU(2)$ gauge group,

$$\mathcal{V}_{\mathcal{N}=2}^A = \left(v_\mu^A, \ \phi^A, \ \lambda_\alpha^{iA}, \ D^{ijA}\right), \qquad D^{ijA} = D^{(ij)A} = \left(D_{ij}^A\right)^\dagger. \tag{B.4}$$

Under the $\mathcal{N} = 1$ supersymmetry algebra generated by $Q_\alpha = Q_\alpha^1$, it decomposes into an $\mathcal{N} = 1$ $SU(2)$ vector multiplet $\mathcal{V}^A$, and an $\mathcal{N} = 1$ chiral multiplet $\Phi^A$ in the adjoint representation of $SU(2)$,

$$\begin{aligned}
\mathcal{V}^A &= \left(v_\mu^A, \ \lambda_\alpha^A = i\lambda_\alpha^{2A}, \ D^A = iD^{12A} - i\varepsilon_{ABC}\overline{\phi}^B\phi^C\right), \\
\Phi^A &= \left(\phi^A, \ \lambda_\alpha^{1A}, \ F^A = \frac{i}{\sqrt{2}}D^{11A}\right).
\end{aligned} \tag{B.5}$$

Here $\lambda_\alpha^A$ is the $\mathcal{N} = 1$ gaugino, and $D^A$ is the corresponding $\mathcal{N} = 1$ $D$-term (note the shift relative to the $\mathcal{N} = 2$ $D$-term). We also define the quantities $\mathcal{V} = \mathcal{V}^A t^A$ and $\Phi = \Phi^A t^A$, where $t^A = \frac{1}{2}\sigma^A$ are the conventionally normalized, hermitian $SU(2)$ generators in (A.4),

$$\left[t^A, t^B\right] = i\varepsilon_{ABC} t^C, \qquad \mathrm{tr}\left(t^A t^B\right) = \frac{1}{2}\delta^{AB}. \tag{B.6}$$

We denote the nonabelian $\mathcal{N} = 1$ field strength superfield corresponding to $\mathcal{V}$ by $W_\alpha$.

In $\mathcal{N} = 1$ superspace, the $\mathcal{N} = 2$ SYM theory is defined by the following lagrangian,

$$\mathcal{L} = \frac{1}{g^2}\int d^4\theta \ \overline{\Phi}e^{-2\mathcal{V}}\Phi + \frac{1}{2g^2}\int d^2\theta \ \mathrm{tr}\left(W^\alpha W_\alpha\right) + (\text{h.c.}). \tag{B.7}$$

Using formulas in [1], this can be expanded in terms of the component fields in (B.4),

$$
\begin{aligned}
\mathcal{L} = \frac{1}{g^2}\Bigg( &-\frac{1}{4}v^A_{\mu\nu}v^{A\mu\nu} + \frac{1}{4}D^{ijA}D^A_{ij} - D^\mu\overline{\phi}^A D_\mu\phi^A - i\overline{\lambda}^A_i\overline{\sigma}^\mu D_\mu\lambda^{iA} \\
&-\frac{1}{2}\left(i\varepsilon_{ABC}\overline{\phi}^B\phi^C\right)^2 + \frac{i}{\sqrt{2}}\varepsilon_{ABC}\overline{\phi}^A\lambda^{iB}\lambda^C_i + \frac{i}{\sqrt{2}}\varepsilon_{ABC}\phi^A\overline{\lambda}^B_i\overline{\lambda}^{iC}\Bigg).
\end{aligned}
\tag{B.8}
$$

The $SU(2)$ field strength $v^A_{\mu\nu}$ is given by

$$
v^A_{\mu\nu} = \partial_\mu v^A_\nu - \partial_\nu v^A_\mu + \varepsilon_{ABC}v^B_\mu v^C_\nu,
\tag{B.9}
$$

while the covariant derivatives in (B.8) are given by

$$
D_\mu\phi^A = \partial_\mu\phi^A + \varepsilon_{ABC}v^B_\mu\phi^C, \qquad D_\mu\lambda^{iA}_\alpha = \partial_\mu\lambda^{iA}_\alpha + \varepsilon_{ABC}v^B_\mu\lambda^{iC}_\alpha.
\tag{B.10}
$$

The $\mathcal{N}=1$ supersymmetry transformation rules for the multiplets in (B.5) can be found in [1]. If we covariantize then with respect to the $SU(2)_R$ symmetry, we obtain the full $\mathcal{N}=2$ supersymmetry transformations of the vector multiplet (B.4),

$$
\begin{aligned}
&Q^i_\alpha\phi^A = i\sqrt{2}\lambda^{iA}_\alpha, \qquad \overline{Q}^i_{\dot\alpha}\phi^A = 0, \\
&Q^i_\alpha\lambda^{jA}_\beta = -\varepsilon^{ij}\left(\sigma^{\mu\nu}\right)_{\alpha\beta}v^A_{\mu\nu} + \varepsilon_{\alpha\beta}\left(D^{ijA} - \varepsilon^{ij}\varepsilon_{ABC}\overline{\phi}^B\phi^C\right), \qquad \overline{Q}^i_{\dot\alpha}\lambda^{jA}_\alpha = \varepsilon^{ij}\sqrt{2}\sigma^\mu_{\alpha\dot\alpha}D_\mu\phi^A, \\
&Q^i_\alpha v^A_\mu = i\sigma_{\mu\alpha\dot\alpha}\overline{\lambda}^{\dot\alpha iA}, \qquad \overline{Q}^i_{\dot\alpha}v^A_\mu = -i\sigma_{\mu\alpha\dot\alpha}\lambda^{\alpha iA}, \\
&Q^i_\alpha D^{jkA} = i\left(\varepsilon^{ij}\sigma^\mu_{\alpha\dot\alpha}D_\mu\overline{\lambda}^{\dot\alpha kA} + \varepsilon^{ik}\sigma^\mu_{\alpha\dot\alpha}D_\mu\overline{\lambda}^{\dot\alpha jA}\right) + i\sqrt{2}\varepsilon_{ABC}\overline{\phi}^B\left(\varepsilon^{ij}\lambda^{kC}_\alpha + \varepsilon^{ik}\lambda^{jC}_\alpha\right), \\
&\overline{Q}^i_{\dot\alpha}D^{jkA} = -i\left(\varepsilon^{ij}\sigma^\mu_{\alpha\dot\alpha}D_\mu\lambda^{\alpha kA} + \varepsilon^{ik}\sigma^\mu_{\alpha\dot\alpha}D_\mu\lambda^{\alpha jA}\right) + i\sqrt{2}\varepsilon_{ABC}\phi^B\left(\varepsilon^{ij}\overline{\lambda}^{kC}_{\dot\alpha} + \varepsilon^{ik}\overline{\lambda}^{jC}_{\dot\alpha}\right).
\end{aligned}
\tag{B.11}
$$

These transformations obey the supersymmetry algebra (B.2) (with $Z = 0$) off shell, and modulo gauge transformations (since we have fixed Wess-Zumino gauge).

## B.2 Supersymmetric QED

At the monopole point $u = \Lambda^2$, the low-energy dynamics is described by massless $\mathcal{N}=2$ supersymmetric QED (SQED). This theory involves two different $\mathcal{N}=2$ multiplets:

- A $U(1)$ vector multiplet $\mathcal{V}_{\mathcal{N}=2}$, which takes the same form as in (B.4),

$$
\mathcal{V}_{\mathcal{N}=2} = \left(\varphi, \rho^i_\alpha, f^{(2)} = db^{(1)}, D^{ij}\right), \qquad D^{ij} = D^{(ij)} = \left(D_{ij}\right)^\dagger.
\tag{B.12}
$$

Its decomposition under the $\mathcal{N}=1$ supersymmetry algebra generated by $Q_\alpha = Q^1_\alpha$ consists of an $\mathcal{N}=1$ vector multiplet $\mathcal{V}$, with field strength $W_\alpha$, as well as an $\mathcal{N}=1$ chiral multiplet $\Phi$,

$$
\mathcal{V} = \left(b^{(1)}, \lambda_\alpha = i\rho^2_\alpha, D = iD^{12}\right), \qquad \Phi = \left(\varphi, \rho^1_\alpha, F_\varphi = \frac{i}{\sqrt{2}}D^{11}\right).
\tag{B.13}
$$

Here $\lambda_\alpha$ is the $\mathcal{N}=1$ gaugino, while $D$ and $F_\varphi$ are $\mathcal{N}=1$ auxiliary fields.

- An $\mathcal{N}=2$ hypermultiplet $\mathcal{H}_i$ of $U(1)$ charge $+1$, and its hermitian conjugate $\overline{\mathcal{H}}^i$ of $U(1)$ charge $-1$,

$$
\mathcal{H}_i = \left(h_i, \psi_{+\alpha}, \overline{\psi}_{-\dot\alpha}\right), \qquad \overline{\mathcal{H}}^i = \left(\overline{h}^i, \overline{\psi}_{+\dot\alpha}, \psi_{-\alpha}\right).
\tag{B.14}
$$

Note that, unlike the supercharges $Q_\alpha^i$ or the gauginos $\rho_\alpha^i$, we define $h_i$ with lower indices, so that hermitian conjugation acts as follows,

$$\overline{h}^i = \left(h_i\right)^\dagger, \qquad \overline{h}_i = -\left(h^i\right)^\dagger. \tag{B.15}$$

The hypermultiplet decomposes into two $\mathcal{N} = 1$ chiral multiplets $\mathcal{H}_\pm$ of $U(1)$ charge $\pm 1$,

$$\mathcal{H}_+ = \left(h_1, \psi_+, F_+\right), \qquad \mathcal{H}_- = \left(\overline{h}^2, \psi_-, F_-\right). \tag{B.16}$$

Here $F_\pm$ are $\mathcal{N} = 1$ auxiliary fields.

In $\mathcal{N} = 1$ superspace, the lagrangian of $\mathcal{N} = 2$ SQED takes the following form,

$$\mathscr{L} = \int d^4\theta \left(\frac{1}{e^2}\overline{\Phi}\Phi + \overline{\mathcal{H}}_+ e^{-2V}\mathcal{H}_+ + \overline{\mathcal{H}}_- e^{2V}\mathcal{H}_-\right) + \int d^2\theta \left(\frac{1}{4e^2}W^\alpha W_\alpha + \sqrt{2}\Phi\mathcal{H}_+\mathcal{H}_-\right) + \text{(h.c.)}. \tag{B.17}$$

Using formulas in [1], as well as (B.13) and (B.16), and integrating out the auxiliary fields $F_\pm$ (which have no simple uplift to $\mathcal{N} = 2$), but not $D, F_\varphi$ (which assemble into the $\mathcal{N} = 2$ auxiliary field $D^{ij}$), we find that (B.17) gives rise to the following component lagrangian,

$$\begin{aligned}
\mathscr{L} = &\frac{1}{e^2}\left(-\partial^\mu\overline{\varphi}\partial_\mu\varphi - i\overline{\rho}_i\overline{\sigma}^\mu\partial_\mu\rho^i - \frac{1}{4}f^{\mu\nu}f_{\mu\nu} + \frac{1}{4}D^{ij}D_{ij}\right) \\
&- \left(\partial^\mu + ib^\mu\right)\overline{h}^i\left(\partial_\mu - ib_\mu\right)h_i - i\overline{\psi}_+\overline{\sigma}^\mu\left(\partial_\mu - ib_\mu\right)\psi_+ - i\overline{\psi}_-\overline{\sigma}^\mu\left(\partial_\mu + ib_\mu\right)\psi_- \\
&- iD^{ij}h_i\overline{h}_j - 2|\varphi|^2\overline{h}^i h_i - \sqrt{2}\left(\varphi\psi_+\psi_- + \overline{\varphi}\overline{\psi}_+\overline{\psi}_-\right) \\
&+ \sqrt{2}\left(\overline{h}_i\rho^i\psi_+ - h_i\rho^i\psi_- - h^i\overline{\rho}_i\overline{\psi}_+ - \overline{h}^i\overline{\rho}_i\overline{\psi}_-\right).
\end{aligned} \tag{B.18}$$

Note that the Yukawa couplings in the last line are real, because $h_i$ satisfies (B.15). If we integrate out the auxiliary field $D_{ij} = 2ie^2 h_{(i}\overline{h}_{j)}$, we find the following scalar potential,

$$V = \frac{e^2}{2}\left(\overline{h}^i h_i\right)^2 + 2|\varphi|^2\overline{h}^i h_i. \tag{B.19}$$

The supersymmetry transformations of the $\mathcal{N} = 1$ multiplets in (B.13) and (B.16) can be found in [1]. By covariantizing these formulas with respect to $SU(2)_R$, we deduce the full $\mathcal{N} = 2$ supersymmetry transformations. For the abelian vector multiplet $\mathcal{V}_{\mathcal{N}=2}$ in (B.12), they can be obtained from the nonabelian transformation rules (B.11) by restricting to a $U(1) \subset SU(2)$ subalgebra,

$$\begin{aligned}
Q_\alpha^i\varphi &= i\sqrt{2}\rho_\alpha^i, & \overline{Q}_{\dot\alpha}^i\varphi &= 0, \\
Q_\alpha^i\rho_\beta^j &= \varepsilon_{\alpha\beta}D^{ij} - \varepsilon^{ij}\left(\sigma^{\mu\nu}\right)_{\alpha\beta}f_{\mu\nu}, & \overline{Q}_{\dot\alpha}^i\rho_\alpha^j &= \varepsilon^{ij}\sqrt{2}\sigma^\mu_{\alpha\dot\alpha}\partial_\mu\varphi, \\
Q_\alpha^i D^{jk} &= i\left(\varepsilon^{ij}\sigma^\mu_{\alpha\dot\alpha}\partial_\mu\overline{\rho}^{\dot\alpha k} + \varepsilon^{ik}\sigma^\mu_{\alpha\dot\alpha}\partial_\mu\overline{\rho}^{\dot\alpha j}\right), & \overline{Q}_{\dot\alpha}^i D^{jk} &= -i\left(\varepsilon^{ij}\sigma^\mu_{\alpha\dot\alpha}\partial_\mu\rho^{\alpha k} + \varepsilon^{ik}\sigma^\mu_{\alpha\dot\alpha}\partial_\mu\rho^{\alpha j}\right), \\
Q_\alpha^i f_{\mu\nu} &= -i\left(\sigma_{\mu\alpha\dot\alpha}\partial_\nu\overline{\rho}^{\dot\alpha i} - \sigma_{\nu\alpha\dot\alpha}\partial_\mu\overline{\rho}^{\dot\alpha i}\right), & \overline{Q}_{\dot\alpha}^i f_{\mu\nu} &= i\left(\sigma_{\mu\alpha\dot\alpha}\partial_\nu\rho^{\alpha i} - \sigma_{\nu\alpha\dot\alpha}\partial_\mu\rho^{\alpha i}\right).
\end{aligned} \tag{B.20}$$

The $\mathcal{N} = 2$ supersymmetry transformations of the hypermultiplet (B.14) are

$$\begin{aligned}
Q_\alpha^i h^j &= -i\sqrt{2}\varepsilon^{ij}\psi_{+\alpha}, & \overline{Q}_{\dot\alpha}^i h^j &= i\sqrt{2}\varepsilon^{ij}\overline{\psi}_{-\dot\alpha}, \\
Q_\alpha^i\overline{h}^j &= i\sqrt{2}\varepsilon^{ij}\psi_{-\alpha}, & \overline{Q}_{\dot\alpha}^i\overline{h}^j &= i\sqrt{2}\varepsilon^{ij}\overline{\psi}_{+\dot\alpha}, \\
Q_\alpha^i\psi_{+\beta} &= 2i\varepsilon_{\alpha\beta}\overline{\varphi}h^i, & \overline{Q}_{\dot\alpha}^i\psi_{+\alpha} &= \sqrt{2}\sigma^\mu_{\alpha\dot\alpha}\left(\partial_\mu - ib_\mu\right)h^i, \\
Q_\alpha^i\psi_{-\beta} &= 2i\varepsilon_{\alpha\beta}\overline{\varphi}\overline{h}^i, & \overline{Q}_{\dot\alpha}^i\psi_{-\alpha} &= -\sqrt{2}\sigma^\mu_{\alpha\dot\alpha}\left(\partial_\mu + ib_\mu\right)\overline{h}^i.
\end{aligned} \tag{B.21}$$

These transformations only close on-shell, because we have integrated out the $\mathcal{N} = 1$ auxiliary fields $F_{\pm}$, as well as modulo gauge transformations, because we are working in Wess-Zumino gauge. The latter fact is also responsible for the covariant derivatives in (B.21).

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
