# Peer review of "Candidate Phases for SU(2) Adjoint QCD$_4$ with Two Flavors from $\mathcal{N}=2$ Supersymmetric Yang-Mills Theory"

_SciPost Physics, doi:SciPost Phys. 16, 139 (2024)_

## Round 1 · Referee Report · Anonymous (Referee 1) · 2023-11-3

Report

This article investigates the possible infrared phases of the $SU(2)$ gauge theory with two Weyl fermion in the adjoint representation. The authors adopted an elegant quantitative method by first considering the 4d $\mathcal{N}=2$ $SU(2)$ super Yang-Mills theory and then deforming it by a non-holomorphic scalar mass term that corresponds to the bottom component of the $\mathcal{N}=2$ stress tensor multiplet. The latter flows to the theory in question in the IR. This deformation preserves all (zero and higher form) global symmetries of the super Yang-Mills theory, except supersymmetry, as well as all ’t Hooft anomalies. The analysis is reliable for a small mass deformation, and the authors proposed possible scenarios for a larger mass deformation base on 't Hooft anomaly matching. At the monopole and dyon points, the authors proposed that the IR phase is described by the $\mathbb{CP}^1$ sigma model. The authors also proposed an exotic phase at generic point on the real axis of the $u$-plane of the super Yang-Mills theory, where there is a confinement characterized by the unbroken $\mathbb{Z}_2$ one-form center symmetry but with the existence of the long-range $\mathbb{Z}_2$ gauge field.

In the referee's opinion, this is a very important article that surely will become a standard reference on the subject. It should definitely be accepted for publication.

Requested changes

There are some minor suggestions from the referee: 1. There is a typo (Time-time-reversalreversal) above Eq. (3.9). 2. The manuscript was uploaded to the arXiv since 2018. Since then there surely have been developments in the subject, the authors should briefly comments on those, particularly on the $SU(N)$ gauge theory with $N>2$.

---

## Round 1 · Referee Report · Anonymous (Referee 2) · 2023-11-5

Strengths

1 - Detailed and rigorous analysis of the symmetries of adjoint SU(2) SQCD with two fermions, and of the 4d N=2 supersymmetric theory it can be embedded into.
2 - Thorough discussion of the possible infrared phases of the theory based on the exact knowledge of supersymmetric vacua.
3 - Clear distinction between rigorous results and conjectures/possibilities.

Weaknesses

None.

Report

The authors explore the possible vacua of the 4d SU(2) gauge theory with two adjoint fermions, based on embedding it into the 4d N=2 SYM theory and giving a mass to the scalars that preserves all the symmetries. They give a detailed discussions of the (0- and 1-form) symmetries of these theories, and analyse how the symmetries and their 't Hooft anomalies are matched in the various (T-preserving) vacua on the Coulomb branch of the SYM theory. That part of the analysis, based on the Seiberg-Witten solution, is rigorous and exact, and provides new subtle checks of the SW solution itself.

In parallel, they explore possible phases for the Nf=2 adjoint SQCD theory, based on deformation of the SUSY theory by a symmetry-preserving mass term. Of course, they cannot reach rigorous results for large mass, but they give a convincing picture of the possible vacua under the assumption that the true vacua is related to the CB vacua of the SYM theory. In particular, they discuss in detail some CP1+CP1 vacuum corresponding to SSB of SU(2)--> U(1) and show how all 't Hooft anomalies are non-trivially matched.

This is a stimulating paper and I strongly recommend it for publication in SciPost. The paper appeared on the arXiv 5 years ago, and it would be useful to add some comments on later developments since then. (See the naive comments I add below.)

Requested changes

Let me list here some technical questions that I came across. Below I also list some typos I encountered.

-middle of page 4: for SU(Nc) Nf adjoint, it is written that the axial symmetry is reduced to Z_4Nc by the ABJ anomaly. Shouldn't it be Z_{2 Nf Nc} ?

-footnote 55: it is written that the monodromies lie in Gamma_0(4). In fact, given your conventions (4.5), the monodromies are in Gamma^0(4). Indeed the monodromy at infinity (as u-> e^{2\pi i} u) is -T^4, as seen from (4.7).

-a side comment that might be of interest to the authors: there is a quick way to derive (4.11) and the value of \tau in (4.17) using the biholomorphism between the u-plane and the \tau upper-half-plane. That explicit map was recently revisited e.g. in https://arxiv.org/abs/2107.04600 and most recently in https://arxiv.org/abs/2308.10225. The fundamental domain of \Gamma^0(4), corresponding to the \tau-plane, is shown in figure 4 of the former paper, and in figure 1 of the latter paper. In your conventions, one just needs to flip the sign of Re(\tau) in that figure 1. Then, the real axis between the two SW singularities corresponds to the half-circle |\tau+1|^2=1, from which (4.11) follows directly.

-Finally, related to this latter work (2308.10225) where the SW curve for N=2 SYM with SO(3)_\pm gauge group was discussed in some detail. I wonder if the authors have thought about the infrared phase of SO(3) SQCD with 2 vectors? It would be interesting to see how gauging the one-form symmetry affects the possible infrared phases. (For instance, I think we loose the Z_2 symmetry that exchanges the two copies of CP1 in the sigma-model phase.)

List of typos:
-after (1.2), there is a missing reference ("see for instance [?]")
-p6: "from now own" should be "from now on"
-p6 and afterwards: there are several references to eq. (2.61) where you mean (1.4).
-p9: "perimeter low" should be "perimeter law"
-p44: "time-time reversalreversal"
-above (B.6): missing reference "(??)"

---

## Round 2 · Referee Report · Anonymous (Referee 2) · 2024-5-7

Report

The authors added useful corrections. This great paper should be published.

Recommendation

Publish (surpasses expectations and criteria for this Journal; among top 10%)

---

## Round 2 · Referee Report · Anonymous (Referee 1) · 2024-5-7

Report

The authors have corrected minor typos in the article. It should be ready for publication.

Recommendation

Publish (surpasses expectations and criteria for this Journal; among top 10%)

---

## Round 2 · List of Changes

We thank the referees for their comments. In response we have made the following changes:

1) corrected all identified typos

2) added the final paragraph of the introduction with reference to more recent work

3) corrected mistake on page 4: 4Nc-> 2NfNc

4) corrected footnote 55

5) added footnote 60 about the biholomorphism between the u plane and the tau plane (with related references)

Finally with regard to the questions about different global forms of the gauge group, eg SO(3) we have not investigated the question in detail, but view it as an interesting topic for future investigations, both in this model and in more general theories with gauge group lie algebra SU(N).

---

## Editorial Decision

published